# Emergent discrete space-time crystal of Majorana-like quasiparticles in chiral liquid crystals

Hanqing Zhao[1,2], Rui Zhang [2,3,4] & Ivan I. Smalyukh [1,2,5,6] ✉

Time crystals spontaneously break the time translation symmetry, as recently has been frequently reported in quantum systems. Here we describe the observation of classical analogs of both 1 + 1-dimensional and 2 + 1-dimensional discrete space-time crystals in a liquid crystal system driven by a Floquet electrical signal. These classical time crystals comprise particle-like structural features and exist over a wide range of temperatures and electrical driving conditions. The phenomenon-enabling period-doubling effect in 1 + 1-dimensional discrete space-time crystals comes from their topological Majorana-like quasiparticle features, where periodic inter-transformations of co-existing topological solitons and disclinations emerge in response to external stimuli and play pivotal roles. Our discrete space-time crystals exhibit robustness against temporal perturbations and spatial defects. Our findings show that the simultaneous symmetry breaking in time and space can be a widespread occurrence in numerous open systems, not only in quantum but also in a classical soft matter context.

Time crystals, originally proposed by Wilczek a decade ago[1,2], have captured the interest of numerous researchers who are fascinated by the search for these new "crystals". In analogy to the conventional space crystals, Wilczek's original proposal suggested that time translation symmetry can be spontaneously broken in the lowest energy states of closed many-body systems, both in a quantum and classical manner[1,2]. Unfortunately, a series of no-go theorems demonstrate that these closed systems are prohibited by nature at equilibrium[3–5]. However, in non-equilibrium situations with Floquet external drives, it is possible to break time translation symmetry discretely, where the period of the system's dynamics is an integer multiple of that of the external drive (the integer must be larger than one), resulting in what is called a discrete (Floquet) time crystal[6–9]. The discrete time crystals were first observed in systems of nuclear spins and trapped ions[10,11], and recently, many-body localization discrete time crystals were observed in quantum simulators and processors[12–14], which use the many-body localization to prevent the system from a thermalized fate. In addition to the many-body localization, discrete time crystals can be made to persist for a long time by incorporating dissipation[15,16] or introducing a prethermal state[17], which are also predicted for classical systems[18–20]. Indeed, the so-called "continuous time crystals" (a different type of time crystals with spontaneous symmetry breaking driven by a constant external source) have been recently observed in both quantum and classical systems, including in liquid crystals (LCs)[21–23]. While quantum mechanical discrete time crystalline effects received a great deal of attention, their long-awaited discovery in classical systems may be equally important from both fundamental and applied perspectives[24,25].

Here, we report the observation of classical discrete space-time crystals (DSTCs) in a chiral nematic LC system widely known for its

[1]Department of Physics, University of Colorado, Boulder, CO, USA. [2]International Institute for Sustainability with Knotted Chiral Meta Matter (WPI-SKCM²), Hiroshima University, Higashi Hiroshima, Hiroshima, Japan. [3]Department of Physics, The Hong Kong University of Science and Technology, Clear Water Bay, Kowloon, Hong Kong, People's Republic of China. [4]State Key Laboratory of Displays and Opto-Electronics, The Hong Kong University of Science and Technology, Clear Water Bay, Kowloon, Hong Kong, People's Republic of China. [5]Materials Science and Engineering Program, University of Colorado, Boulder, CO, USA. [6]Renewable and Sustainable Energy Institute, National Renewable Energy Laboratory, and University of Colorado, Boulder, CO, USA. ✉e-mail: ivan.smalyukh@colorado.edu

widespread technological use. Electrical switching of LCs is at the heart of the modern LC-enabled trillion-dollar industries, including information displays and electro-optic devices, but the possibility of discrete-time-crystal emergence in these soft matter systems was never analysed. By applying a Floquet electrical signal to a chiral nematic LC sandwiched between parallel electrodes, we find that both the spatial and temporal symmetries of the emergent LC's structure revealed by optical images can be broken discretely and spontaneously under well-defined experimental conditions, and the internal temporal periodicity of the system doubles in relation to the external drive. Both 1 + 1 dimensional (1 + 1D) and 2 + 1 dimensional (2 + 1D) discrete space-time crystals are observed, with the time crystallization phases depending on temperature and external driving parameters, as illustrated by constructing a phase diagram. Utilizing computer simulations, we illustrate that the period-doubling effect is intimately related to the inter-transformations, generations, and annihilations of

coexisting topological solitons and singular disclinations. Remarkably, the different states of these topological objects can be viewed as the particle and anti-particle states of the observed Majorana-like quasi-particles (a classical analog of Majorana particles[26–28]) forming our 1 + 1D space-time crystals. We verify the rigidity (robustness) of the time crystals against temporal perturbation and spatial defects, with the DSTC phase maintaining order locally for a remarkably long time. Moreover, a potential candidate for a fractional discrete space-time crystal is observed when changing the sample thickness. Our findings may lead to a new paradigm of time-crystalline LC meta matter, with potential fundamental science impacts and technological utility.

## Results

### Emergent space-time crystal in a chiral nematic medium

A typical studied sample is prepared by sandwiching a chiral nematic LC between two electrically conductive transparent substrates (Fig. 1),

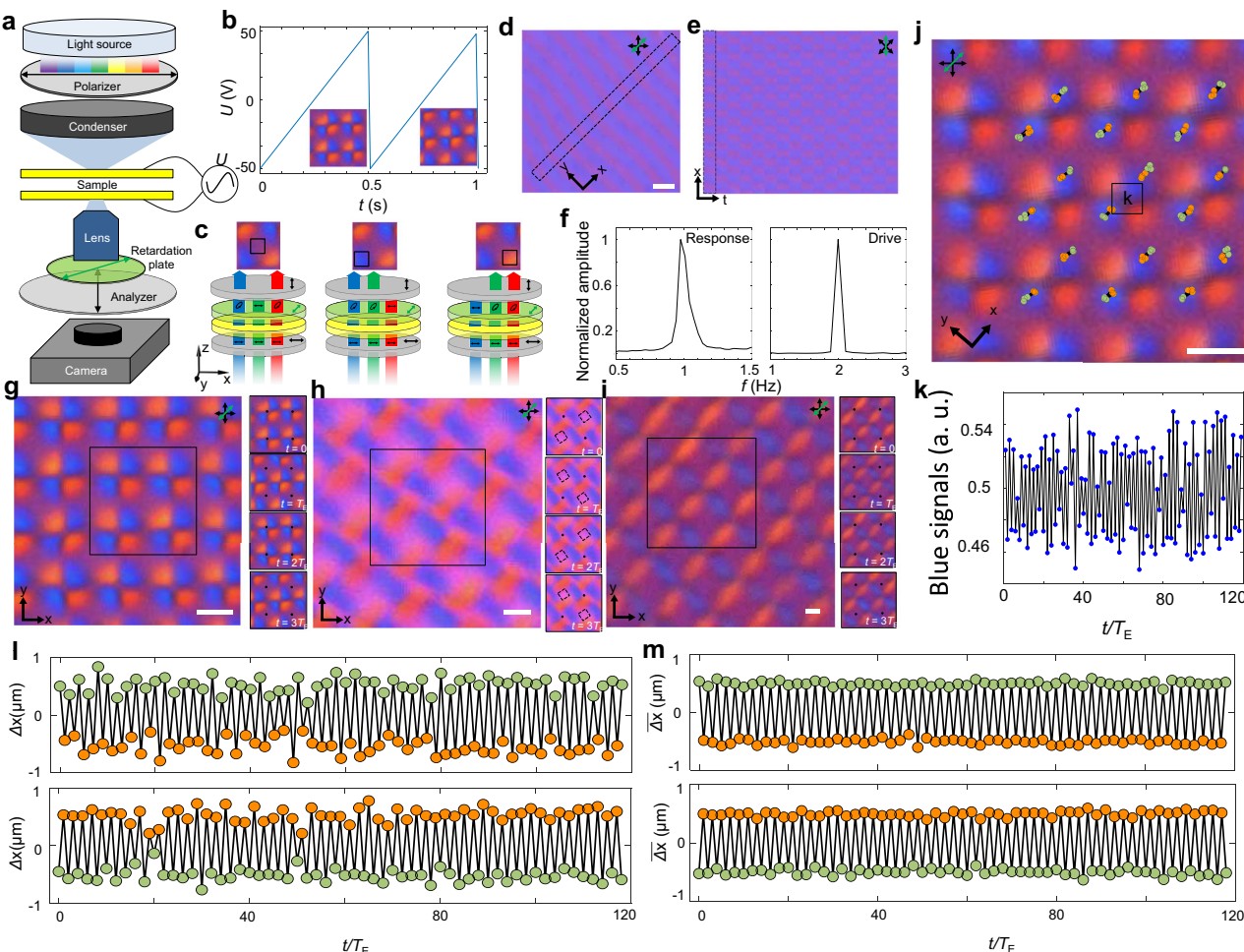

**Fig. 1 | DSTCs in chiral LCs. a** Schematic of an experimental setup. **b** A Floquet sawtooth electrical signal applied to the LC, where the temporal periodicity is $T_E = 0.5$ s, and the amplitude (half of the peak to peak voltage) is $U_{max} = 50$ V; the inset images are captured by the camera within $2T_E$. **c** Schematic shows how complex $\mathbf{n}(\mathbf{r})$ alters the polarization of light within different sample regions, resulting in varying polarized interference colors. The components of the optical setup and sample are colored in the same way as in (**a**). **d** Polarizing optical micrograph (POM) with an inserted retardation plate showing a snapshot of a 1 + 1D DSTC. **e** Space-time image captured for the same time interval $T_E$ within the spatial region marked in (**d**). **f** Fast Fourier transform (FFT) analysis of the movie of 1 + 1D DSTC (left) and the electrical signals of the external drive (right). For the left panel, we record the blue signal of all pixels varying with time. After FFT of each pixel, we sum and normalize the results. **g**–**i** POMs with retardation plate show snapshots of

the 2 + 1D DSTC of configuration 1 (**g**), configuration 2 (**h**) and configuration 3 (**i**), respectively. Right images are snapshots captured within $4T_E$ as marked on the left; the black dots and dashed squares in the images serve as references. **j** Trajectories tracking the center of each blue-colored region marked for the odd (green circle) and even (orange circle) drive period. **k** Average blue color signals versus time within the marked region in (**j**). a.u., arbitrary units. **l, m** Individual (**l**) and average (**m**) displacements of trajectories in (**j**) from odd (top) and even (bottom) spatial lattice lines, respectively. The reference point for each blue-colored region is selected as the midpoint between two neighboring drives. Scale bars indicate 10 μm in (**d**) and 5 μm in (**g**–**j**). In (**a**, **c**–**e**) and (**g**–**j**), the transmitting axes of the polarizer and analyser are marked by black double arrows; the slow axis of the retardation plate is marked by a green double arrow.

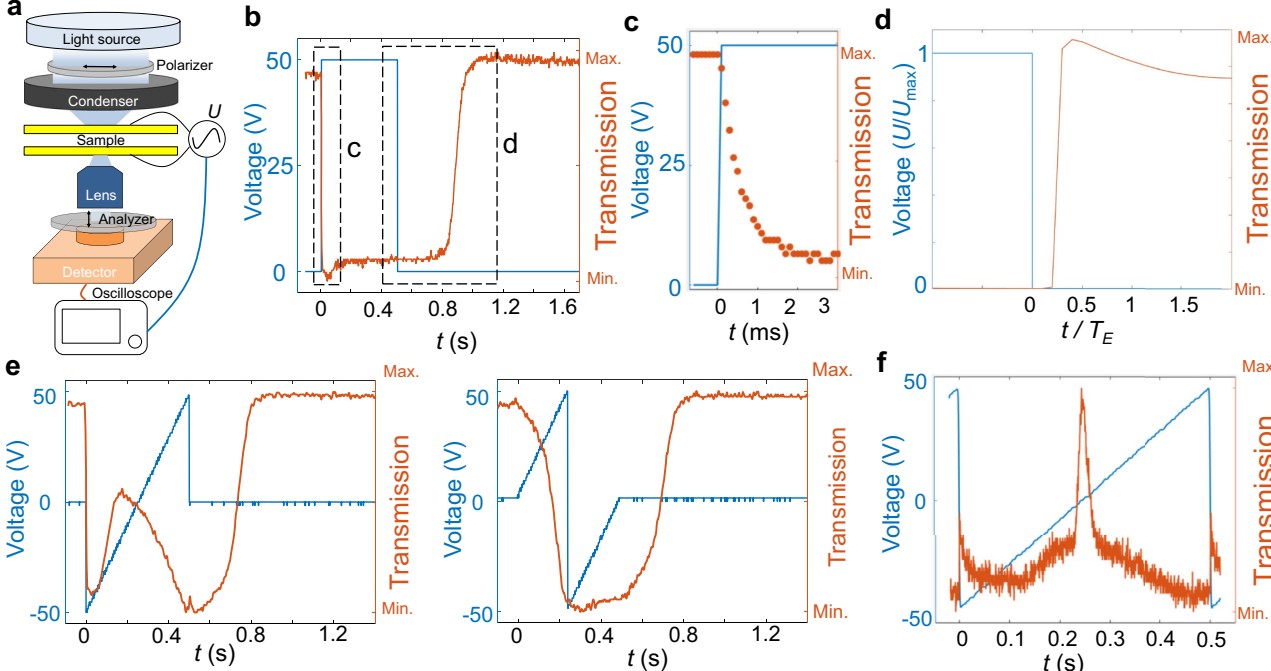

**Fig. 2 | Chiral LC's response to electrical pulses. a** Schematic of the microscope-based experimental setup. Without voltage, light with initially linear polarization passing through the chiral LC becomes elliptically polarized. When the voltage is large, the LC director becomes perpendicular to the polarizers, so the linearly polarized light passes through the sample and is blocked by the analyser. The signal is captured over a large area and detected by a photodiode coupled to an oscilloscope. **b** Electrical signal's magnitude and transmitted light intensity versus time for a single pulse. **c** Time-zoom-in plot marked in (**b**) shows that the response time is about 2 ms when the square pulse is switched on. **d** Numerically simulated electrical signal's magnitude and transmitted light intensity versus time corresponding to the relaxation process marked in (**b**). $T_E$ denotes the temporal periodicity of the external drive for dynamic simulations. **e** Transmission versus time for a sawtooth electrical signal and for one-pulse waves. **f** Transmission versus time for a periodic sawtooth electrical signal within one $T_E$, the pattern emerges when the voltage is close to zero. The response time of the LC system for a sawtooth signal is on the same scale as $T_E$.

where the LC is further doped with ionic substances[29–31]. In response to a Floquet electrical signal (Figs. 1a, b, 2), the confined LC can spontaneously form spatially periodic configurations (Supplementary Movie 1), which are captured by a camera of an optical microscope system (Fig. 1a). The spatially varying optical phase retardation pattern is produced by the LC with complex structure of director orientation driven by the field, which can be vividly revealed by inserting an additional first-order full-wave retardation plate (Fig. 1c and "Methods"). The alternating blue and purple spatial regions indicate spatial variations in the LC's three-dimensional (3D) structures represented by the locally-averaged molecular orientation direction **n** (dubbed the "director").

Because our system consists of a chiral nematic LC with ion doping under surface confinement with perpendicular boundary conditions, which is incompatible with the uniform twist of the chiral nematic phase, it is important to examine its response to different electrical signals. Using the experimental setup shown in Fig. 2a, we measure the light transmission over a relatively large area (~1 mm²) captured by the detector. The system responds within a few milliseconds after turning the voltage on (Fig. 2b, c), whereas the relaxation back to the initial state is a slower process when the voltage is turned off (Fig. 2b, d). As we apply different sawtooth electrical signals, we observe complex responses (Fig. 2e, f), where the relaxation process occurs on the timescales of ~0.1–1 s. These temporal responses are generally consistent with the timescales of the complex space-time configurations (Supplementary Movie 1), driving rich dynamic phenomena discussed below.

The polarized light interference pattern exhibits clear spatial periodicity under Floquet sawtooth electrical signal (Figs. 1d, 2f), which can be reproduced by numerical simulations of the director configurations based on both Frank−Oseen and Landau−de Gennes free

energy models (Figs. 3, 4, Supplementary Fig. S1 and "Methods")[32,33] and the subsequent modeling of polarized light propagation through such a system with the Jones-matrix method[34]. While the spatio-temporal crystallization in light polarization (and its color) patterns is apparent from imaging, it also reveals similar behavior in the spatio-temporal response of the chiral nematic LC medium. By analysing the interference-color signal within the camera-captured micrographs of these regions over time (Fig. 1e, Supplementary Movie 1), we find that the spatial pattern recurs every two periods of the external Floquet drive (Fig. 1f). This behavior is often called the "period-doubling" phenomenon[7–9], indicating that the time translation symmetry is broken discretely. The near-neighbor correlations are antiferromagnetic-like both in time and in space, similar to the case of previous theoretical studies[18–20] for different systems. Furthermore, the discrete time crystallization phase can be observed for different chemical chiral LC substances ("Methods"). We have identified three distinct configurations of the 2 + 1D DSTCs (Fig. 1g–i). Notably, configuration 1 (Fig. 1g) and configuration 2 (Figs. 1h, 3g–k) are observed under the same conditions (to be detailed later), while configuration 3 (Fig. 1i) is typically observed only for higher $d/p$ ratios, where $d$ is the cell thickness, and $p$ is the helicoidal pitch of the chiral nematic LC. Although these three configurations of 2 + 1D classical discrete time crystals can have different spatial lattice periodicities that are comparable to the helicoidal pitch, along the temporal axis, they feature the period-doubled response to the external drive (Fig. 1g–i). To characterize the period-doubling phenomenon over long times, we track the colored regions ("Methods"), which correspond to different localized structures of the director field **n(r)**. Figure 1j shows positions-vs-time trajectories of the center of the blue regions, overlaid on top of the images, revealing that these regions in odd/even lines of the lattice move in opposite directions within each drive period (anti-ferromagnetic-like), which is also

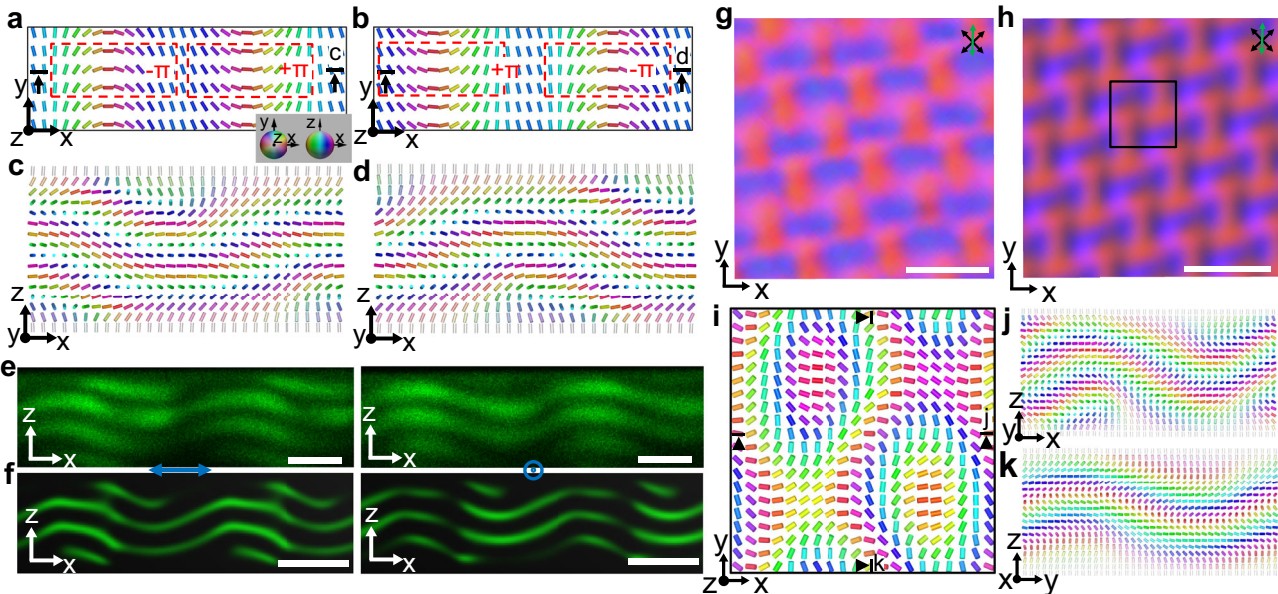

**Fig. 3 | Quasi-static initial director structures. a, b** $\mathbf{n}(\mathbf{r})$ in mid-planes (*x-y* planes) of the initial states, which can be interpreted as containing a pair of -π and +π elementary domain-wall solitons, as marked. **c, d** $\mathbf{n}(\mathbf{r})$ in vertical *x-z* planes marked in (**a, b**), respectively, shown in an inset of (**a**) with cylinders colored according to the order parameter manifold, the sphere with diametrically opposite points identified. **e, f** Experimental (**e**) and numerically simulated (**f**) three-photon excitation fluorescence polarizing microscopy images[33] of two repeat units of a

structure shown in (**c**), where polarizations of excitation light are marked by blue arrows. **g, h** Experimental (**g**) and numerically simulated (**h**) POMs of the 2 + 1D DSTC (configuration 2). **i–k** $\mathbf{n}(\mathbf{r})$ (**i**) in mid-plane corresponding to the marked region in (**h**) and in *x-z* (**j**) and *y-z* (**k**) cross-sectional planes marked in (**i**). Scale bars are 10 μm in (**g, h**) and 5 μm in (**e, f**). Transmitting axes of polarizer and analyser are marked by black double arrows; the slow axis of a retardation plate is marked by a green double arrow.

evident by tracking the average blue signals (Fig. 1k). Furthermore, the individual (Fig. 1l) and collective spatial displacements (Fig. 1m) take place while maintaining the period-doubling behavior.

## Majorana-like quasi-particle nature of building blocks of time crystals

Our computer simulations based on the Ginzburg–Landau equation accounting for dielectric, flexoelectric, and screening-charge effects ("Methods") reproduce the experimental observations of the sample's nonsingular initial state at zero field, as well as the topological quasi-particles and the period-doubling phenomenon in the DSTC states with both nonsingular domain wall solitons and singular line defects (Fig. 4a–f, Supplementary Movie 2)[35]. When the initial state (Fig. 3) transforms into the 1 + 1D DSTC state (Fig. 4), the symmetry breaking becomes different during the time crystal formation over tens of electrical driving periods. This transition takes place while complying with the sample's perpendicular boundary conditions for $\mathbf{n}(\mathbf{r})$ on confining substrates ("Methods"). The period-doubling phenomenon within the 1 + 1D DSTC is directly observed in the snapshots of the director field and the scalar order parameter's spatial distribution (Fig. 4c, f). To understand the underlying mechanism, we thoroughly analyse the dynamic process, finding the fascinating inter-transformations, generations, and annihilations of coexisting topological solitons and singular disclinations (Fig. 4g–o). The singular disclination structures, as well as the fragments of solitonic nonsingular walls between them, are the building blocks of the 1 + 1D time crystal, which can be treated as Majorana-like quasiparticles and respective anti-particles (Fig. 4k)[26–28], because the studied disclination profiles in their cross-sections smoothly transform as spinors following the Majorana equation[27]. The solitonic walls exhibit temporal evolutions that comply with the transformations of the singular line defects at their ends (Fig. 4g–i).

In the time crystal phase, the particle-antiparticle inter-transformation occurs when the external voltage *U* smoothly changes from

negative to positive (Fig. 4g–j). The director field around the translationally invariant disclination line region can be quantified by the characteristic twist angle $\beta$ ($\beta \in [0, \pi]$)[36–38], whereas the core (center) of the disclination line is a singularity in $\mathbf{n}(\mathbf{r})$, within which the director orientations cannot be defined. When the amplitude of instantaneous negative voltage *U* is relatively large, $\beta$ equals π for the top and 0 for the bottom disclinations (Fig. 4g), respectively, which means that $\mathbf{n}(\mathbf{r})$ confined to the plane perpendicular to disclination lines (with the effective order parameter space being $\mathbb{S}^1/\mathbb{Z}_2$, the one sphere - or circle - with diametrically opposite points identified, for the director confined to the cross-sections). These defect lines can be referred to as wedge disclinations[39] with winding numbers −1/2 and +1/2 defined in the two-dimensional cross-sections orthogonal to the defect lines (Fig. 4j), respectively[39]. The 2D winding numbers ±1/2 of disclinations relate to the accumulated angle of $\mathbf{n}(\mathbf{r})$ rotation as one circumnavigates the disclination core once, divided by 360°, with the positive sign corresponding to the counterclockwise rotation, matching that of circumnavigation, and the negative sign referring to the clockwise case. However, once the director is allowed to rotate out of the 2D plane orthogonal to the defect lines, becoming 3D in nature, the initial −1/2 and +1/2 structures become topologically the same. While the −1/2 and +1/2 cross-sectional structures would correspond to topologically distinct defect states in a purely 2D system, they are topologically analogous and deformable to one another in our system because 3D rotations of director are allowed. Indeed, these structures morph one to another over time via smooth director rotations (Fig. 4g–i), so that they can be treated as Majorana-like quasiparticles and respective antiquasiparticles. These defect lines within our LC's 3D space are the topologically nontrivial elements of the first homotopy group $\pi_1(\mathbb{S}^2/\mathbb{Z}_2) = \mathbb{Z}_2$, where $\pi_1$ refers to the measuring circle around the defect line, and the director's order parameter space is $\mathbb{S}^2/\mathbb{Z}_2$, the two-sphere with diametrically opposite points identified.

Between the two disclination regions, the −π-rotation Néel domain wall solitons (with +/− defined by counterclockwise/

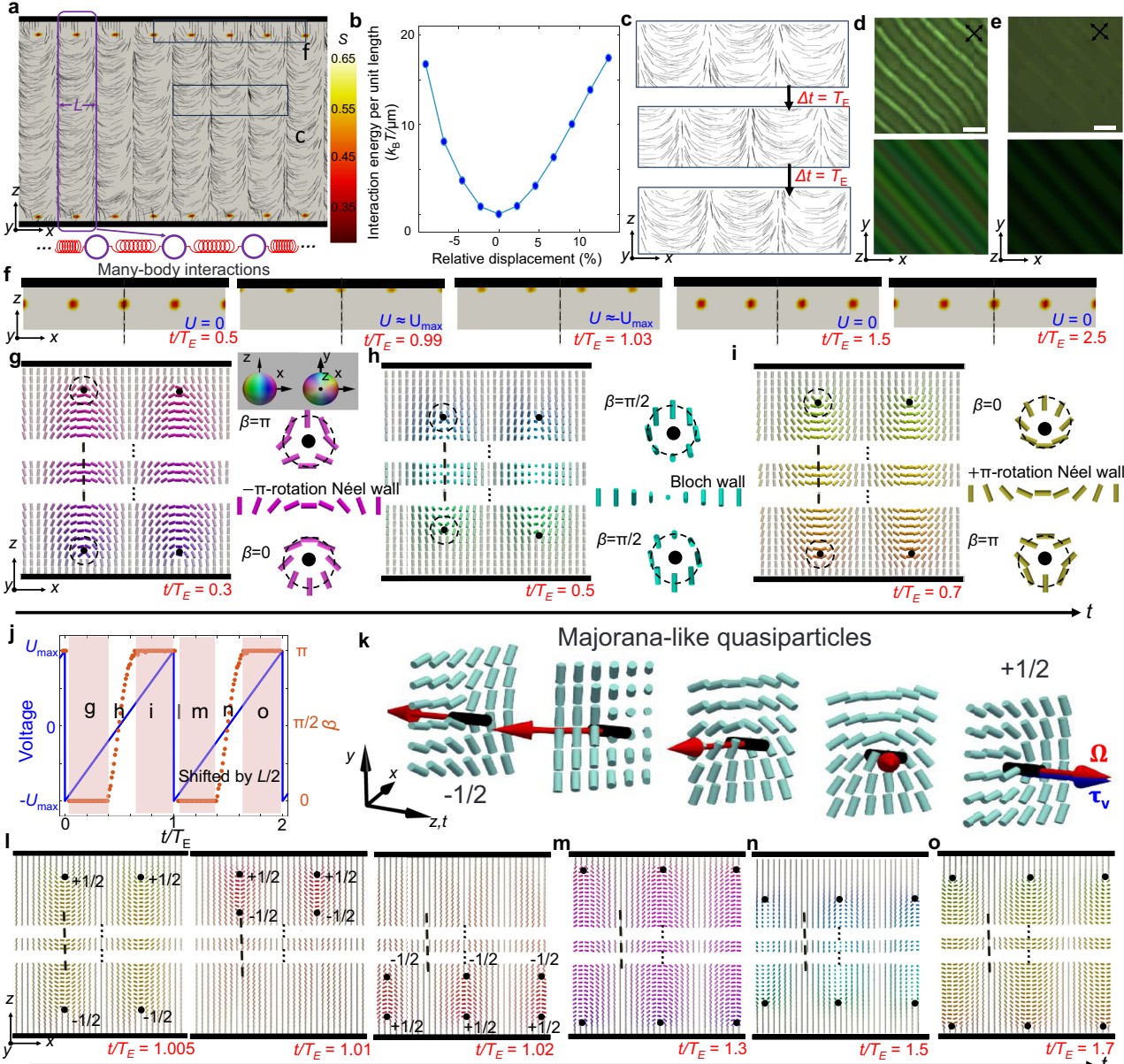

**Fig. 4 | Majorana quasiparticles and period-doubling in DSTCs. a** Numerically simulated director field of 1 + 1D DSTC based on the Landau-de Gennes free energy functional, the background is colored by the scalar order parameter $S$ (right-side inset). The bottom inset schematically shows the many-body interactions among neighboring building blocks, which consist of topological solitons and disclinations. **b** Interaction energy per unit length (translation invariant along $y$) versus displacement relative to the equilibrium length (along $x$). **c** Time-dependent director field showing the period-doubling phenomenon, with the selected region marked in (**a**). **d**, **e** Enhanced contrast experimental (top) and numerically simulated (bottom) POM images under a small (**d**) and large (**e**) electric field within one $T_E$. **f** Snapshots of the $S$ distribution at different times within the selected region marked in (**a**). Black dashed lines in fixed positions are shown for reference. **g–i** Snapshots of the director field (left) showing the topological transformation, angle $\beta$ of the disclination and the topological solitons in between vary smoothly when the voltage is close to zero. The corresponding schematics revealing their topological nature are shown on the right. Black circles show disclination cores; black dashed rings characterize the director on closed loops. **j** External drive and twist angle $\beta$ of the disclination near the bottom surface within two external drive periods, $T_E$. In the second $T_E$, the structures are shifted by half of the spatial period. **k** Director field profiles showing the transition from −1/2 to +1/2 configurations of disclinations, which can be viewed as a Majorana-like quasiparticle and its antiparticle. The twist angle $\beta = \cos^{-1}(\boldsymbol{\tau_v} \cdot \boldsymbol{\Omega})$, where $\boldsymbol{\tau_v}$ (blue arrow) is the tangent vector of the disclination (black cylinder), and $\boldsymbol{\Omega}$ is the rotation vector (red arrow). **l** Snapshots of the director field showing the transition after the voltage suddenly changes from $+U_{max}$ to $-U_{max}$ (left), followed by annihilation (middle) and generation (right) processes. **m–o** Snapshots of the director field during the second $T_E$ (**j**), when topological solitons and disclinations shift by $L/2$ compared to (**g–i**), respectively. Black dashed lines in (**g–i**, **m–o**) show relative positions for reference.

clockwise rotation when traversing the topological soliton from left to right) are spatially embedded (Fig. 4g). The domain wall solitons are also labeled as elements of the first homotopy group[40], $\pi_1(\mathbb{S}^2/\mathbb{Z}_2) = \mathbb{Z}_2$, and are also allowed to smoothly inter-transform to have 2D cross-sections of different types (Néel or Bloch type) while here being

terminated by the two disclination lines. Thus, both singular and solitonic $\pi_1(\mathbb{S}^2/\mathbb{Z}_2) = \mathbb{Z}_2$ structures morph between geometrically different but topologically the same states under electric driving—akin to deforming a donut into a coffee mug without changing the topological properties (Fig. 4g–i). As the voltage $U$ increases from negative values

to zero (Fig. 4h), the angle $\beta$ characterizing both wall-terminating disclinations smoothly changes to $\pi/2$ (corresponding to pure twist winding), and the initial $-\pi$-rotation Néel domain wall solitons (with bend-splay structures) transform to Bloch domain wall solitons (with pure twist structures). As the voltage $U$ further increases to $U > 0$ (Fig. 4i), $\beta$ changes to 0 for the top and $\pi$ for the bottom defect lines, and the Bloch domain wall solitons transform to the $+\pi$-rotation Néel domain wall solitons. The transition is continuous during the linear voltage change, and the positions of the disclinations and solitons do not shift spatially, just morph geometrically.

The array of topological quasiparticles formed by solitonic walls terminated on singular defect lines maintains its topological nature (Fig. 4i) until $U$ suddenly changes from the positive maximum value to the negative maximum value (Fig. 4l), at which point topological particle-antiparticle pairs annihilate (Supplementary Fig. S2a) and re-generate (Supplementary Fig. S2b), while the locations of the emerged quasiparticles are found synchronously shifted by half a spatial period ($L/2$) away from the previous ones. The origin of quasiparticles' annihilation and regeneration refers to the discontinuity of external drive: the voltage changes dramatically, but the director field cannot deform correspondingly fast. After the generation, compared to the structures of the director field one temporal period ($T_E$) before (Fig. 4g–i, m–o), the disclinations and topological solitons are the same, with an $L/2$ shift, so that the system adopts the exact same configuration every $2T_E$, resulting in the period-doubling phenomenon. In the experiments and simulations alike (Fig. 4d, e), even when the optical images appear almost homogeneously dark under a high electric field between crossed polarizers, the defects and director deformations do not fully disappear, as evident from the enhanced contrast POM images that reveal effective "memory" of the periodic deformations that allows for correlating positions of these topological objects within $2T_E$.

Serving as building blocks of time crystals, the self-free energy of a quasiparticle unit (Fig. 4a, including a pair of Majorana-like singular defect quasiparticles and domain wall solitons in-between them) relative to the ground state in a passive LC is $\sim 1.3 \times 10^3 \, k_B T/\mu m$ ($k_B$ is the Boltzmann constant and $T$ is the temperature), as the system is powered by the external drive to overcome the energy gap, consistent with the anticipated existence of Majorana-like quasiparticles in active and driven nematic LC systems out of equilibrium[26,27]. Moreover, the elasticity-mediated energy of the interaction between neighboring units is $\sim 10 \, k_B T$ with a 10% compression or stretching calculated from Landau–de Gennes free energy (Fig. 4b), further indicating that these topological quasiparticles behave as particle-like objects, which can collectively form a crystal. In addition to reproducing the topological time-crystalline character, by changing the voltage amplitude in the computer simulations, the Fourier analysis of the time-dependent $\mathbf{n}(\mathbf{r},t)$ reveals the transitions of period-doubling time-crystalline states to other dynamic states (Supplementary Figs. S3, S4), which are consistent with experimental results, as elaborated in the next section below.

## Stability and robustness

The stability range of the emergent phase of discrete time crystallization depends on the cell thickness, the LC's helicoidal pitch, temperature, voltage amplitude $U_{max}$, and external drive period $T_E$. To construct the phase diagram, we employ samples of different cell thickness but the same LC pitch while controlling temperature, $U_{max}$, and $T_E$ (Fig. 5a–d). We find five distinct phases (Fig. 5e–i) within explored ranges of these parameters: a time-symmetry-unbroken phase (Fig. 5e, Supplementary Movie 3) with the temporal periodicity of the LC system being the same as the external drive, a disordered phase (Fig. 5f, Supplementary Movie 3) with a disordered response to the external drive, and a phase co-existence region (Fig. 5i, Supplementary Movie 3), where both the 1 + 1D DSTC (Fig. 5g) and 2 + 1D DSTC (Fig. 5h) phases can co-exist. For a thin cell, we observe both the

1 + 1D and 2 + 1D DSTC phases (Supplementary Movie 4) depending on $T_E$ (ranging from 0.35 s to 1 s) and temperature (ranging from 24 °C to 31 °C). This indicates that the DSTC is quite robust within a broad range of parameters, as the maximum drive period can be ~ 3 times longer than the minimum one, whereas the intrinsic elastic and dielectric properties of the LC can undergo a ~40% change with such temperature variations, correlating with changes of the LC's scalar order parameter[32,41]. The temporal range of $T_E$ in DSTC phases is on the same timescale as the response time of chiral LCs to a sawtooth electrical signal (Fig. 2e, f), which further demonstrates the importance of the coupling between the electrical drive and the LC director field that is structurally complex even in its initial state at no applied voltages (Fig. 3). Interestingly, we find that the phase diagrams in the coordinates of temperature and $T_E$ exhibit an unexpected resemblance: increasing the temperature yields behavior similar to that of decreasing $T_E$. This is because when $T_E$ is small, the director field fails to maintain its topological structure as the external electric field changes too rapidly; similarly, as the temperature increases, the elastic and dielectric constants decrease, decreasing the interaction energy between the quasiparticles, so the system cannot maintain its topological structure either. For a thick cell, the DSTC phase also exists over a broad range (Supplementary Movie 4).

We examine the rigidity (robustness) of the DSTC starting from its formation, where the DSTC spontaneously "boils out" from the disordered state (Fig. 6a, b, Supplementary Movie 5), resembling previous theoretical findings[18], which indicates the spontaneous symmetry breaking both in space and time. Within the DSTC phase, the DSTC region grows as the disordered region shrinks (Fig. 6c) because the disordered state is "unstable" relative to the ordered configuration. We further examine the rigidity by adding random temporal perturbations $\Delta T_E$ to the external drive period at each period, finding that both the 1 + 1D and 2 + 1D DSTC phases are robust under such temporal perturbations (Fig. 6d, e, Supplementary Movie 6). The spontaneous symmetry breaking both in space and time and robustness against temporal perturbations are important properties of space-time crystals identified in recent literature[23], serving as verification criteria of space-time crystals that our system appears to satisfy.

## Lattice defects and long-range order

Just like conventional space crystals often have defects[39], various crystal lattice imperfections (like dislocations, vacancies and self-interstitial points) can emerge in DSTCs (Fig. 7, Supplementary Movie 7). However, because of the rigidity of the time crystals, the discrete space-time crystallization phase tends to recover from such lattice imperfections after tens of drives (Fig. 7a). Additionally, lattice defects can be introduced by manipulating the LC with a focused infrared beam of laser tweezers ("Methods"). The laser beam (Supplementary Movie 8) forms defects in space coordinates of the DSTC, which also tend to disappear with time (Fig. 7b). These results motivate us to experimentally measure how long our classical DSTC (Fig. 8, Supplementary Movie 9) can show the period-doubling phenomenon, revealing that the 2 + 1D DSTCs maintain period-doubling locally (within the camera captured region) for hours ( ~ $10^4$ drives) and 1 + 1D DSTCs for about 24 h ( ~ $10^5$ drives). To verify the local temporal correlation, we define the temporal correlation function $G(t) = \langle \Phi(t)\Phi(0)\rangle - \langle \Phi(t)\rangle\langle \Phi(0)\rangle$, where the brackets indicate an average, and $\Phi$ is the signal captured by the camera ("Methods"). It's well-known that for a smectic phase, the spatial correlation function $G(r)$ decays in a power-law manner depending on the distance $r$, where the power-law index should be small (as shown in Supplementary Fig. S5). Interestingly, an exponential decay (Fig. 9, Supplementary Fig. S6) on $t$ could fit the temporal correlation function of 1 + 1D DSTC, with both the spatial and temporal fluctuations tending to eventually destroy the corresponding correlation in the long ranges[9,18,42–44] (Supplementary Figs. S7–9). Achieving

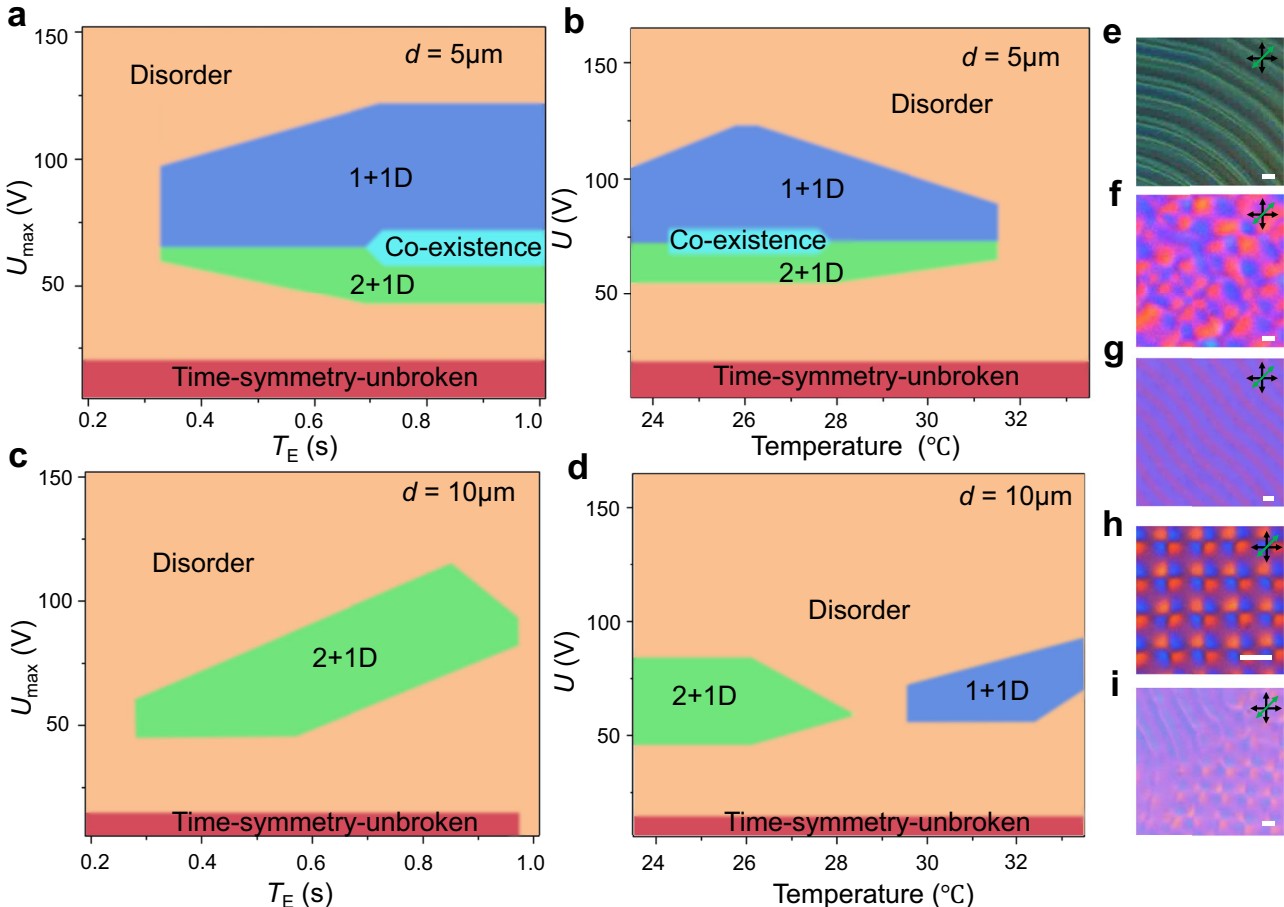

**Fig. 5 | Phase diagrams containing DSTCs. a** Phase diagram in coordinates of $T_E$ and the voltage amplitude $U_{max}$ for a sample of cell thickness $d = 5\,\mu m$ at room temperature. **b** Phase diagram as a function of the temperature and $U_{max}$ for the same sample as in (**a**) for $T_E = 0.5\,s$. **c** Phase diagram as a function of $T_E$ and $U_{max}$ for $d = 10\,\mu m$ at room temperature. **d** Phase diagram as a function of temperature and $U_{max}$ for the same sample as in (**c**) for $T_E = 0.5\,s$. **e–i** POM snapshots of the time-symmetry-unbroken phase (**e**), disordered phase (**f**), 1 + 1D DSTC phase (**g**), 2 + 1D DSTC phase (**h**) and co-existence phase (**i**), respectively. The helicoidal pitch is $p = 5\,\mu m$. Scale bars indicate 5 μm. Transmitting axes of polarizer and analyser are marked by black double arrows; the slow axis of a retardation plate is marked by a green double arrow.

extended correlations, therefore, requires enhancing system controls that preclude issues like thermally- or vibrations-induced drift. Numerically, by introducing a white-noise stochastic perturbation $f_{noise}(t)$, with an autocorrelation $<f_{noise}(t)f_{noise}(t')> = 2\eta_O T_{eff}\delta(t-t')$ (where $\eta_O$ is the friction coefficient and $T_{eff}$ is the effective temperature of the thermal perturbation), we observe that the order of the system decreases dramatically as the effective temperature increases (Supplementary Fig. S10). More defects emerge in the bulk, and the lifetime (correlation time) decreases exponentially with increasing $T_{eff}$, suggesting a (thermally) activated time crystal[9,18,43]. Under conditions around the phase transition (Fig. 5), the lifetime of the DSTC can be very short due to frequent occurrences of line dislocations (Fig. 7c), causing the emergence of the disordered states within only ten-to-hundred drives.

### Potential candidates of fractional discrete time crystals

The emergent spatiotemporal response of chiral nematic LCs under the Floquet drive can give rise to plentiful other phenomena, such as the quasi-hexagonal lattices seen in the spatial coordinates (Fig. 10a, Supplementary Movie 10) when $d/p > 3$, which potentially can be a fractional discrete time crystal[19,45–47] (Fig. 10). The intrinsic time periodicity is not an integer multiple of the external drive (Fig. 10c,

Supplementary Fig. S11), and the FFT analysis (Fig. 10d) reveals a peak around $10T_E/3$ $(f \approx 0.3f_E)$. The strong noise around the peak in FFT analysis originates from the emergence of lattice imperfections (such as 5–7 defects, Fig. 10b) in spatial coordinates. We observe 15 lattice imperfections within a region of ~10 hexagonal units over 200 external drives. In addition, the temporal correlation function (Fig. 10e) further suggests the internal temporal period is approximately $33T_E/10$ or $10T_E/3$ $(40T_E/12)$. Interestingly, fractional discrete time crystals of similar fractional numbers (~3.3 and 100/29) have been observed in two different quantum systems[46,47]. Whether they share common universal mechanisms remains an open question.

## Discussion

Observations of classical discrete space-time crystals reveal the generality of time crystallization dynamics. In our classical LC-based discrete space-time crystals, the time symmetry is discretely broken while being accompanied by space symmetry breaking, yielding 1 + 1D and 2 + 1D space-time crystals. These DSTCs can be described as comprising arrays of spatially and temporally localized quasiparticles interacting with each other within the overall out-of-equilibrium setting (Fig. 4a, b). The future studies should extend the detailed analyses of the reported period-doubling effect in 1 + 1D space-time crystals to

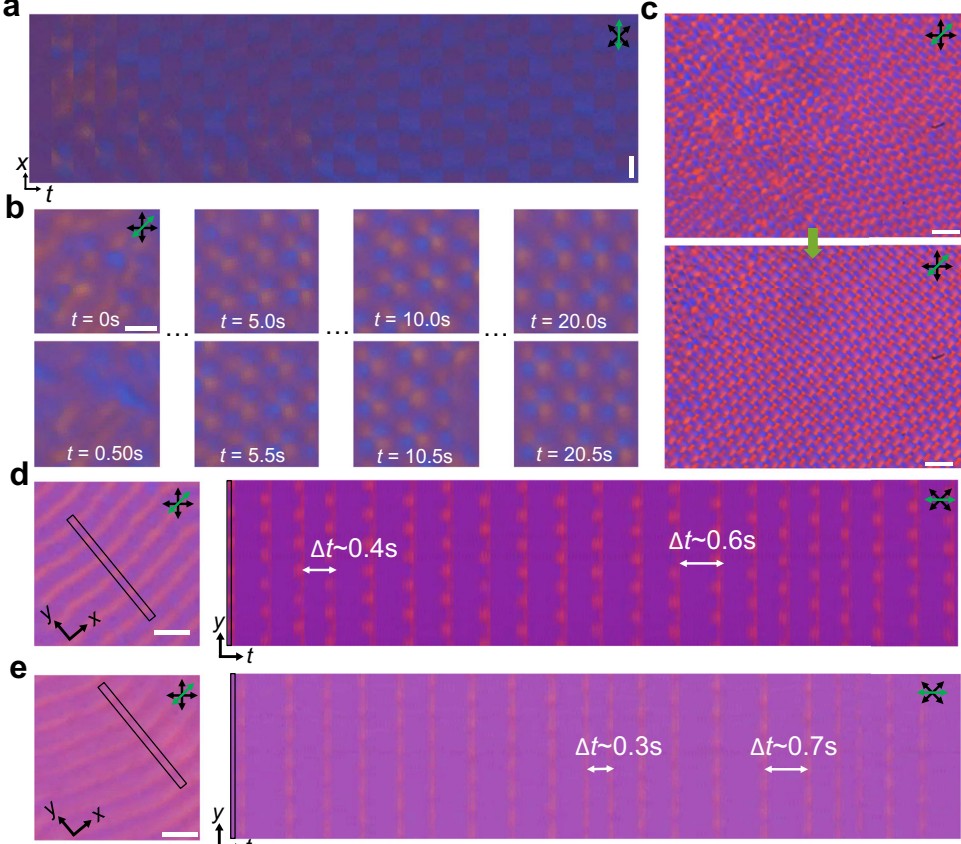

**Fig. 6 | Formation of DSTCs and their rigidity against temporal perturbations.** **a** Space-time image captured for the same time interval $T_E$ within a spatial stripe-like region showing the 1 + 1D DSTC "boil out"[18] from a disordered state. **b** POM snapshots showing the 2 + 1D DSTC "boil out" from a disordered state. The elapsed time is marked on the panels. **c** POM snapshots showing the nucleation and growth of the DSTC region. The time interval of the two snapshots is 20 s (40 drives).

**d**, **e** POM snapshot (left) and space-time plot (right) of a 1 + 1D DSTC (**d**) and a disordered state (**e**) with temporal perturbation $\triangle T_E$ randomly distributed within $[-0.2\bar{T}_E, 0.2\bar{T}_E]$ (**d**) and $[-0.4\bar{T}_E, 0.4\bar{T}_E]$ (**e**), where the average external temporal periodicity $\bar{T}_E = 0.5$ s. The tracked region of the space-time plot is marked on the left, where the $0.1\bar{T}_E$ time interval of each snapshot step. Scale bars are 10 μm in (**a**), 5 μm in (**b**) and 20 μm in (**c**–**e**).

high-dimensional cases, including 2 + 1D, which we already observed experimentally. These classical DSTCs may offer a new route to creating various forms of meta matter[48–50], where the basic building blocks are localized not only in space, but also in time, as well as have the emergent topological nature. They may allow designing spatially or temporally localized structures as versatile reconfigurable beam deflectors, steerers, and lasing elements[48,51,52]. The examined rigidity of our classical DSTC allows for maintaining order locally over times much longer than the discrete time crystals in quantum systems, which is because, although the classical system cannot enjoy the benefits of many-body localization, there is no quantum coherence, and the relative noise from thermal fluctuations is much smaller for soft matter systems when considering the system's internal elasticity-mediated interactions. Our findings are consistent with the recent theoretical proposals[18–20] that the spontaneous symmetry breaking both in time and in space can be a widespread occurrence in numerous open systems, not only in a quantum but also in the classical context.

Our findings may inspire further interest in the spatiotemporal properties of well-studied subharmonic and temporally periodic systems[53], potentially driving additional experimental analyses and providing interpretation from the perspective of time-crystalline order. Furthermore, our study also naturally opens a question of time liquid crystallinity, where features of orientational order co-existing with no or only partial positional order can also emerge in the temporal domain or simultaneously in temporal and spatial domains. Particularly interesting time-liquid-crystallinity effects can be anticipated to emerge in various active matter systems, where external drive could be potentially substituted by the periodic supply of energy, e.g., via light illumination for filamentous cyanobacterial systems[54].

## Methods

### Materials and sample preparation

Chiral nematic LCs are prepared by mixing nematic LCs 4-Cyano-4′-pentylbiphenyl (5CB, EM Chemicals) or E7 (Shijiazhuang Chengzhi Yonghua Display Material Co.) with a small amount of a left-handed chiral additive, cholesterol pelargonate (Sigma-Aldrich). The helicoidal pitch, $p$, of the mixtures is fixed to be 5 μm and controlled by the concentration, $C_{dopant}$, of the chiral additive with known helical twisting power $h_{htp}$, according to the relation $p = 1/(h_{htp} \cdot C_{dopant})$, where the helical twisting power $h_{htp} = 6.25 \mu m^{-1}$ for the cholesterol pelargonate. The chiral nematic LCs are further doped by ~0.1 wt% of a cationic surfactant hexadecyltrimethylammonium bromide (CTAB, Sigma-Aldrich) in order to boost LC's conductance[29–31], the maximum screening ability can be ~$10^2$V; we present results for the chiral nematic LCs based on 5CB unless specified otherwise.

LC cells are assembled from indium-tin-oxide (ITO)-coated glass slides or coverslips treated with polyimide SE5661 (Nissan Chemicals) to obtain strong perpendicular (homeotropic) boundary conditions on their inner surfaces without any pre-patterning. The polyimide is applied to the substrates by spin-coating at 2,700 rpm for 30 s, followed by baking (5 min at 90 °C and then 1 h at 180 °C). Then, the two ITO-coated glass slides or coverslips are glued together with optical

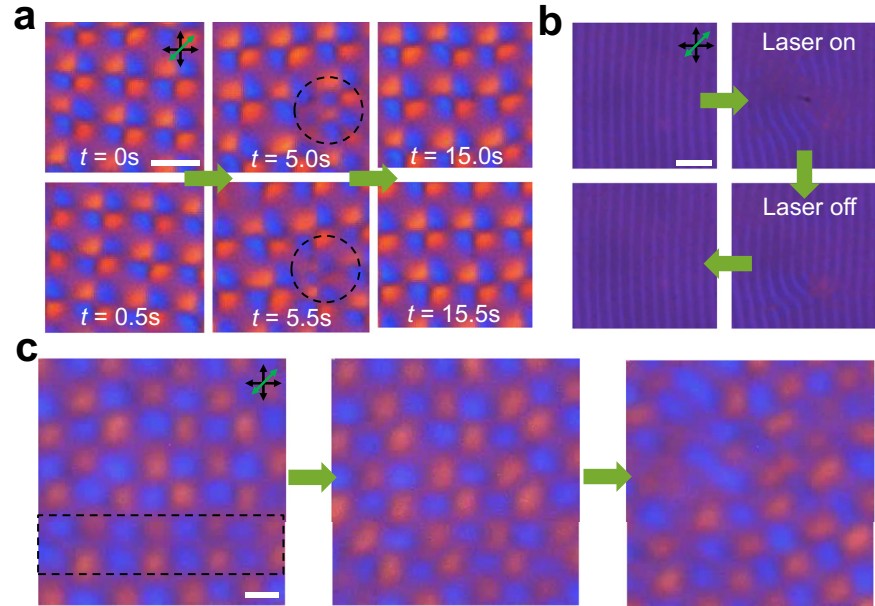

**Fig. 7 | Rigidity of DSTCs against space-time imperfections. a** The emergence and disappearance of a defect region marked by a black dashed circle. Elapsed times are marked on the panels. **b** Recovery of 1 + 1D DSTC where a defect region is generated by a laser tweezer. **c** A quasi-dislocation with a missing row of "quasi-particles" within a strip region close to the transition phase. The POM snapshots are captured at elapsed times of $t = 0$ (left), $t = 10T_E$ (middle) and $t = 30T_E$ (right), respectively. The transmitting axes of the polarizer and analyser are marked by black double arrows, and the slow axes of the retardation plate are marked by green double arrows. Scale bars indicate 5 μm in (**a**, **c**) and 20 μm in (**b**).

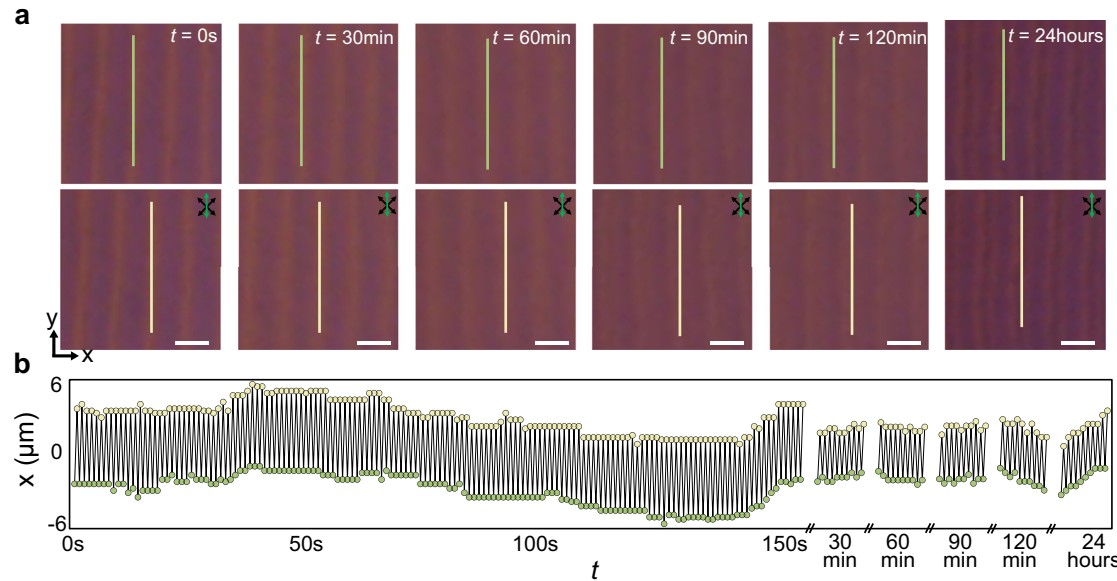

**Fig. 8 | Period-doubling phenomenon of the 1 + 1D DSTC probed at long times. a** POM snapshot of neighboring odd (top) and even (bottom) drives at different times. Scale bars indicate 10 μm. **b** Positions of the topological quasiparticles in the middle of the selected region versus time for the same time interval of 0.5 s, with positions marked atop frames in (**a**).

adhesive (NOA 63, Norland Products), and the LC cell gap thickness is defined by silica spheres as spacers between the two substrates to be 5–15 μm. Metal wires are attached to ITO and connected to a data acquisition board (NIDAQ-6363, National Instruments) with a signal amplifier (Model 7600, Krohn-Hite) for electrical control. Additionally, we use custom-created Matlab codes controlling the data acquisition board, connected to a computer for modulation of the voltage output. The amplitude of the sawtooth wave is $U_{max}$ ($U$ ranges from - $U_{max}$ to +

$U_{max}$) and the temporal periodicity $T_E = 0.5$ s (unless specified differently).

### Optical microscopy and laser tweezers
POM is a method that utilizes polarized light to image the birefringent materials with two crossed polarizers. The sample is illuminated by a wide spectrum of visible light from a lamp, and the light first passes through a polarizer, becoming linearly polarized. Because of the

spatially varying optical phase retardation patterns produced by the LC with a complex structure of director driven by the field, the linearly polarized illumination light transforms into patterns of generally elliptically polarized light with different polarization ellipse's major axis orientations. These polarized-light spatiotemporal periodic patterns can be vividly revealed by inserting an additional first-order full-wave retardation plate (Fig. 1c), where addition and subtraction of the phase retardations due to the LC and the accessory plate convert the spatial variations of light's polarization ellipse orientations into that of first- and second-order polarized interference colors. By measuring the polarized light interference patterns, we find that the spatial spacings of configuration 1 and configuration 2 2 + 1D DSTCs are 8.2 μm and 11.6 μm, respectively, in a sample with a cell thickness of 10 μm. For configuration 3, the spatial spacing is 18.4 μm in a sample with a cell thickness of 15 μm.

POM images and movies are obtained with a multi-modal imaging setup built around an IX-81 Olympus inverted microscope, which is also integrated with an ytterbium-doped fiber laser (YLR-10-1064, IPG

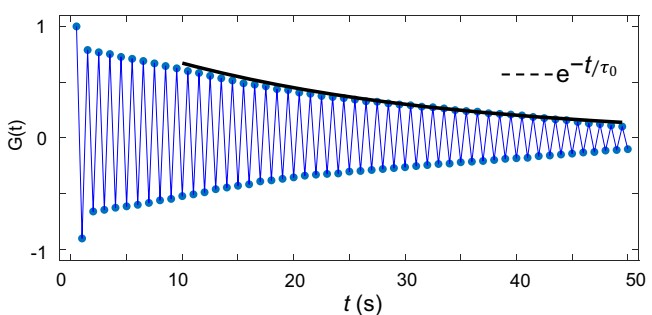

**Fig. 9 | Temporal correlation function of the 1 + 1D DSTC.** Correlation function G(t) versus time for the same time interval $T_E = 0.5$ s. The fitting curve (black solid line) might be indicative of an exponential decay $e^{-t/\tau_0}$ with $\tau_0 = 100$ s, though additional studies will be needed in the future to further confirm this.

Photonics, operating at 1064 nm). All presented POM snapshots and movies are captured with charge-coupled-device cameras (Grasshopper, Point Grey Research). Olympus objectives 100×, 40×, 20× and 10× with numerical apertures of 1.4,b0.75, 0.4, and 0.4, respectively, are used.

Non-contact manipulation by laser tweezers is achieved using a tightly focused 1064 nm laser beam at powers of less than 20 mW. For this, we utilize the Ytterbium-doped fiber laser and a phase-only spatial light modulator (P512–1064, Boulder Nonlinear Systems) integrated into a holographic laser tweezers setup[40,55,56]. The laser tweezers are integrated with the three-dimensional nonlinear optical imaging setup described below, enabling the simultaneous optical control and non-destructive imaging of the LC structures.

## Three-dimensional nonlinear optical imaging of quasi-static director configurations

Three-dimensional nonlinear optical imaging of the LC structures is key to understanding many physical phenomena in LCs[33]. For time-crystalline structures, ideally, the temporal evolution of director field configurations should be probed as well. While doing this is challenging for our time crystals that have the 3D field configuration changing completely within a fraction of a second, our 3D imaging can still provide valuable insights based on imaging quasi-static configurations from which or into which the time-crystalline structures evolve. Our 3D imaging is performed using the three-photon excitation fluorescence polarizing microscopy setup built around the IX-81 Olympus inverted optical microscope[33]. We use a Ti-Sapphire oscillator (Chameleon Ultra II; Coherent) operating at 870 nm with 140-fs pulses at an 80 MHz repetition rate, as the source of the linearly polarized laser excitation light. An oil-immersion 100× objective (NA = 1.4) is used to collect the fluorescence signals, which are detected by a photomultiplier tube (H5784-20, Hamamatsu) after a 417/60-nm bandpass filter. The LC molecules are excited via the three-photon absorption process, and the signal intensity scales $\propto \cos^6\beta_0$, where $\beta_0$ is the angle between the linear polarization direction of the excitation light and the LC director.

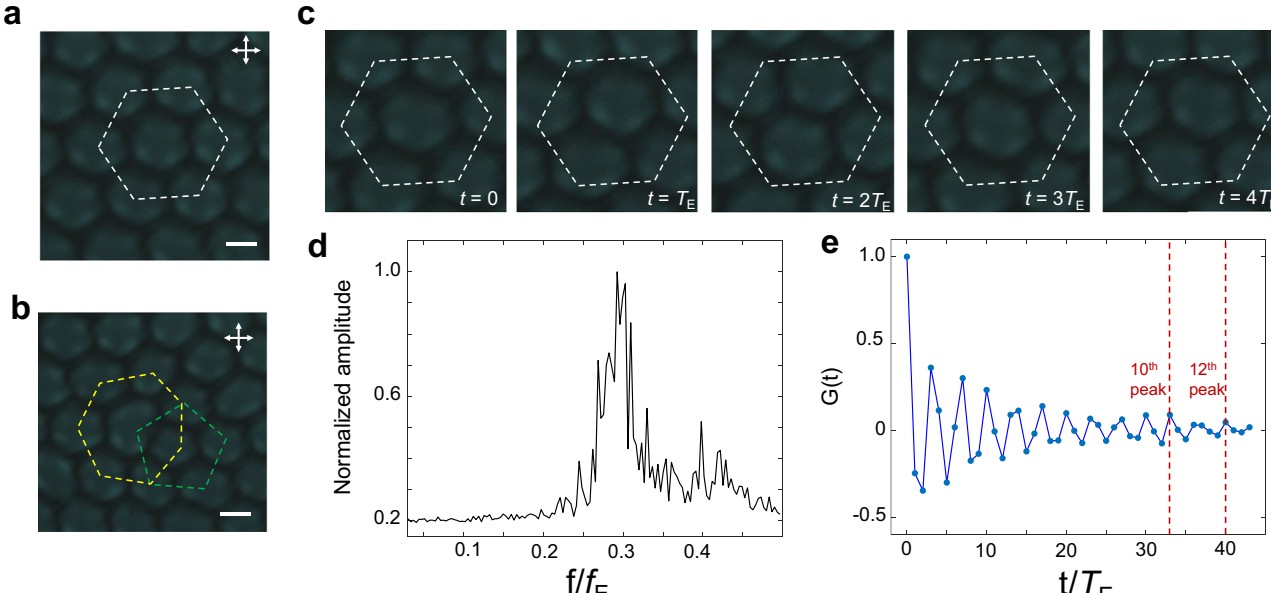

**Fig. 10 | A potential candidate of fractional discrete space-time crystals.**
**a**, **b** POM snapshots show an ordered quasi-hexagonal lattice (a, marked with a white dashed hexagon) and a hexagonal lattice with a 5–7 defect pair (**b**) (marked with a green pentagon and a yellow heptagon, respectively). **c** POM snapshots captured within $5T_E$, and the white dashed hexagons in the images serve as references. Although the images after a $3T_E$ look similar, they indeed differ slightly. **d** FFT analysis of the movie of the fractional DSTC, where the peak is located around $0.3 f/f_E$. **e** Correlation function G(t) versus time for the same time interval $T_E = 0.5$ s, the 10th peak is located at $t = 33T_E$, and the 12th peak is located at $t = 40T_E$. Transmitting axes of the polarizer and analyser are marked by white double arrows. Scale bars indicate 10 μm.

Polarization states of the excitation (as shown in Fig. 3e, f) are controlled by a half-wave plate. When $\mathbf{n}(\mathbf{r})$ is nearly parallel to the linear polarization of the laser beam, the large $\cos \beta_0$ corresponds to the strong three-photon excitation fluorescence polarizing microscopy signal intensity. Computer simulations of the three-photon excitation fluorescence polarizing microscopy images are also based on this dependence of the signal intensity.

## Numerical modeling of quasi-static structures based on the Frank−Oseen free energy functional

While the full modeling of observed time-crystalline structures is challenging, helpful insights can be obtained by 3D imaging and energy-minimization-based modeling of quasi-static structures that the time crystalline patterns evolve from or lead to. For chiral nematic LCs, the energy cost of spatial deformations of the director field $\mathbf{n}(\mathbf{r})$ can be expressed by the Frank−Oseen free energy functional[57]:

$$F_{\text{elastic}}^{\text{FO}} = \int \mathrm{d}^3\mathbf{r} \left\{ \frac{K_{11}}{2}(\nabla \cdot \mathbf{n})^2 + \frac{K_{22}}{2}\left[\mathbf{n}\cdot(\nabla\times\mathbf{n}) + \frac{2\pi}{p}\right]^2 + \frac{K_{33}}{2}[\mathbf{n}\times(\nabla\times\mathbf{n})]^2 \right\},$$

(1)

where the Frank elastic constants $K_{11}$, $K_{22}$ and $K_{33}$ determine the energy costs of splay, twist and bend deformations, respectively. The surface energy is

$$F_{\text{surface}} = -\int \mathrm{d}^2\mathbf{r} \frac{W}{2}(\mathbf{n_0}\cdot\mathbf{n})^2$$

(2)

where $W$ is the surface anchoring strength and $\mathbf{n}_0$ is the director's easy axis orientation at the surface, which is perpendicular to the substrate. When an external electric field $\mathbf{E}$ is applied, the dielectric response of the LC yields an additional dielectric coupling term, so that the free energy is supplemented by the following term:

$$F_{\text{electric}} = -\frac{\varepsilon_0 \Delta\varepsilon}{2}\int \mathrm{d}^3\mathbf{r}(\mathbf{E}\cdot\mathbf{n})^2$$

(3)

where $\varepsilon_0$ is the vacuum permittivity, and $\Delta\varepsilon$ is the dielectric anisotropy of the LC. The total free energy $F = F_{\text{elastic}}^{\text{FO}} + F_{\text{surface}} + F_{\text{electric}}$.

To computer-simulate the spontaneously formed spatial configuration of DSTCs, we assume that these patterns can emerge under a locally relatively weak electric field since the ions can screen the electric field applied to substrates. Inspired by the corresponding 3D imaging insights (Fig. 3a−d), we set the translationally invariant undulated configuration as an initial configuration, where the $1+1$D DSTC-related quasi-static spatial configurations emerge as local or global minima of $F$, and a relaxation routine based on the variational method is used to identify an energy-minimizing $\mathbf{n}(\mathbf{r})$ configuration. Applying this method, at each iteration of the numerical simulation $\mathbf{n}(\mathbf{r})$ is updated using a formula derived from the Euler-Lagrange equation, $\mathbf{n}_i^{\text{new}} = \mathbf{n}_i^{\text{old}} - \frac{\text{MSTS}}{2}[F]_{\mathbf{n}_i}$, where subscript $i$ denotes spatial coordinates, $[F]_{\mathbf{n}_i}$ denotes the functional derivative of $F$ with respect to $\mathbf{n}_i$, and MSTS is the maximum stable time step of the minimization routine, determined by the elastic constants and the spacing of the computational grid. The end-of-the-relaxation condition is identified by monitoring the change in the spatially averaged functional derivatives in consecutive iterations. When the free energy change approaches zero, it signifies proximity of the system to a steady state, and the relaxation routine comes to a halt, yielding the equilibrium or metastable structure for given conditions.

When we apply a time-dependent voltage to our samples, a viscous torque associated with rotational viscosity $\gamma$ opposes the fast rotation of $\mathbf{n}$ within the LC in response to the competing electric and elastic torques. The system tends to evolve towards the energy-minimizing configuration, even though it may never approach one due

to the changing voltage amplitude. The resulting director dynamics is then governed by a torque balance equation, $[F]_{\mathbf{n}_i} = -\gamma\frac{\partial\mathbf{n}_i}{\partial t}$, from which we can obtain the temporal evolution $\mathbf{n}_i(t)$ towards the equilibrium, and the time interval equals $\frac{\text{MSTS}}{2}\gamma$ for each iteration. When we set the $1+1$D DSTC spatial structures as initial configurations and apply the time-dependent Floquet electrical field (sawtooth wave, maximum amplitude $U_{\text{max}} = 0.2$ V and temporal periodicity $T_E = 0.5$ s), the $2+1$D DSTC (configuration 2) spatial configurations spontaneously emerge (Fig. 3i−k). The system does not exhibit a period-doubling phenomenon over every two periods, as the driving voltage is small, thus showing a time-symmetry-unbroken behavior. The spatial discretization is performed on large 3D square-periodic $80 \times 80 \times 40$ grids, and the spatial derivatives are calculated using finite-difference methods with second-order accuracy. For all simulations, the following parameters are used if not specified: $d/p = 2$, $K_{11} = 6.4 \times 10^{-12}$N, $K_{22} = 3 \times 10^{-12}$N, $K_{33} = 10 \times 10^{-12}$N, $W = 10^{-4}$ J m$^{-2}$ and $\gamma = 77$ mPas. While this approach captures many fine features of quasi-static configurations seen in experiments for some parts of phase diagrams, it does not yield the dynamic features of the time-crystalline structures, this is because the numerical simulation based on Frank-Oseen free energy cannot simulate disclinations or low scalar order parameter regions[35,40], for which we adopt a different approach described below.

## Numerical modeling based on the Landau−de Gennes free energy functional

The quasi-static spatial structures of $1+1$D DSTC (Fig. 3a−d, Supplementary Fig. S1) can be revealed by the Landau−de Gennes free energy functional, in which flexoelectric terms and ion-induced electric field are incorporated. The static or quasi-static spatial structures (Fig. 3) obtained via vectorial or tensorial modeling can be used as initial states of the tensor-based numerical simulations to model the dynamic behaviors. Only tensorial modeling is capable of capturing the details of time crystalline structural evolution because of the singular nature of the observed defect lines.

The tensor order parameter is defined as $\mathbf{Q} = S(\mathbf{nn} - \mathbf{I}/3)$, where $S$ is the LC's scalar order parameter and $\mathbf{I}$ is the identity matrix, and the energy cost of the spatial deformations of $\mathbf{Q}(\mathbf{r})$ can be expressed as:

$$F_{\text{elastic}}^{\text{LdG}} = \int \mathrm{d}^3\mathbf{r}\left\{ \frac{L_1}{2}\frac{\partial Q_{ij}}{\partial x_k}\frac{\partial Q_{ij}}{\partial x_k} + \frac{L_2}{2}\frac{\partial Q_{ij}}{\partial x_j}\frac{\partial Q_{ik}}{\partial x_k} + \frac{L_3}{2}\frac{\partial Q_{ij}}{\partial x_k}\frac{\partial Q_{ik}}{\partial x_j} \right.$$
$$\left. + \frac{L_6}{2}Q_{ij}\frac{\partial Q_{kl}}{\partial x_i}\frac{\partial Q_{kl}}{\partial x_j} + \frac{4\pi}{p}L_4\varepsilon_{ikl}Q_{ij}\frac{\partial Q_{lj}}{\partial x_k} \right\}$$

(4)

where $\varepsilon_{ikl}$ is the Levi-Civita symbol, and $L_i$'s are the elasticity parameters. In addition, the Landau−de Gennes free energy functional includes thermotropic terms that describe the nematic-isotropic transition of the LC:

$$F_{\text{thermotropic}}^{\text{LdG}} = \int \mathrm{d}^3\mathbf{r}\left\{ \frac{A}{2}\left(1 - \frac{U_{\text{LdG}}}{3}\right)Q_{ij}Q_{ji} \right.$$
$$\left. - \frac{AU_{\text{LdG}}}{3}Q_{ij}Q_{jk}Q_{ki} + \frac{AU_{\text{LdG}}}{4}\left(Q_{ij}Q_{ji}\right)^2 \right\}$$

(5)

where $A$ and $U_{\text{LdG}}$ are the nematic material parameters. When an external electric field $\mathbf{E}_{\text{external}}$ is applied, the total electric field $\mathbf{E}$ in the LC is a superposition of $\mathbf{E}_{\text{external}}$ and $\mathbf{E}_{\text{ion}}$, where $\mathbf{E}_{\text{ion}}$ is the ion-induced electric field, and can be calculated by the Poisson equation[29]:

$$\nabla \cdot \left(\boldsymbol{\varepsilon}\cdot\mathbf{E} + \mathbf{P}_{\text{flexo}}\right) = \rho_{\text{el}}$$

(6)

where $\boldsymbol{\varepsilon}$ is the dielectric tensor, $\mathbf{P}_{\text{flexo}}$ is the polarization field due to flexoelectric, and $\rho_{\text{el}}$ is the ionic charge satisfying $\nabla \cdot (\boldsymbol{\sigma}\cdot\mathbf{E}) = -\partial_t\rho_{\text{el}}$, with $\boldsymbol{\sigma}$ being the conductivity tensor. The dielectric and conductivity tensors are related to the Q-tensor via $\boldsymbol{\varepsilon} = \bar{\varepsilon}\mathbf{I} + \varepsilon_a^{\text{mol}}\mathbf{Q}$ and $\boldsymbol{\sigma} = \bar{\sigma}\mathbf{I} + \sigma_a\mathbf{Q}$,

where $\bar{\varepsilon}$ and $\bar{\sigma}$ are the mean dielectric and conductivity constants, respectively, and $\varepsilon_a^{mol}$ and $\sigma_a$ are the dielectric anisotropy and conductivity anisotropy, respectively.

The free energy is supplemented by the following electric coupling terms:

$$F_{electric}^{LdG} = \int d^3\mathbf{r} \left\{ -\frac{1}{2}\varepsilon_0 \bar{\varepsilon} E_i^2 - \frac{1}{3}\varepsilon_0 \varepsilon_a^{mol} Q_{ij} E_i E_j + \zeta_1 \frac{\partial Q_{ij}}{\partial x_j} E_i + \zeta_2 Q_{ij} \frac{\partial Q_{jk}}{\partial x_k} E_i \right\} \tag{7}$$

where $\zeta_i$'s are the flexoelectric constants. The first two terms describe dielectric coupling between $\mathbf{Q}$ and the electric field $\mathbf{E}$, and the last two terms describe the flexoelectric effect.

The surface free energy describing the surface anchoring at the substrates reads

$$F_{surface}^{LdG} = -\int d^2\mathbf{r} \frac{W}{2}(Q_{ij} - Q_{ij}^{(0)})^2 \tag{8}$$

where $Q_{ij}^{(0)}$ defines the preferred orientation and order of LC at the surfaces, corresponding to perpendicular boundary conditions in our experiments. The total Landau−de Gennes free energy $F = F_{elastic}^{LdG} + F_{thermotropic}^{LdG} + F_{electric}^{LdG} + F_{surface}^{LdG}$. The evolution of the system follows the Ginzburg−Landau equation[35]:

$$\partial_t \mathbf{Q} = -\Gamma\left[\frac{\delta F}{\delta \mathbf{Q}}\right]^{st},$$

where $[\ldots]^{st}$ is a symmetric and traceless operator and the relaxation coefficient $\Gamma$ is determined by the rotational viscosity $\gamma_1$ via $\Gamma = 2S_0^2/\gamma_1$ with $S_0$ being the constant equilibrium bulk order parameter ($S_0 = \frac{1}{4} + \frac{3}{4}\sqrt{1 - \frac{8}{3U_{LdG}}}$). Note that the above free energy is time-dependent because of the time-varying external electric field $\mathbf{E}_{external}$.

By applying a constant external electric field, the ions play a similar role to screen the external field as in the Frank−Oseen free energy method, and the results are the same as the quasi-static field configuration shown in Fig. 3. When applying a large sawtooth external electric field, we observe that both the director field in the bulk and the ion-induced electric field exhibit the period-doubling phenomenon locally. In the simulation, a stripe-like periodic pattern appears periodically over time (Fig. 4a, c). After one external temporal period $T_E$, the pattern shifts by a half spatial period. Thus, the full temporal period of the simulated director field corresponds to $2T_E$. The Fourier analysis of the director field shows a clear signal of the period-doubling phenomenon (Supplementary Fig. S3). When we increase the amplitude of the electric field, the period-doubling phenomenon transitions to a disordered phase, as the Fourier spectrum of the director field spans a broad range of frequencies. As we decrease the amplitude further, the temporal periodicity of the local director field is the same as $T_E$, which is in agreement with the experimental results (Fig. 5).

The numerical model parameters are set to be the following: $U_{LdG} = 5$, leading to $S_0 \cong 0.76$, $L_1 = 1.0$, $L_2 = L_3 = L_6 = 0$, $p = 75$, $A = 1$, $\bar{\varepsilon} = 1$, $\varepsilon_a^{mol} = 1$, $\bar{\sigma} = 1 \times 10^{-4}$, $\sigma_a = 5 \times 10^{-5}$, $\zeta_1 = 2$, $\zeta_2 = 11$, and $U_{max}$ has a maximum magnitude of 2.0. The simulation box size is chosen to be $[L_x, L_z] = [300, 150]$ such that channel height to pitch ratio is $\frac{L_z}{p} = 2$. Infinite homeotropic anchoring condition with $W = \infty$ is applied for the two confining substrates, and the periodic boundary conditions are assumed along the $x$-direction.

Our tensorial modeling captures the change of the symmetry breaking during the transition from the initial state at zero field to the dynamic state of the 1 + 1D DSTC. Under homeotropic boundary conditions, the system is topologically neutral as the initial state consists of alternating $+\pi$ and $-\pi$ domain-wall solitons in the x-y plane (Fig. 3a, b). In the 1 + 1D DSTC phase, pairs of disclination lines can be annihilated (Fig. 4l), which also indicates an overall topologically neutral state, dictating that the topological objects' invariants self-compensate. Both in simulations and experiments, we observe that the initial state transforms into the 1 + 1D DSTC state within tens of driving periods, marked by the change of symmetry breaking upon the time crystal formation.

## Simulation of polarizing optical micrographs

The POMs are simulated for the studied structure by means of the Jones-matrix method[34]. We first split the cell into 40 thin sublayers along the z direction. Then we calculate the Jones matrix for each pixel in each sublayer by identifying the local optical axis and ordinary and extraordinary modes' phase retardation for the light traversing the LC medium. The optical axis is determined by the direction of the local average molecular orientation, while the phase retardation originates from the LC's optical anisotropy. We obtain the Jones matrix for the whole LC cell by sequentially multiplying Jones matrices corresponding to each sublayer, and a first-order full-wave retardation plate is included and also described by a Jones matrix. The simulated single-wavelength POM is obtained as the respective component of the product of the ensuing Jones matrix and Jones vectors describing polarizers. To properly reproduce the colored features in POMs seen in experiments, we generate images separately for three different wavelengths spanning the entire visible spectrum (450, 550, and 650 nm) and then superimpose them, according to light source intensities at corresponding wavelengths.

## Calculation of the correlation functions

To calculate the spatial, temporal and spatiotemporal correlation functions, we directly use the data from movie frames. For 1 + 1D DSTC, the $\Phi_{i,j}(\tau)$ represents the signal at column $i$, row $j$ at time $\tau$, and we set $j$ and $r$ are along the direction of spatial spacing, $i$ is perpendicular to the spatial spacing. The spatial correlation function $G(r) \sim \sum_\tau \sum_{i,j} \Phi_{i,j+r}(\tau)\Phi_{i,j}(\tau)$, the temporal correlation function $G(t) \sim \sum_\tau \sum_{i,j} \Phi_{i,j}(\tau)\Phi_{i,j}(\tau+t)$ and the spatiotemporal correlation function $G(r,t) \sim \sum_\tau \sum_{i,j} \Phi_{i,j+r}(\tau)\Phi_{i,j}(\tau+t)$, with background subtraction and normalization applied. The sum over $t$ for Supplementary Fig. S8b is over 1500 drives, and the sum over $r$ is over 200 μm.

## Tracking of quasi-particle-like regions

The displacement trajectories of quasi-particle-like regions in Fig.1j, l, m are analysed using freeware (ImageJ) from the National Institutes of Health. We first convert the movies into grayscale and retain only the frames (when the 2 + 1D DSTC pattern appears) with a time interval $T_E$. Since the quasi-particle-like regions can be recognized as particles in the software, the center position of each quasi-particle-like region, together with its time information, can be directly obtained using the software's tracking functions. We also perform manual tracking, and the results are consistent with those obtained from the automatic tracking.

# Data availability

Source data are available for this paper. All other data that support the plots within this paper and other findings of this study are available from the corresponding author upon request. Source data are provided with this paper.

# Code availability

The codes used for the numerical calculations are available upon request.

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

## Acknowledgments

We thank T. Lee for technical assistance. This research was supported by the US Department of Energy, Office of Basic Energy Sciences, Division of Materials Sciences and Engineering, under contract DE-SC0019293 with the University of Colorado at Boulder. I.I.S. and R.Z. thank the International Institute for Sustainability with Knotted Chiral Meta Matter at Hiroshima University for supporting exchange visits that initiated this collaboration.

## Author contributions

H.Z. performed experiments under the supervision of I.I.S. H.Z. and R.Z. performed the numerical modeling. I.I.S. initiated and directed the research. H.Z. and I.I.S. wrote the manuscript, with feedback and contributions from all authors.

## Competing interests

The authors declare the following competing financial interests: I.I.S. and H.Z. filed patent applications related to discrete space-time crystals submitted by the University of Colorado, and an additional patent was filed concurrently with this paper. The other authors declare no competing interests.
