## [Transparent Peer Review file · Nature Communications]

Emergent discrete space-time crystal of Majorana-like quasiparticles in chiral liquid crystals

Corresponding Author: Professor Ivan Smalyukh

Version 0:

Reviewer comments:

Reviewer #1

(Remarks to the Author)

The authors report on the experimental demonstration of the non-equilibrium system, presented by the electrically driven liquid crystal, that spontaneously breaks the temporal symmetry of the drive. In other words, a subharmonic response to the driving frequency is observed. The work is very well presented, and the physical content of the observed oscillations is an excellent presentation of the non-linear system with parametric response. However, I have difficulty calling this subharmonic response a Time Crystal. As discussed, for example, by R. E. Goldstein in *Physics Today* N. 9, p32-38 (2018), this effect can be related to Faraday instability and is known in non-linear physics. Including the classical systems in the time crystals realm is unjustified for the discussed system.

It might be that time crystal is a useful concept in quantum mechanics, where the equations are linear, and the issue of heating under external forcing is nontrivial. However, for systems like the one presented here, the concepts of classical non-linear physics appear to be adequate.

Finally, the studied system presents a novel platform for studying non-linear phenomena. However, I would suggest resubmitting the paper to a more specialized journal.

Reviewer #2

(Remarks to the Author)

I reviewed the paper entitled "Observation of discrete space-time crystals in a classical soft matter system" from H. Zhao and coauthors where they demonstrated a time and spatial control of a specific liquid crystal by applying surface alignment and periodic electrical drive. The overall space-time periodic order has been observed via polarizing optical microscopy with a lambda plate, enhancing the static birefringence to observe the periodic driven orientation. The spatial periodicity seems to be comparable to the chiral pitch while the temporal periodicity is doubled within a period of the electrical drive.

The paper is overall well written and the several videos available in supplementary materials are certainly helping the visualization of what the authors define a classical time-space crystal. The periodic control over space and time of a large surface in a thin LC cells is certainly extremely interesting, exploiting as far as I can understand a right combination between electrical spatial modulation, chiral pitch, elastic constant and temperature. The chosen LC mixture moreover appears to be robust to several (hours up to 24 hours) voltage-cycles using high to very high fields amplitudes (150V-25/ 5-10 micron~ 300-25 kV/cm is certainly a high field drive) most probably thanks to the ionic doping which avoid screening effects at the electrodes (only explained in the methods at line 474) and ensure local "weak" electric fields – quite in contrast with the actual values used in the experiment.

Despite the exhaustive description of the phenomenon as it appears and which conditions allows different symmetries to take place, in space and time, the physical explanation detailing the overall effect is somehow not clear. This would certainly help the reader to understand many aspects of the work here presented.

For example, in lines 121-122, the authors refer to "LC's complex response to the external drive depending on TE and elastic and dielectric constants" which is quite vague.

I completely miss an elucidation about the origin of the doubling period. The closest the authors come to explaining the mechanism is by saying "These DSTCs can be described as comprising arrays of spatially and temporarily localized solitonic domain walls interacting with each other within the overall out-of-equilibrium setting" which is only appearing at line 167, and left quite unclear. This non-clarity comes quite unexpected given the fact that the authors are presenting 2 models

which matches the observed effects. There should be an intuitive explanation for example for the torque balance equation only mentioned in the methods at line 492, or about the apparently important role of the ion-induced electric field introduced in the LdG free energy functional. Why the Frank-Oseen functional fails in reproducing the doubling period while the LdG does? This is not clear to me, and this part should find a place in the main text too in my opinion.

In the same spirit, looking at the extended data Fig. 5, why as the voltage amplitude increases the noisier the response of the system? What's the correlation between voltage amplitude and the LC frequency response? More generally, how do the details of the driving voltage impact the behavior? Does it only work for uni-polar saw-tooth signals?

What's the role of the anchoring at the surfaces? Only strong homeotropic conditions will allow to form such time-space pattern?

More in detail:

- Line 61: the authors claim to boost the conductance of their LC materials by ionic doping, but do not comment any further. Does the conductivity contribute to the pattern formation and doubling of periodicity observed? How does the period doubling scale with ionic doping? And would it lead to another dimension in the phase diagrams presented (fig 2)? How do the ionic currents expected at these low modulation frequencies play into the underlying mechanism for the period doubling?
- Lines 64 – 72: I would suggest to shorten at least in the main text the discussion about the Polarizing Microscope and move the long discussion in the methods, also because is quite a standard method.
- Line 74: The authors confuse the optic axis of a crystal with the optical axis (geometrical center of an optical system).
- Line 99: what does it mean 'while maintaining a good order'? It is quite vague.
- Lines 136-137: what the authors mean with higher dimensional space-time coordinates?
- Lines 158-160: I am confused. In the first place because this sentence is located just at the end before basically the conclusions where the last words are: "...which potentially could be related to the fact that the DSTC phase is not observed.", and in the second place because I do not understand which is the phase under doubt: is the 1+1D compared to 2+1D? How many defects prevents one to define this structure a time/space crystal?

Reviewer #3

(Remarks to the Author)

Reviewer #4

(Remarks to the Author)

In this work, the authors consider the dynamics of a liquid crystal system under a periodic modulation of the electric field. Depending on the details of the drive (and thickness) they observe a variety of spatial-temporal ordering, in the orientation of the liquid crystal molecules. These patterns are measured through the polarization of the light polarization, which is mapped onto a color scheme.

While the observations in the paper are intriguing and study time crystalline behavior in a novel setting, I find that the manuscript does not address which aspects of the physical setup and protocol enable the observations of the robust period doubling behavior. As a result, it is difficult to relate the observations in the current manuscript with other time crystalline behavior and how the authors' observations can be generalized to other settings.

For example, while the authors emphasize the observation of the time crystalline behavior in a soft matter system, it is unclear what aspect of the soft matter system are crucial for the observation. Is it the director nature of the liquid crystal? Is it the chiral nature of the interactions? Or the strong coupling to a finite temperature bath?

Given that the authors have access to an effective model that accurately captures the dynamics, I think it is important to address some of these questions before I can support the publication in Nature Communications.

Below I expand upon these and other questions (as well as some broad comments), whose answers would greatly improve the manuscript.

Questions:

1) In the abstract, the authors note that the system behaves "like a time-crystal analogues of a **smectic** phase". However the observed time crystalline behaviors corresponds to a period doubling behavior -- in what sense is that related to the smectic phase? Is it because of the spatial ordering?

2) The authors introduce two different free energy models to capture the observed phenomena. From my understanding of discussion in the Methods section, only the Landau-de Gennes approach is able to capture the dynamical observations. Can the authors expand on the differences between the two approaches? Besides a more complex order parameter (built upon the same director as in the Frank-Oseen approach), what physical content is present in the second free energy that enables the better modeling of the system? At the same time, aspect is captured by the Frank-Oseen approach that is not captured by the Landau-de Gennes one?

3) The observations correspond to a simultaneous breaking of the spatial and time translation symmetry. Are the observed spatial symmetry breaking patterns known in previous literature? Are they a requirement for the observation of the temporal ordering?

I ask because there are classes of time crystals that arise from the presence of different symmetry broken states, with the drive serving to toggle the system between the different stable configurations and thus induce a period doubling behavior. Is this an example of one such time crystal or are the spatial patterns only stable in the presence of the period doubling behavior? If not, are these spatial patterns metastable?

4) The authors discuss the presence of different "types" of DSTCs. Is there a precise distinction between the different types of DSTCs? While the visual patterns are slightly different, they appear to have the same symmetry properties.

1) Focusing on Fig. 1h, it is actually unclear that the system is exhibiting a period-doubling response--in particular, all the snapshots appear to have the same pattern. Why is that a DSTC?

5) What happens for larger TE in Fig2a? And smaller temperature for 2b?

6) Looking through the videos, it seems like the patterns described in the paper only emerge for very brief moments, with the system mostly looking homogeneous.

6.1) Do these patterns occur at the max/min of the voltage or near the zero voltage point?

I think that emphasizing this point in the main text is important.

6.2) How does the system preserve a "memory" of the previous pattern even when looking homogeneous?

7) What sets the orientation of the checkerboard pattern in the 2+1D DSTC? Naively, the system has a rotation symmetry around the z-axis, yet the patterns appear to form along the picture axis. Is this some feature of the boundary conditions of the experiments that are not made explicit.

8) I was unable to parse the statement "While defect regions tend to disappear in the space coordinates, they still occasionally emerge in higher-dimensional space-time coordinates.". Can the authors clarify the meaning of higher-dimensional in this context?

9) " Φ is the signals [sic] captured by the quasi-long-range the camera." In this sentence, what constitutes the signal? Naively, there are multiple colors being displayed, so how is that information mapped onto a scalar quantity? Why is such quantity not used when displaying the period doubling behavior in Fig 1k, and instead the position of the middle region is preferred?

10) What is the meaning of the quasi-long-range order in this context? While the system displays a U(1) symmetry (for a rotation of the order parameter around the z axis), the behavior in the time direction has a discrete symmetry, so what causes the power-law decay in time? Is it inherited from the spatial decay of correlations?

11) What is the FT response of the hexagonal order? What is the ratio between the drive frequency and the response frequency?

It would be very surprising if there there is a non-fractional ratio, yet I couldn't find additional data on this observation.

13) How is the "relative noise" quantified in the context of the comment in the discussion? The fact that the system is coupled to a bath at some temperature suggests that there is indeed a large source of noise that is much bigger than the prethermal regime of an isolated quantum system? Or is this a comment with regards to the coherence time of modern quantum platforms?

14) I'm confused how to compatibilize two claims of the paper: (i) that the DSTC can have day long lifetime (Extended Data Fig 7), and (ii) that you observe power-law decay of the spatial temporal correlations (Extended Data Fig 8). These two observations seem to be at odd since one seems compatible with very stable long-lived two point correlation function, where the later suggests that there is no sharp way of defining a long lifetime for the behavior being explored. Can the authors clarify what they are claiming in this context?

15) As a non-expert in the physics of liquid crystal, I was confused by the description of the measurement scheme. Why do different colors get shifted differently by the same pattern of LC directors and how is that related to the average director along the beam path?

Comments:

1) I found the lack of discussion of the periodic drive makes the discussion of the DSTC behavior harder to parse. For example, the authors only note the shape of the profile in the figure and never the main text, and do not address how the shape of the profile was chosen and how much it impacts the observations.

While the authors have a mean-field model that is able to capture the observed dynamics, it remains unclear what feature of the system/drive is able to generate the time crystalline behavior. As such it is hard to understand the origin and nature of the observed phenomena.

2) For 1j, the markers might benefit from having two colors that differ more -- i.e. green and orange. This can be combined with different markers for the two parity of periods.

3) Because the processing for the "center" of the blue region is somewhat obscure, I might suggest the authors append their data with the value/color of the image as a function of time -- e.g. near the black points of 1g, it is clear that the data fluctuates between blue and purple with a sub-harmonic response.

Version 1:

Reviewer comments:

Reviewer #1

(Remarks to the Author)

I have carefully reviewed the authors' responses to the referee reports, including their detailed replies to my own comments and concerns. I appreciate the thoroughness with which the authors have addressed the issues raised, particularly regarding the terminology around time crystals, the distinction from classical nonlinear phenomena, and the broader context and implications of their findings.

I agree with the authors' clarifications and the modifications made to the manuscript. The revisions strengthen the presentation and improve the accessibility of the work. I also find the discussions and suggestions from the other referees to be constructive and well-integrated into the authors' revised manuscript.

In light of the authors' comprehensive and satisfactory responses, as well as the work's overall strong scientific merit, I recommend that the manuscript be accepted for publication.

Reviewer #2

(Remarks to the Author)

The authors responded to all comments raised and thoughtfully explained the missing parts. I can only raise a point regarding the very detailed explanation on the doubling period. I am not an expert in "inter-transformations, generations, and annihilations of coexisting topological solitons and singular disclinations" which seems the founding cause of such phenomena but for a non-expert is really not easy to follow. Could you mitigate the entire added paragraph which I saw transferred 1to1 from the reviewer response to the paper in order to be more open to a non-expert audience?

Reviewer #3

(Remarks to the Author)

Reviewer #4

(Remarks to the Author)

In the reviewed version of the manuscript, the authors have extensively revised their observations, appending two new results: the description of the period doubling dynamics in terms of the creation and annihilation of pairs of topological "Majorana-like" defects, and the observation of a fractional discrete time crystal whereby the sub-harmonic response is not a precise fraction of the original frequency.

While I thank the authors for their efforts in improving the manuscript, I found that they were not able to address some of the questions I raised, and the new results beg a few new points that I think the authors should address before the manuscript should be published.

In general, I have found that the paper is exploring an interesting new regime and demonstrates a novel platform for studying novel out of equilibrium phenomena. However, I have found that the relevance of the work is diminished because the paper fails to engage with the broader set conceptual and categorization questions that come from the observation of time crystalline phenomena, specially in open classical systems. Without such discussion, some of the claims of the paper feel unsubstantiated and unclear. I think that to appeal to the broad audience of Nature Comms, such discussions are relevant, otherwise I agree with Referee 1 that the manuscript is more suitable to a more specialized manuscript.

In an attempt to help the authors, I have divided my comments into main concerns and minor concerns, as per the scope my comment.

Main Concerns:

1) Categorization of the time crystalline phenomena and nature of the ordered phase

As also alluded by Referee 1, the nature of period doubling (and other time crystalline phenomena) in classical systems is

subtle. That is because the role of the external bath, as well as the type and interactions between the different degrees of freedom leads to fundamental different expectations about the nature and the stability of the behavior. For example, in few particle systems, near-integrability can enable the stabilization of long-lived period doubling behavior via the KAM theorem; by contrast, when coupling to a bath that can extract entropy without providing fluctuations, the system can be toggled between different stable sectors in a trivial manner. In both these scenarios, the life-time of the DTC behavior is infinitely long-lived owing to the details of the system.

The present work does not fall into these conditions: it is a strongly open system (the authors must include a meaningful damping rate on their simulations) at a very non-zero temperature, and it treats the dynamics of an extensive number of interacting particles.

As a result, the expectation is much more subtle: the interplay of noise and interactions is expected to destroy the time crystalline order, unless the particular model has some novel mechanism for stabilizing the expected proliferation of defects and their destruction of the time crystalline phase. In this case, the lifetime of the time crystalline phenomena is finite, albeit very long.

(These considerations are extensively explored in arXiv:2305.08904, a paper that the authors reference but do not engage too strongly with).

I understand that the current manuscript is not meant to revisit the entire understanding of the time crystalline phenomena, however, I find that the lack of any discussion of these considerations (in particular the lifetime and what determines it), specially where they authors understand their observations to lie, severely limits ones capability to parse their observations and to contextualize their results. As a result, the authors appear to emphasize the discovery of a completely distinct time crystalline phenomena, but without having strong enough evidence to make such a point.

For the clarity of this work and its contextualization with the broader time crystalline community, I think it is important for the authors to directly engage with the question of stability and heating within the referee report and the manuscript.

With that in mind, I still have some conceptual questions about the nature of their observations.

1.1) Throughout the paper, the authors utilize the center location of the blue domain to build the two-point correlation function that they use to state the observation of quasi-long-range order. I am still confused about this choice. In the book that they reference, power-law correlations are given in terms of the order parameter of the smectic transition (Chapter 6 if I am not mistaken), and they see power-law in spatial correlations with a power that is controlled by the parameters of the system (namely temperature).

The authors claim to see the power-law behavior in the two-time (albeit the evidence is not very strong given the small dynamical range of their fit), using a correlation function over the position center. What is the relationship between the spatial and the time fluctuations? What is the expected form of the power-law exponent in this case? And what does that inform us about the stability of the time crystalline phenomena? And what prevents the authors from considering data for thousands of time steps since they observe the robustness over hours and each period is second scale?

1.2) In the rebuttal, the authors emphasize that the file size for the data is very large, but it seems like they have access to the videos. Is this a statement that the data analysis is very challenging or a statement that they were unable to store videos for all the experiments they conducted and so they had to only store this subset of the information?

Regardless, a discussion about how the middle of the blue regions is obtained should be more prominent as it constitutes a non-trivial piece of data analysis that is important to parse the observations, as well as the correlation function discussed later.

2) Role of the "Majorana-like quasiparticles"

One of the main changes done to the manuscript is the inclusion of a discussion of the period doubling behavior in terms of the dynamics of topological defects within the liquid crystal. These defects, which take half-integer value assuming that the orientation of the liquid crystal is within the 2d plane, are actually their own anti-particles, owing to the rotation freedom of the liquid crystal molecules in 3d space. The authors can then describe the observation of the time crystalline behavior by the creation and annihilation of such pairs of defects along the vertical direction, shifted by $L/2$ every period of the drive.

2.1) I am a bit confused about the meaning of Fig 4 and the dynamics being depicted. In figs g-i, it seems like the dynamics being depicted is the transmutation of the $+1/2$ into $-1/2$ defects and vice versa, rather than their annihilation and creation. Indeed, from the discussion it appears that the notion of $\pm 1/2$ defect is only valid in the large $|U|$ limit where the liquid crystal molecules' orientation is restricted to the XZ plane. In what way is that language (of $\pm 1/2$ defects) meaningful in this context?

2.2) It appears that the dynamics are divided into two moments; a very fast one $0 < t < 0.02T_e$, around which the topological defects bounce a lot between the top and the bottom of the system after the quench, and then an entirely different setting where their motion follows smoothly with the electric field intensity, albeit shift by $L/2$. What guarantees that the oscillations observed in the first few moments die down, and that the defects merge? What determines the position of the formation of the defects as the electric field is ramped up?

One reading of the observed dynamics is that the system has two stable configurations, corresponding to two different lattices of domain walls, and the drive is such that the system is moved between the two and the stability is guaranteed by the stability of the lattice of defects. Is this picture accurate? If so what is added by the discussion of the microscopic dynamics of the "Majorana-like" defects?

2.3) Given these questions (and the previous point about the characterization of the time crystalline behavior), I find that the revised title of the manuscript is not entirely clear. Can the authors clarify what they mean by "discrete space-time crystal of Majorana-like quasiparticles"?

2.4) Finally, I am a bit confused about the (apparent) inconsistency between Fig 3 and Fig 4. In Fig 3, the steady state obtained by the mean field method is such that the defect structure is alternating ($+1/2, -1/2, +1/2, \dots$), whereas in Fig 4, the system now exhibits a fixed pattern of topological defects ($+1/2, +1/2, +1/2$). What is the difference between the two settings? Given that the authors claimed that the mean field accurately captures the dynamics, I am confused by the observations.

3) Observation of the Fractional DTC physics

The other novel result is the observation of the fractional discrete time crystal. Looking through Fig 10, it seems like the system returns to its initial configuration after 3 periods, and thus should exhibit a $1/3$ sub-harmonic response. The shift in the correlation function appears to occur due to the presence of a non-negligible number of defects in the hexagonal lattice that is being prepared.

The assertion that the system is at a fraction p/q suggests that the system performs p orbits over q periods -- this is not supported by the observations, since the response is clearly affected by the presence of defects.

Unless the authors have additional evidence that they have not included in the current version of the manuscript, I am unconvinced of the claim that they have a "fractional discrete time crystal".

Indeed, returning to the point of the characterization of the behavior using the center of the blue region, I find that the authors' analysis further complicates one's ability to parse the significant of this response.

Minor Comments:

1) In the rebuttal, the authors claim. "After several hours, we still observe the time crystal behaviours in the same camera-captured region." Can they make precise what they mean? Locally (in time) the system is still toggling between two configurations? Or there is some form of locking with the initial oscillation in the system? What does this mean about the stability of the time crystalline phenomena?

2) In the rebuttal, the only time the authors engage with the concern about the analysis of the data is in the addition of the oscillation of the amount of blue within an arbitrarily chosen square of the image. It is not explained why this is the correct metric (the blue channel or the chosen region), nor how it relates to the metric being analyzed. I find that the authors engaged with the letter of my previous comment but didn't engage with the point that I was raising, namely that the data analysis pipeline is complex so it is hard to parse the importance of the correlation function

3) In the rebuttal, the authors note the experimental settings when asked about the "types" of DSTCs. I am confused about this response since the authors continue to emphasize that they observe different "types" of DSTCs, but cannot highlight what is the sharp feature that distinguishes them. It seems like all exhibit some square pattern of correlations that shifts between different periods of the drive. While the shape of the pattern is different, the symmetry properties seem to be the same, they all live in a square lattice. Is there something that I am missing? How do the authors define the different types? Are these patterns captured by the mean-field numerics?

4) The authors now emphasize the defect perspective on the dynamics of the $1+1D$ DSTC. Does something apply to the $2+1D$ case? If not why? What about the hexagonal case?

Version 2:

Reviewer comments:

Reviewer #2

(Remarks to the Author)

I reviewed the manuscript on "Emergent discrete space-time crystal of Majorana-like quasiparticles in chiral liquid crystals" in its entirety, in light of all the changes made due to all review feedback in the last round. The result is a well rounded overview of the period-doubling phenomena found in experiment. While I am not an expert on time crystals, and therefore hesitant to apply this terminology, the computational modeling based on Frank-Oseen and Landau-de Gennes theory appears to capture the overall dynamics well.

In particular, my request for a description suitable to a more general audience is addressed satisfactory. I have no further remarks that would improve the manuscript within a reasonable time frame.

Reviewer #3

(Remarks to the Author)

Reviewer #4

(Remarks to the Author)

The authors have significantly improved their paper, adding important discussions and clarifying some important aspects of their work. At this point, I understand their experiment and observations and I find them novel and interesting to the broader audience. However, I still find some points confusing or ill-explained in the manuscript. Pending addressing these points, I support publication in Nature Communications.

Clarification of the definition of the Majorana-like quasiparticles

1) I remain confused about the definition of the Majorana quasi-particle and its relationship to the director orientation profile. The authors emphasize that the homotopy class of the director in 3D is $\pi^1(S^2/Z_2) = Z_2$, which leads either defect or no-defect. In this language the origin of the Majorana is somewhat clear, since two defects annihilate one another -- this is also the discussion within the PNAS cited by the paper.

2) Despite being equivalent, when the applied field is large, the director appears to be restricted to the xz plane and the winding number becomes well-defined to the point that one can assign $\pm 1/2$ charge to each "Majorana-like" particle. The authors emphasize how the DTC dynamics can be understood in terms of the annihilation and creation of pairs of such pairs of particles (which is the more general language are the Majorana quasiparticles)

3) This begs three questions:

1) This discussion all occurs in the translationally invariant case, where these defects are not point-like but rather extended objects along the y direction. In this case the director orientation is ill-defined, not on a single point, but along a line, changing the dimensionality of the defect. In this case, why does the same characterization of the defect still persist?

2) The authors also emphasize (in Page 6) how the nature of the domain wall is also captured by the same homotopy class. These, however, are extended two dimensional objects. How does the homotopy class of the solitons affect our understanding of their dynamics? Does it simply state that any two solitons can annihilate one another? At the same time, is there some important connection between the presence of the aforementioned defect lines and the solitons (more on this below)? And what is the relationship between the solitons and the Neel/Bloch domain walls emphasized in Fig 4?

3) The numerical evidence (and ability to peer into the dynamics of the director) seems to be limited to the 1+1D case. In this case, I understand how the line defects (labeled by $\pm 1/2$) paint a simple description of the dynamics. However, the authors extend their claims of the Majorana description to the 2+1D DTC (see abstract quote below), where the authors have no picture or description in terms of the director topological defect.

In the report, they have also stated that "these phenomena are rather complex and more disciplinary, with the power of numerical modelling still insufficient to capture details that could be presented in an accessible form".

To this end, I think that either additional evidence / descriptions are added to the paper, or the claim of a time crystal driven by the dynamics of Majorana-like particles should be restricted to the analysis of the 1+1D case (upon a clarification of the above point)

> "Here we describe the observation of classical analogues of both 1+1-dimensional and 2+1-dimensional discrete space-time crystals in a liquid crystal system driven by a Floquet electrical signal. These classical time crystals comprise particle-like structural features and exist over a wide range of temperatures and electrical driving conditions. The phenomenon-enabling period-doubling effect comes from their topological Majorana-like quasiparticle features"

Confusing point about the initial state:

In my previous reply I alluded to some confusion with regards to the relationship between the profile in Fig 3 and 4 which I now better understand how the two correspond to the initial state vs a description of the dynamical process, respectively. From reading the text, I still find these two plots and their relationship hard to parse (magnified by the lack of description of Fig 3 in the main text, something that is more extreme for Fig 2 which I would recommend moving to the Methods or SM).

However, even with this clarification, the role of the initial state remains confusing. Let me first start by laying out my understanding/confusion.

1) In Fig 3 the authors emphasize the presence of soliton domain whose winding along the XY plane is alternative between $+$ and $- \pi$ and there are no defect lines of the form presented in Fig 4 (the $\beta = \pm 1/2$ defects). Indeed, in Fig.4 it seems like the directors predominantly point along the z axis even when the applied field is small (around $t = 0.5TE$ Fig 4h).

2) This suggests that the first period is very important because it takes a symmetry breaking state (lattice of solitons) and maps it into a completely different symmetry breaking state (a lattice of Néel domain walls) which are embedded in a different plane of the liquid crystal (note that in the response, the authors still called both domain walls, Néel domains, although they emphasize they are distinct because of the plane of the orientation).

3) The different defects and their transformations are made extra confusing when looking at Fig 1d where the 1+1D response looks the same as the initial state presented in Fig 3a, however, the authors are emphasizing that the director ordering is actually very different (Fig 3g vs Fig 4h).

4) Moreover, it seems to be the case that the same initial state (1D symmetry breaking) induces the dynamics of the 2+1D state ("When we set the 1+1D DSTC spatial structures as initial configurations and apply the time-dependent Floquet electrical field (sawtooth wave, maximum amplitude $U_{max} = 0.2V$ and temporal periodicity $TE = 0.5s$), the 2+1D DSTC

(configuration 2) spatial configurations spontaneously emerge (Fig. 3m-o) in each drive"). This point is stated in the methods, but further emphasizes that there is a lot of complexity in the way that the initial state induces the time crystalline behavior.

This distinction between the initial state and the dynamical state is very important, but it is not addressed. Much like a careful discussion of the nature of the defects (and a clearer justification of their Majorana nature), it warrants more space in the text to guide the reader through the differences between the different defects considered (the two domain walls and the Majorana defects), what their relationship is and how to understand the transformation between the initial state and the dynamical steady state.

Minor points:

- In S7, the authors emphasize having data averaged over 1500 drives (I assume this means 1500 periods since the caption emphasized different driving durations), yet $G(t)$ only goes to 100. This axis should be extended to help emphasize the nature of the decay.

- Ln 9. Page 1: "period of the driven system" should read "period of the system's dynamics"

Version 3:

Reviewer comments:

Reviewer #4

(Remarks to the Author)

The authors have carefully engaged with the comments I have made in previous reports and have expanded and clarified their observations and conclusions.

At present, I do not have more comments that I think are crucial for the understanding of the paper, and, as such, I am happy supporting the publication of the manuscript in Nature Communications.

Emergent discrete space-time crystal of Majorana-like quasiparticles in a chiral liquid crystals

H. Zhao, R. Zhang and I. I. Smalyukh

Summary of Changes and Responses to Requests/Remarks

Authors:

We appreciate the consideration of our manuscript. We thank the Referees for their positive reviews and cogent comments. We are pleased that Referees report that our work “ is very well presented, and the physical content of the observed oscillations is an excellent presentation ... ” and our studied system “presents a novel platform for studying non-linear phenomena” (Referee #1), that our paper “ is overall well written”, evaluate our studies “define a classical time-space crystal” and “... is certainly extremely interesting” (Referee #2) and that our observations are “are intriguing and study time crystalline behavior in a novel setting ...” (Referee #4). In accordance with the comments and suggestions put forward by the referees, we have made clarifications and significant revisions, which are detailed point-by-point below. These revisions fully address all remarks and suggestions. We also feel that they improve the accessibility of the paper significantly. We would like to thank the Referees for the thoughtful valuable comments that helped direct these improvements.

Changes to Account for the Referees’ Suggestions.

Report of Referee 1:

The authors report on the experimental demonstration of the non-equilibrium system, presented by the electrically driven liquid crystal, that spontaneously breaks the temporal symmetry of the drive. In other words, a subharmonic response to the driving frequency is observed. The work is very well presented, and the physical content of the observed oscillations is an excellent presentation of the non-linear system with parametric response.

Authors:

We thank the Referee #1 for the above positive remarks, as well as for the helpful comments.

Referee 1:

*However, I have difficulty calling this subharmonic response a Time Crystal. As discussed, for example, by R. E. Goldstein in *Physics Today* N. 9, p32-38 (2018), this effect can be related to Faraday instability and is known in non-linear physics. Including the classical systems in the time crystals realm is unjustified for the discussed system.*

It might be that time crystal is a useful concept in quantum mechanics, where the equations are linear, and the issue of heating under external forcing is nontrivial. However, for systems like the one presented here, the concepts of classical non-linear physics appear to be adequate.

Authors:

We thank the Referee for these comments. As mentioned in the suggested paper, the same issue of *Physics Today* includes a follow-up paper by Norman Y. Yao and Chetan Nayak, which discusses

why time crystals differ from previous studies, including classical time crystals. Indeed, the "time crystal" referred to in the original paper should be classified as a many-body-localized discrete (Floquet) time crystal, which, as the Referee points out, exhibits nontrivial properties under external driving and avoids thermalization due to many-body localization. However, as the field has rapidly evolved, researchers used different mechanisms to avoid heating (thermalization), such as introducing dissipation (dissipative time crystals) or preparing the system in a prethermal state (prethermal time crystals, which can exhibit exponentially long lifetime of the time crystal phase). Although these findings were initially based on quantum systems, classical counterparts of theoretical prediction have been identified in many cases. For example, friction and noise have been used to realize "classical discrete time crystals" (Ref. 18), and prethermal states have been used to study "classical prethermal discrete time crystals" (Refs. 19 and 20). These systems, described by classical nonlinear physics, not only provide new ways to study and understand subharmonic behavior but also bridge the gap between quantum and classical systems.

Additionally in the revised paper, we carefully studied the quasi-hexagonal lattice pattern and found that its response exhibits a fractional relationship ($\sim 10/3$) to the external drive (Fig. 10). Interestingly, similar fractional responses (0.3 and $100/33$) have recently been observed in two quantum systems: a numerical simulation of a spin-1/2 system and an experimental study on Rydberg atoms. The observation of such close fractional responses across different systems raises the question of whether there is a general underlying mechanism (like period doubling arising from \mathbb{Z}_2 symmetry or \mathbb{Z} symmetry). We believe that aligning these systems under the same framework will help distinguish their differences and identify commonalities, advancing our understanding of "time crystals."

Furthermore, our system may differ from Faraday instability. Systems like Faraday waves are described by the Mathieu equation, which treats the system as a single-body or continuum medium. In contrast, our system is challenging to describe using a single-body equation: utilizing computer simulations, we illustrate that the period-doubling effect is intimately related to the inter-transformations, creations and annihilations of coexisting topological solitons and singular disclinations. Remarkably, the different states of these topological objects can be viewed as the particle and anti-particle states of the observed Majorana-like quasiparticles (a classical analogue of Majorana particles that was recently discussed in the context of active nematic liquid crystal systems [26-28]) forming our 1+1D space-time crystals (Fig. 4 and Supplementary Fig. S1).

We thank the Referee for these comments again, which prompted us to carefully reconsider the description and the use of terminology in the rapidly evolving research field of time crystalline phenomena. In the revised manuscript, overall, we have carefully checked terminology and softened claims.

Referee 1:

Finally, the studied system presents a novel platform for studying non-linear phenomena. However, I would suggest resubmitting the paper to a more specialized journal.

Authors: We appreciate Referee #1 noting that "the studied system presents a novel platform for studying non-linear phenomena." As shown in the manuscript, our system not only serves as a platform for observing discrete time symmetry breaking but also provides a unique opportunity to study spatio-temporal defects and classical Majorana-like quasiparticles, which have not been elaborately studied in previous work. Our system also exhibits similarities to both classical time crystal systems (e.g., the "boil out" effect in Ref. 18, anti-ferromagnetic order in space and time as in Refs. 18–20, etc.) and quantum time crystal systems (e.g., similar fractional responses, etc.) that have not been reported in a

classical system context, to the best of our knowledge.

Additionally, our system has potential real-world applications. Liquid crystals are technologically important materials, with the LC-based global industry estimated to approach 1 trillion USD annually. Electrically controlled LCs are at the heart of modern display technologies, where the refresh rate of screens ranges from 60 Hz to hundreds of Hz. Could discrete time symmetry breaking occur under certain conditions, and could these phenomena play a role in technological applications? The transition from purely fundamental curiosity to technological utility can lead to further growth of research efforts studying these phenomena. Moreover, we are pleased to share a good news: following a careful review, our university's Technology Transfer Office has expressed strong interest in discrete space-time crystals within liquid crystalline systems & filed our patent application with the United States Patent and Trademark Office, making our work ready for public disclosure.

Therefore, we believe our study is of very broad interest for the scientific, technological, and engineering communities.

Report of Referee 2:

I reviewed the paper entitled "Observation of discrete space-time crystals in a classical soft matter system" from H. Zhao and coauthors where they demonstrated a time and spatial control of a specific liquid crystal by applying surface alignment and periodic electrical drive. The overall space-time periodic order has been observed via polarizing optical microscopy with a lambda plate, enhancing the static birefringence to observe the periodic driven orientation. The spatial periodicity seems to be comparable to the chiral pitch while the temporal periodicity is doubled within a period of the electrical drive.

The paper is overall well written and the several videos available in supplementary materials are certainly helping the visualization of what the authors define a classical time-space crystal. The periodic control over space and time of a large surface in a thin LC cells is certainly extremely interesting, exploiting as far as I can understand a right combination between electrical spatial modulation, chiral pitch, elastic constant and temperature.

Authors:

We thank the Referee #2 for this very thoughtful and positive report noting that our paper "is overall well written", evaluate our studies "define a classical time-space crystal" and "... is certainly extremely interesting", as well as for the helpful and detailed suggestions. We have accounted for all suggestions, as detailed below, which helped us to further improve the manuscript.

Referee 2:

The chosen LC mixture moreover appears to be robust to several (hours up to 24 hours) voltage-cycles using high to very high fields amplitudes (150V-25/ 5-10 micron~ 300-25 kV/cm is certainly a high field drive) most probably tanks to the ionic doping which avoid screening effects at the electrodes (only explained in the methods at line 474) and ensure local "weak" electric fields – quite in contrast with the actual values used in the experiment.

Authors: We thank the Referee for this comment. As an estimate, the maximum screening voltage $U_{\text{screen}} \sim 0.001 d^2 \rho e / m \epsilon_r \epsilon_0$, where 0.001 refers to the weight concentration 0.1%, m is the mass of single molecule of CTAB, d is the cell thickness, ρ is the density of LC, e is the charge of electron and

ϵ_r is relative permittivity of the LC. Considering that d is 5-10 μm , ρ is 1.03kg/m³ and ϵ_r is in the order of 10^2 at 2Hz, the maximum ability of U_{screen} is about 30-120V. When combined with the screening effect, the actual electric field applied across the nematic bulk can be one to two orders of magnitude smaller than the maximum external drive field. Additionally, as suggested by Referee 3, we have added a new figure (Fig. 2f) showing that the most obvious spatial patterns emerge when the amplitude of the external drive voltage is close to zero. As the external voltage increases, the LC pattern tends to be uniform under crossed polarizers but preserves “memory cores” (Fig. 4e, can be seen with the help of an enhanced contrast POM images) related to the mechanism of period-doubling detailed in the revised manuscript.

Referee 2:

Despite the exhaustive description of the phenomenon as it appears and which conditions allows different symmetries to take place, in space and time, the physical explanation detailing the overall effect is somehow not clear. This would certainly help the reader to understand many aspects of the work here presented.

Authors: We thank the Referee for this helpful remark. We have detailed the mechanism of the overall effect in the responses to the comments below, as well as in the revised manuscript, we provide a deeper description of the phenomena observed in this work.

Referee 2:

For example, in lines 121-122, the authors refer to “LC’s complex response to the external drive depending on TE and elastic and dielectric constants’ which is quite vague.

Authors: We thank the Referee for this comment. In the revised manuscript, we have clarified the corresponding sentences and provided a more detailed and precise explanation to address the vagueness problem.

Referee 2:

I completely miss an elucidation about the origin of the doubling period. The closest the authors come to explaining the mechanism is by saying “These DSTCs can be described as comprising arrays of spatially and temporarily localized solitonic domain walls interacting with each other within the overall out-of-equilibrium setting” which is only appearing at line 167, and left quite unclear. This non-clarity comes quite unexpected given the fact that the authors are presenting 2 models which matches the observed effects. There should be an intuitive explanation for example for the torque balance equation only mentioned in the methods at line 492, or about the apparently important role of the ion-induced electric field introduced in the LdG free energy functional. Why the Frank-Oseen functional fails in reproducing the doubling period while the LdG does? This is not clear to me, and this part should find a place in the main text too in my opinion.

Authors: We thank the Referee for these helpful questions. The Frank–Oseen free energy based model describes very well the initial states when only nonsingular solitonic structures are present, but it cannot describe the states with singular half-integer disclinations that emerge during the voltage driving scheme, which ultimately yields the period doubling effect. Utilizing LdG free energy functional based simulations, we reveal that the period doubling effect is intimately related to the inter-transformations, generations, and annihilations of coexisting topological solitons and singular disclinations (Fig. 4g–o). The +1/2 and –1/2 singular disclination structures are the building blocks of 1+1D time crystals, which can be treated as a Majorana-like quasiparticle and its anti-particle (Fig.

4k) [26-28], because the disclination profiles smoothly transform as spinors following the Majorana equation (see Ref. 27 for details). In the time crystal phase, the particle-antiparticle inter-transformation occurs when the external voltage U smoothly changes from negative to positive (Fig. 4g-i). The director field around the disclination region can be characterized by the characteristic twist angle β ($\beta \in [0, \pi]$)[36,37]. When $U < 0$, β equals to π at the top and 0 at the bottom, indicating a $-1/2$ and $+1/2$ wedge disclination[38] in the two-dimensional cross-sections orthogonal to the defect lines, respectively. The winding number $\pm 1/2$ of disclinations relates to the accumulated angle of $\mathbf{n}(\mathbf{r})$ rotation as one circumnavigates the disclination core once, divided by 360° , with the positive sign corresponding to the counterclockwise rotation matching that of circumnavigation and negative sign referring to the opposite clockwise case. Between the two disclination regions, the $-\pi$ -rotation Néel domain wall solitons (with $+/-$ defined by counterclockwise/clockwise rotation when traversing the topological soliton from left to right) are spatially embedded (Fig. 4g). The domain wall solitons are labelled as elements of the first homotopy group[39], $\pi_1(\mathbb{S}^2/\mathbb{Z}_2) = \mathbb{Z}_2$, and allowed to smoothly inter-transform to have 2D cross-sections of different types while here being terminated by two disclinations $\pi_1(\mathbb{S}^2/\mathbb{Z}_2) = \mathbb{Z}_2$ also inter-transforming between geometrically different but topologically the same states. As the voltage U increases from negative values to zero (Fig. 4h), the angle β characterizing both wall-terminating disclinations smoothly changes to $\pi/2$ (corresponding to pure twist winding), and the initial $-\pi$ -rotation Néel domain wall solitons (bend-splay structures) transform to Bloch domain wall solitons (with pure twist structures). As the voltage U further increases to $U > 0$ (Fig. 4i), β changes to 0 at the top and π at the bottom, and the Bloch domain wall solitons transform to the $+\pi$ -rotation Néel domain wall solitons. The transition is continuous during the linear voltage change, and the positions of the disclinations and solitons do not shift spatially. The array of topological quasiparticles formed by solitonic walls terminated on singular defect lines maintains its topological nature (Fig. 4i) until U suddenly changes from the positive maximum value to the negative maximum value (Fig. 4l), at which point topological particle-antiparticle pairs annihilate (Supplementary Fig. S1a) and re-generate (Supplementary Fig. S1b), while the locations of the emerged quasiparticles are found synchronously shifted by half a spatial period ($L/2$) away from the previous ones. After the generation, compared to the structures of the director field one temporal period (T_E) before (Fig. 4g-i,m-o), the disclinations and topological solitons are the same, with an $L/2$ shift, so that the system adopts the exact same configuration every $2T_E$, resulting in the period-doubling phenomenon. In the experiments and simulations alike (Fig. 4d,e), even when the optical images appear almost homogeneously dark under a high electric field between crossed polarizers, the defects and director deformations do not fully disappear, as evident from the enhanced contrast POM images that reveal effective "memory" of the periodic deformations that allows for correlating positions of these topological objects within $2T_E$.

The details of the topological transformations make it especially clear why the Frank–Oseen functional based numerical modeling fails to reproduce the period-doubling behavior: this is because, unlike the LdG free energy functional, it cannot simulate disclinations and reduced scalar order parameter regions (Refs. 35 and 39). However, the LdG free energy functional is computationally time-consuming for simulating large-scale structures. Therefore, we use the Frank–Oseen free energy to computationally obtain fully nonsingular quasi-static configurations in spatiotemporal diagram regions, which can be directly compared in detail to experimental POM and 3PEF-PM images. We thank the Referee again for all these insightful questions. In the revised manuscript, we have detailed the methodology and explanations.

Referee 2:

In the same spirit, looking at the extended data Fig. 5, why as the voltage amplitude increases the noisier the response of the system? What's the correlation between voltage amplitude and the LC frequency response?

Authors: We thank the Referee for these questions. In the simulation, we find that when the system is near the period doubling point or when the voltage is low enough, the noise is small. Otherwise, the noise is relatively large. In the revised manuscript, we add more results as we change the voltage amplitude to clearly show this effect (Supplementary Fig. S2).

Referee 2:

More generally, how do the details of the driving voltage impact the behavior? Does it only work for uni-polar saw-tooth signals?

Authors: We thank the Referee for these questions. As shown in the phase diagrams in Fig. 5, the amplitude and temporal periodicity of the driving voltage are important control parameters to affect the time crystal behavior, and larger temporal perturbations of the driving field can also destroy the time crystal phase (Fig. 6d,e). Regarding the waveform shape of the external drive, we have not observed a clear time crystal behavior with other waveform shape (triangle, square, and sinusoid wave), but we do not exclude the possibility for proper conditions and material parameters to be found for this to occur. Related to the period-doubling mechanism described above, the transitions of topological solitons and disclination regions could be a key point to design a different waveform.

Referee 2:

What's the role of the anchoring at the surfaces? Only strong homeotropic conditions will allow to form such time-space pattern?

Authors: We have tried planar conditions, but we did not observe time crystal behavior under these conditions. For homeotropic conditions with a weak anchoring strength, it is harder to observe the time crystal phase either, with the relatively strong homeotropic anchoring based on described preparation seemingly being ideal. We thank the Referee for these questions and the opportunity to provide this information.

Referee 2:

More in detail:

• Line 61: the authors claim to boost the conductance of their LC materials by ionic doping, but do not comment any further. Does the conductivity contribute to the pattern formation and doubling of periodicity observed? How does the period doubling scale with ionic doping? And would it lead to another dimension in the phase diagrams presented (fig 2)? How do the ionic currents expected at these low modulation frequencies play into the underlying mechanism for the period doubling?

Authors: We thank the Referee for these questions. In experiments, we find that the DSTC phase occurs when ion doping is between 0.05-0.1 wt%. We tested pure LC without ionic doping and observed the disordered and the time-symmetry-unbroken states. With higher ion doping (>0.2 wt%), no time crystal behavior is observed either. We observed the same behavior in simulations, when slightly increase or decrease ion concentrations (Supplementary Fig. S3a), the DSTC preserves. However, without ion doping or with too high ion doping concentration, there is no DSTC phase. In addition, we also plot the ionic current from simulations, we find that the ionic current also has a spatial/orientation distribution dependence (Supplementary Fig. S3b), which relates to the topological solitons and disclinations in the system coupled with medium's anisotropy.

Referee 2:

• Lines 64 – 72: *I would suggest to shorten at least in the main text the discussion about the Polarizing Microscope and move the long discussion in the methods, also because is quite a standard method.*

Authors: We thank the referee for this suggestion. We have shortened and moved the corresponding description to the methods part.

Referee 2:

• Line 74: *The authors confuse the optic axis of a crystal with the optical axis (geometrical center of an optical system).*

Authors: We thank the referee for this helpful remark, which we accounted for.

Referee 2:

• Line 99: *what does it mean 'while maintaining a good order'? It is quite vague.*

Authors: We thank the referee for this question. In the revised manuscript, we have modified it to "while maintaining the period-doubling behaviour" to provide a clearer and more precise description.

Referee 2:

• Lines 136-137: *what the authors mean with higher dimensional space-time coordinates?*

Authors: We thank the Referee for this question. Illustrating this as an example, in spatial coordinates, a defect in the xy-plane may appear at $t=t_0$ and disappear at $t=t_1$. If we consider only the two-dimensional xy-plane, the defect eventually vanishes. However, when we include the time coordinate in an xy-t frame (referred to as higher-dimensional space-time coordinates compared to the xy frame), the temporal axis (though intrinsically different from spatial axes) behaves similarly to the z-axis. In this extended frame, the defect does not simply "appear" or "disappear"; instead, it emerges at specific positions along the t-coordinate. We thank the Referee again for this remark. Since the related discussion is not central to the DSTC behaviours and may cause confusion, in the revised manuscript, we removed this sentence to ensure the clarity of the defect descriptions.

Referee 2:

• Lines 158-160: *I am confused. In the first place because this sentence is located just at the end before basically the conclusions where the last words are: "...which potentially could be related to the fact that the DSTC phase is not observed.", and in the second place because I do not understand which is the phase under doubt: is the 1+1D compared to 2+1D? How many defects prevents one to define this structure a time/space crystal?*

Authors: We thank the Referee for this helpful remark. From the video, we observe that the quasi-hexagonal lattice patterns do not recur every two or three periods in response to the external drive. By carefully analyzing the relationship using Fourier Transform and temporal correlation functions, we find that the internal temporal periodicity is approximately 10/3 times the external temporal periodicity, indicating a fractional discrete time crystal. Regarding defects, we observe 15 defects within a region (including 10 hexagonal units) over 200 external drives. This explains the strong noise around the peak

in the FT spectrum and the rapid decay of the temporal correlation function compared to the case shown in Fig. 9. In the revised manuscript, we have added a new figure (Fig. 10) and provided a detailed description of this phenomenon. We thank the referee 2 again for this thoughtful report, which has significantly enhanced the presentation and clarity of our findings.

Report of Referee 3:

Reviewer #3 (Remarks to the Author):

Report of Referee 4:

Reviewer #4 (Remarks to the Author):

In this work, the authors consider the dynamics of a liquid crystal system under a periodic modulation of the electric field. Depending on the details of the drive (and thickness) they observe a variety of spatial-temporal ordering, in the orientation of the liquid crystal molecules. These patterns are measured through the polarization of the light polarization, which is mapped onto a color scheme.

While the observations in the paper are intriguing and study time crystalline behavior in a novel setting, I find that the manuscript does not address which aspects of the physical setup and protocol enable the observations of the robust period doubling behavior. As a result, it is difficult to relate the observations in the current manuscript with other time crystalline behavior and how the authors' observations can be generalized to other settings.

For example, while the authors emphasize the observation of the time crystalline behavior in a soft matter system, it is unclear what aspect of the soft matter system are crucial for the observation. Is it the director nature of the liquid crystal? Is it the chiral nature of the interactions? Or the strong coupling to a finite temperature bath?

Given that the authors have access to an effective model that accurately captures the dynamics, I think it is important to address some of these questions before I can support the publication in Nature Communications.

Authors:

We thank the Referee #4 for this very thoughtful and detailed report saying that our observations “are intriguing and study time crystalline behavior in a novel setting ...”, as well as appreciate the helpful suggestions.

In the revised manuscript, we have expanded the discussion on the connections, similarities, and commonalities between our space-time crystals and other time crystals. We have also further elaborated on the unique advantages of utilizing the particle like topological solitons and disclinations within the liquid crystal platform for realizing time crystals. While we observe it as an emergent phenomenon in a very specific type of soft matter system, our work demonstrates that such effects can emerge in a classical setting, which is a result of general significance, and we foresee that they will lead to future discoveries of similar effects in other types of liquid crystals and other types of

soft/classical matter.

Regarding the discussion of generalization to other time crystalline behavior, we would first like to thank Referee 4 for these insightful questions (Questions 11 & 13), which inspired and guided us to investigate the emergence of temporally fractional response periods of the spatially quasi-hexagonal lattice. Surprisingly, although our system is classical, the fractional periods we observed are close to those reported in a theoretical study and an experimental study on quantum systems, which we will describe in detail later. Additionally, we have discovered that our system exhibits intrinsic and profound connections to other time crystals in several aspects. For example, we observed the "boil out" phenomenon—where space-time crystals emerge from disorder (Ref. 18). Our system also provides a validation of space-time group theory (Ref. 24). Furthermore, the role of anti-ferromagnetic order in both space and time, and other features highlights more significant connection (Refs. 18-20). These findings not only help bridge the gap between classical and quantum time crystals but also establish a broader connection to soft matter and other classical systems.

In terms of the advantages of utilizing the liquid crystal platform, we show that the observed phenomena arise from a combination of intrinsic soft matter properties, particle-like behavior, topological features, and other factors. The many-body interactions between particle-like solitonic and singular defect objects are of key importance. The strong (relative to thermal energy) elasticity-mediated interactions between quasi-particles ensure that the system remains relatively stable even in the presence of temporal perturbations and spatial defects (Fig. 4a,b). As a result, once the time crystal phase emerges under suitable conditions, we can often observe the long lifetime of time crystallization. Regarding the mechanism of time crystal formation in our system, we have further elaborated on the mechanism in the revised manuscript using a 1+1D space-time crystal as an example (Fig. 4). We find during each drive, the inter-transformations, creations/generations, and annihilations of coexisting topological solitons and singular disclinations play a crucial role, which will be detailed later in the response to Question #2&6. Remarkably, the different states of these topological objects can be viewed as the particle and anti-particle states of the observed Majorana-like quasiparticles (a classical analogue of Majorana particles that were recently studied in active nematic LC systems [26-28]). These particle-like structures in liquid crystal systems also show potential realizations in other soft matter systems. Furthermore, the spatial scale of the liquid crystal system that forms space-time crystals ranges from micrometers to millimeters. This allows us to directly observe the time crystal under a microscope. In contrast, many previous studies on discrete time crystals in quantum systems were limited to either measuring on-site period-doubling signals in systems with fewer than tens of spins or qubits or detecting global signals such as total magnetization. Our liquid crystal system, however, enables the analyses of large-scale spatiotemporal structural relationships.

We anticipate that soft matter media will allow for discoveries of many discrete, continuous, and fractional time and space-time crystals of different dimensionalities.

We would like to thank the Referee 4 again for these insightful and helpful questions and comments. We have accounted for all suggestions, as detailed below, which helped us to further improve the manuscript.

Referee 4:

Below I expand upon these and other questions (as well as some broad comments), whose answers would greatly improve the manuscript.

Questions:

*1) In the abstract, the authors note that the system behaves "like a time-crystal analogues of a ****smectic**** phase". However the observed time crystalline behaviors corresponds to a period doubling behavior -- in what sense is that related to the smectic phase? Is it because of the spatial ordering?*

Authors:

We thank the Referee for this helpful question, in LC systems a strictly 1D spatially periodic lattice cannot exist due to thermal fluctuations (Ref. 41). The spatial location where each stripe region emerges has a small randomness, leading to a quasi-long-range order in spatial coordinate, which is the characteristic of the well-known smectic phase in LCs. However, a more interesting thing we observed is that when we calculate the temporal correlation of the pattern, the correlation function also exhibits a power-law decay behavior, which indicates a quasi-long-range order (smectic-like) in time (Fig. 9) and (as we see from informal discussions) is already attracting strong interest of theorists as a phenomenon. In the revised manuscript, we have improved the corresponding descriptions to provide a clearer explanation of this phenomenon.

Referee 4:

2) The authors introduce two different free energy models to capture the observed phenomena. From my understanding of discussion in the Methods section, only the Landau-de Gennes approach is able to capture the dynamical observations. Can the authors expand on the differences between the two approaches? Besides a more complex order parameter (built upon the same director as in the Frank-Oseen approach), what physical content is present in the second free energy that enables the better modeling of the system? At the same time, aspect is captured by the Frank-Oseen approach that is not captured by the Landau-de Gennes one?

Authors:

A major difference between these two approaches is that the Frank–Oseen free energy functional is vector based, while the Landau–de Gennes free energy is tensor based. The latter not only includes the orientation information of the director field but also captures the degree of orientational alignment relative to the reference direction, described by the scalar order parameter. For example, the Landau–de Gennes free energy model can accurately simulate singular disclination structures (with low scalar order parameter), whereas the Frank–Oseen free energy cannot (Refs. 35 and 39). The interactions and transformations of the disclinations and topological domain wall solitons under external drive play an important role for the period-doubling phenomenon, which we will describe in detail below in our responses to Question #6.

However, the Landau–de Gennes approach involves more parameters than the Frank–Oseen approach, making it computationally time-consuming for simulating large-scale structures while using all elastic constants and material parameters known from experiments. Therefore, we use the Frank–Oseen free energy to obtain quasi-static configurations, which can be directly compared in detail to experimental POM and 3PEF-PM images, effectively depicting what is the initial state for the dynamic evolution of the discrete time crystal. We thank the referee for this insightful question. In the revised manuscript, we have emphasized the differences between these two free energy-based methods and elaborated on why using both approaches provides a more comprehensive understanding of our time crystals.

Referee 4:

3) *The observations correspond to a simultaneous breaking of the spatial and time translation symmetry. Are the observed spatial symmetry breaking patterns known in previous literature? Are they a requirement for the observation of the temporal ordering?*

I ask because there are classes of time crystals that arise from the presence of different symmetry broken states, with the drive serving to toggle the system between the different stable configurations and thus induce a period doubling behavior. Is this an example of one such time crystal or are the spatial patterns only stable in the presence of the period doubling behavior? If not, are these spatial patterns metastable?

Authors:

We thank the Referee for these helpful questions. We note that similar spatial-temporal patterns have not been observed, to the best of our knowledge. Our system is a common model system, with a positive dielectric constant LC and uniform homeotropic boundary condition, which we then additionally dope with ions. When an electric field is applied, the LC molecules tend to align parallel to the field following the dielectric coupling. Interestingly, we observed not only the spontaneous breaking of spatial symmetry, leading to the formation of lattice patterns under Floquet drives, but also rich periodic dynamics in time, including time-symmetry unbroken phase, period-doubling phenomena, fractional, and disordered response behavior. We find that in both simulations and experiments under a constant external field, the 1D spatially periodic structures can be observed at low voltages (Fig. 3) but become unstable at higher voltages. Additionally, none of the 2D spatial patterns remain stable under a constant field. So we believe these spatio-temporal structures only emerge as the novel space-time crystal behavior due to many-body interactions and inter-transformations of topological defects.

Referee 4:

4) *The authors discuss the presence of different "types" of DSTCs. Is there a precise distinction between the different types of DSTCs? While the visual patterns are slightly different, they appear to have the same symmetry properties.*

Authors:

The spatial spacings of type 1 and type 2 DSTCs are 8.2 μm and 11.6 μm , respectively, in a sample with a cell thickness of 10 μm . For type 3, the spatial spacing is 18.4 μm in a sample with a cell thickness of 15 μm . We observe that type 1 and type 2 DSTCs can co-exist in different regions of the same sample, while type 3 DSTCs are only observed when the cell thickness exceeds 15 μm . We thank the referee for this question and have clarified these distinctions in the revised manuscript.

Referee 4:

1) *Focusing on Fig. 1h, it is actually unclear that the system is exhibiting a period-doubling response--in particular, all the snapshots appear to have the same pattern. Why is that a DSTC?*

Authors:

The pattern in the same reference area is surrounded by four rectangular stripes (two in blue and two in yellow), exhibiting an open-close or close-open like cycle. We thank the Referee for this helpful question. In the revised manuscript, we have re-plotted the figures for the type 2 DSTC to more

clearly demonstrate the period-doubling phenomenon.

Referee 4:

5) What happens for larger T_E in Fig2a? And smaller temperature for 2b?

Authors:

We thank the Referee for this question. In the experiment, the dynamic phase becomes disordered when T_E is increased. However, we did not explore the phase diagram at temperatures lower than room temperature ($\sim 23^\circ\text{C}$), as the liquid crystal 5CB we used may undergo a phase transition from the nematic phase to the solid molecular crystal phase below room temperature.

Referee 4:

6) Looking through the videos, it seems like the patterns described in the paper only emerge for very brief moments, with the system mostly looking homogeneous.

*6.1) Do these patterns occur at the max/min of the voltage or near the zero voltage point?
I think that emphasizing this point in the main text is important.*

Authors:

We thank the Referee for this helpful question. In the revised manuscript, we add a new figure (Fig. 2f) to illustrate when these patterns emerge. Indeed, these patterns appear when the voltage is close to zero. For example, under an external drive with $T_E = 0.5$ s ($U \in [-50$ V, +50 V]), a clear pattern emerges for around 0.04s when $U \sim 0$ V. This is consistent with the results of our modeling (twist structures of birefringent materials can allow the light to pass through under crossed polarizers).

Referee 4:

6.2) How does the system preserve a "memory" of the previous pattern even when looking homogeneous?

Authors:

We thank the Referee for this question, which is also related to the mechanism of period-doubling phenomenon in our systems now described above in responses and in the paper in more details. Utilizing LdG free energy functional based simulations, we reveal that there are inter-transformations, generations and annihilations of the particle-like topological structures of director field within one temporal driving period with all the details. Compared to the structures of the director field one temporal driving period before, the twist angle of the disclinations and topological solitons are the same, with half of the spatial period shift, thus every two driving periods the system adopts the same configuration, resulting in the period-doubling phenomenon. The memory effect stems from the fact that defects and director deformations never fully disappear, just inter-transform. Even when the optical images appear almost homogeneous under a high electric field (the director is perpendicular to the polarizer), the experiments and simulations reveal differences from the uniform background, which directly reveals these memory-related transformations.

We thank the Referee again for these insightful questions. In the revised manuscript, we have elaborated on the origin of the period-doubling phenomenon in our systems, which preserve the “memory” by retaining the topological structures.

Referee 4:

7) *What sets the orientation of the checkerboard pattern in the 2+1D DSTC? Naively, the system has a rotation symmetry around the z-axis, yet the patterns appear to form along the picture axis. Is this some feature of the boundary conditions of the experiments that are not made explicit.*

Authors: The boundary conditions on both substrates are uniform and homeotropic. We did not pre-pattern or pre-coat any structures on the surfaces; the checkerboard patterns emerged spontaneously with spontaneously selected orientations, as shown in Fig. 6a and 6b, which is then retained over time. Regarding the lateral boundary effects, the sample size is approximately 10 mm × 10 mm, while the space-time crystal area is larger than 500 μm × 500 μm and is always at least 1 mm away from the lateral cell edges. The space-time crystal phase patterns are typically surrounded by disordered regions (Fig. 6c). When the voltage and external drive frequency are within the time crystal phase (Fig. 5), we observe that the space-time crystal pattern area grows, as the disordered regions are relatively unstable compared to the space-time crystal patterns. We thank the Referee for this question and have explicitly detailed the boundary conditions in the revised manuscript.

Referee 4:

8) *I was unable to parse the statement "While defect regions tend to disappear in the space coordinates, they still occasionally emerge in higher-dimensional space-time coordinates.". Can the authors clarify the meaning of higher-dimensional in this context?*

Authors: We thank the Referee for this question, which has helped us clarify the description of defects in the manuscript. Here we refer to the defects in the space-time crystal lattice. In spatial coordinates, for example, a lattice defect in the xy-plane may appear at $t = t_0$ and disappear at $t = t_1$. If we consider only the two-dimensional xy-plane, the defect finally vanishes. However, when we include the time coordinate in an xy-t frame (higher-dimensional space-time coordinates compared to the xy frame), the temporal axis (though intrinsically different from spatial axes) behaves similarly to the z-axis. In this extended frame, the defect does not simply "appear" or "disappear"; instead, it emerges at specific positions along the t-coordinate. Since the related discussion is not central to the DSTC behaviours and may cause unnecessary confusion, whereas characterizing it further is outside the scope of this work, we have removed this sentence in the revised manuscript for clarity.

Referee 4:

9) *" Φ is the signals [sic] captured by the quasi-long-range the camera." In this sentence, what constitutes the signal? Naively, there are multiple colors being displayed, so how is that information mapped onto a scalar quantity? Why is such quantity not used when displaying the period doubling behavior in Fig 1k, and instead the position of the middle region is preferred?*

Authors: We thank the Referee for these questions. The optical signals are composed of red, green, and blue components, as our videos are captured in RGB mode over 644 external drive periods. We convert the RGB images to grayscale using ImageJ, and the scalar quantity is obtained from the grayscale video. We have added an image of the time correlation functions for the red, green, and blue signals separately (Supplementary Fig. S4). We find that both the red and blue signals exhibit the same power-law decay behavior, while the green signal is significantly weaker, as evident from Fig. 1d, e, g-i, etc. The time correlation function calculation provides information about temporal correlations when averaged over a large dataset.

In Fig. 1m (Fig. 1k in the previous version), they not only show the positions of the particle-like units over time but also show their motion direction and displacement. While this information could also be revealed by calculating a space-time correlation function (represented as a 3D image with two spatial axes and one temporal axis), in our opinion, such an approach for presenting results is less clear and straightforward compared to providing the trajectories shown in Fig. 1m. Additionally, it would require an even larger dataset to achieve reliable results (recording a high resolution video over 600 drives demands 1 GB of hard drive space).

Referee 4:

10) What is the meaning of the quasi-long-range order in this context? While the system displays a $U(1)$ symmetry (for a rotation of the order parameter around the z axis), the behavior in the time direction has a discrete symmetry, so what causes the power-law decay in time? Is it inherited from the spatial decay of correlations?

Authors:

We thank the Referee for this insightful question. The quasi-long-range order in time is indeed related to the spatial symmetry. For example, in the case of the 1+1D space-time crystal, stripe-like patterns emerge during each drive and recur every two drives. However, the positions of these stripes are not always exactly the same, exhibiting slight randomness due to the nature of the 1D periodic lattice, because of thermal fluctuations (Ref. 38). When we calculate the local temporal correlation function, the randomness in the spatial coordinates translates into the temporal domain. Spatially, we know that the correlation function can exhibit a power-law decay if fluctuations are small. Interestingly, we find that the temporal correlation function also follows a power-law decay, which is inherited from the spatial behavior. Additionally, we note that significant temporal perturbations can destroy this order (Fig. 6e). However, in the time crystal phase with small temporal perturbations (Fig. 6d) or without perturbations (Fig. 9), the temporal order does not decay rapidly. We also note that our work calls for the development of theories that could help deepening the understanding of such time crystalline order, though they are outside the scope of this present study.

Referee 4:

11) What is the FT response of the hexagonal order? What is the ratio between the drive frequency and the response frequency?

It would be very surprising if there there is a non-fractional ratio, yet I couldn't find additional data on this observation.

Authors:

We thank the Referee for this helpful suggestion. In the revised paper, we have added a new figure to exhibit the potential fractional temporal response of the quasi-hexagonal lattice. From the FFT analysis, we find that the frequency $f/f_E \approx 0.3$ ($T/T_E \approx 3.33$). We also calculated the temporal

correlation function and found that the internal temporal periodicity is around $40T_E/12 = 10T_E/3$ ($3.33T_E$) or $33T_E/10(3.3T_E)$, where the temporal periodicity of the external drive is T_E . Interestingly, although our system is very different, a theoretical study on a clean (non-disordered) spin-1/2 system and an experimental study on Rydberg atoms also reported fractional responses around 10/3 (e.g., $f/f_E \approx 0.3$ in Ref. [43] and $f/f_E = 100/29$ in Ref. [44]). We thank the Referee again for this question, which has helped us improve our understanding of the fractional response in the quasi-hexagonal lattice. This system could potentially have the fractional space-time crystal properties of other different fractional responses, which is awaiting for future studies.

Referee 4:

13) How is the "relative noise" quantified in the context of the comment in the discussion? The fact that the system is coupled to a bath at some temperature suggests that there is indeed a large source of noise that is much bigger than the prethermal regime of an isolated quantum system? Or is this a comment with regards to the coherence time of modern quantum platforms?

Authors:

The relative noise we mention here refers to the thermal fluctuations within the system. In Fig.4b, we calculate the interaction energy between neighboring quasiparticles; we find it $\sim 10k_B T$ of a 10% compression and stretching. As a result, intrinsic thermal fluctuations do not significantly affect the structures unless considered over a large spatial scale or a long temporal scale (for example, quasi-long-range order in space and time). We thank the Referee for this helpful question and have provided a clearer description of this point in the revised manuscript.

Referee 4:

14) I'm confused how to compatibilize two claims of the paper: (i) that the DSTC can have day long lifetime (Extended Data Fig 7), and (ii) that you observe power-law decay of the spatial temporal correlations (Extended Data Fig 8). These two observations seem to be at odd since one seems compatible with very stable long-lived two point correlation function, where the later suggests that there is no sharp way of defining a long lifetime for the behavior being explored. Can the authors clarify what they are claiming in this context?

Authors:

In Fig. 8 (Extended Data Fig. 7 in the previous version), the stripe-like features emerge during each drive and recur every two drives. After several hours, we still observe the time crystal behaviours in the same camera-captured region. However, if we calculate the time correlation over time, we find the power-law decay correlation function (Fig. 9). We thank the Referee for this question and make sure to have clarified these points in the revised manuscript.

Referee 4:

15) As a non-expert in the physics of liquid crystal, I was confused by the description of the measurement scheme. Why do different colors get shifted differently by the same pattern of LC directors and how is that related to the average director along the beam path?

Authors:

The sample is illuminated by a wide spectrum of visible light from a lamp. The light first passes through a polarizer, becoming linearly polarized. As it passes through the sample, the light becomes elliptically polarized due to the LC's large birefringence. This birefringence induces a phase

accumulation that depends on both the wavelength of the light and the angle between the LC director orientation and the polarization of the incident light. Finally, the light passes through a second polarizer (analyzer), which is oriented perpendicular to the first polarizer. After passing through the analyzer, the intensity of light at different wavelengths varies, resulting in a superposition of colors that produces the observed color patterns. Such polarized interference analyses are widely used to study liquid crystalline materials. We thank the Referee for this question. In the revised manuscript, we have expanded and detailed the description of the measurement scheme and moved it to the Methods part.

Referee 4:

Comments:

1) I found the lack of discussion of the periodic drive makes the discussion of the DSTC behavior harder to parse. For example, the authors only note the shape of the profile in the figure and never the main text, and do not address how the shape of the profile was chosen and how much it impacts the observations.

While the authors have a mean-field model that is able to capture the observed dynamics, it remains unclear what feature of the system/drive is able to generate the time crystalline behavior. As such it is hard to understand the origin and nature of the observed phenomena.

Authors: In the revised manuscript, we have provided a detailed explanation of the mechanism behind the period-doubling phenomenon, as outlined earlier and in response to Question #2&6. This explanation is supported by both experimental and simulation results. In addition, we have numerically calculated the energy landscape of many-body interactions between neighboring topological solitons (Fig. 4a,b), commonly described as quasiparticles or particle-like objects. We have added additional insets and schematics, making sure that our presentation is clear. We thank the Referee again for these insightful questions and comments, which have greatly enhanced our presentation of the origin of time crystal behavior in our system.

Referee 4:

2) For 1j, the markers might benefit from having two colors that differ more -- i.e. green and orange. This can be combined with different markers for the two parity of periods.

Authors: We thank this helpful suggestion, which we account for.

Referee 4:

3) Because the processing for the "center" of the blue region is somewhat obscure, I might suggest the authors append their data with the value/color of the image as a function of time -- e.g. near the black points of 1g, it is clear that the data fluctuates between blue and purple with a sub-harmonic response.

Authors: We thank this helpful suggestion, in the revised paper we have added the related results in Fig. 1k. We thank Referee 4 again for the thoughtful report and suggestions and comments that helped us to further improve presentation of our findings.

We believe that the above-described revisions address all remarks and account for all suggestions that were raised by the original manuscript. We also feel that they improve the paper significantly. We thank the Referees for the thoughtful valuable comments that directed these changes.

Sincerely,

Prof. Ivan I. Smalyukh, Department of Physics, 390 UCB

University of Colorado at Boulder

Phone: 303-492-7277 (office); Email: Ivan.Smalyukh@colorado.EDU

<http://www.colorado.edu/physics/SmalyukhLab/index.html>

Nature Communications manuscript NCOMMS-24-59238B
**Emergent discrete space-time crystal of Majorana-like quasiparticles in chiral
liquid crystals**

H. Zhao, R. Zhang and I. I. Smalyukh

Summary of Changes and Responses to Requests/Remarks

Changes to Account for the Referees' Suggestions.

Report of Referee 1:

I have carefully reviewed the authors' responses to the referee reports, including their detailed replies to my own comments and concerns. I appreciate the thoroughness with which the authors have addressed the issues raised, particularly regarding the terminology around time crystals, the distinction from classical nonlinear phenomena, and the broader context and implications of their findings.

I agree with the authors' clarifications and the modifications made to the manuscript. The revisions strengthen the presentation and improve the accessibility of the work. I also find the discussions and suggestions from the other referees to be constructive and well-integrated into the authors' revised manuscript.

In light of the authors' comprehensive and satisfactory responses, as well as the work's overall strong scientific merit, I recommend that the manuscript be accepted for publication.

Authors:

We thank the Reviewer #1 for the recommendation and the positive reports.

Report of Referee 2:

The authors responded to all comments raised and thoughtfully explained the missing parts. I can only raise a point regarding the very detailed explanation on the doubling period. I am not an expert in "inter-transformations, generations, and annihilations of coexisting topological solitons and singular disclinations" which seems the founding cause of such phenomena but for a non-expert is really not easy to follow. Could you mitigate the entire added paragraph which I saw transferred 1to1 from the reviewer response to the paper in order to be more open to a non-expert audience?

Authors:

We thank the Reviewer #2 for the remarks and the positive reports. In the revised manuscript, we have modified the paragraphs accordingly (page 6,7) to make the explanation more accessible for non-expert readers.

Report of Referee 3:

Report of Referee 4:

In the reviewed version of the manuscript, the authors have extensively revised their observations, appending two new results: the description of the period doubling dynamics in terms of the creation and annihilation of pairs of topological "Majorana-like" defects, and the observation of a fractional discrete time crystal whereby the sub-harmonic response is not a precise fraction of the original frequency.

While I thank the authors for their efforts in improving the manuscript, I found that they were not able to address some of the questions I raised, and the new results beg a few new points that I think the authors should address before the manuscript should be published.

In general, I have found that the paper is exploring an interesting new regime and demonstrates a novel platform for studying novel out of equilibrium phenomena. However, I have found that the relevance of the work is diminished because the paper fails to engage with the broader set conceptual and categorization questions that come from the observation of time crystalline phenomena, specially in open classical systems. Without such discussion, some of the claims of the paper feel unsubstantiated and unclear. I think that to appeal to the broad audience of Nature Comms, such discussions are relevant, otherwise I agree with Referee 1 that the manuscript is more suitable to a more specialized manuscript.

In an attempt to help the authors, I have divided my comments into main concerns and minor concerns, as per the scope my comment.

Authors: We appreciate the reviewer for this very thoughtful and detailed report saying that “*the paper is exploring an interesting new regime and demonstrates a novel platform for studying novel out of equilibrium phenomena*”. We also thank the reviewer for the helpful suggestions, which we have carefully addressed in the revised manuscript and detailed in our responses below.

Reviewer #4:

Main Concerns:

1) Categorization of the time crystalline phenomena and nature of the ordered phase

As also alluded by Referee 1, the nature of period doubling (and other time crystalline phenomena) in classical systems is subtle. That is because the role of the external bath, as well as the type and interactions between the different degrees of freedom leads to fundamental different expectations about the nature and the stability of the behavior. For example, in few particle systems, near-integrability can enable the stabilization of long-lived period doubling behavior via the KAM theorem; by contrast, when coupling to a bath that can extract entropy without providing

fluctuations, the system can be toggled between different stable sectors in a trivial manner. In both these scenarios, the life-time of the DTC behavior is infinitely long-lived owing to the details of the system.

The present work does not fall into these conditions: it is a strongly open system (the authors must include a meaningful damping rate on their simulations) at a very non-zero temperature, and it treats the dynamics of an extensive number of interacting particles.

As a result, the expectation is much more subtle: the interplay of noise and interactions is expected to destroy the time crystalline order, unless the particular model has some novel mechanism for stabilizing the expected proliferation of defects and their destruction of the time crystalline phase. In this case, the lifetime of the time crystalline phenomena is finite, albeit very long.

(These considerations are extensively explored in arXiv:2305.08904, a paper that the authors reference but do not engage too strongly with).

I understand that the current manuscript is not meant to revisit the entire understanding of the time crystalline phenomena, however, I find that the lack of any discussion of these considerations (in particular the lifetime and what determines it), specially where they authors understand their observations to lie, severely limits ones capability to parse their observations and to contextualize their results. As a result, the authors appear to emphasize the discovery of a completely distinct time crystalline phenomena, but without having strong enough evidence to make such a point.

For the clarity of this work and its contextualization with the broader time crystalline community, I think it is important for the authors to directly engage with the question of stability and heating within the referee report and the manuscript.

Authors: We thank the reviewer for these helpful questions and remarks. In the revised manuscript, we have added a discussion on the lifetime of the time crystal in relation to thermal noise. Our numerical simulations show that noise strongly affects the lifetime, as expected, which is consistent with our experimental observations. To numerically quantify this effect, we introduce a white-noise stochastic perturbation $f_{\text{noise}}(t)$ that directly couples to the director, with the auto-correlation $\langle f_{\text{noise}}(t)f_{\text{noise}}(t') \rangle = 2\eta_0 T_{\text{eff}} \delta(t-t')$, where η_0 is the friction coefficient and T_{eff} is the effective temperature of the thermal perturbation. In the simulations, as we increase the effective temperature, we observe that the order of the system decreases, with more defects emerging in the LC bulk (Supplementary Fig. S9a). We then measure the lifetime τ_0 (correlation time), calculated from the Fourier transform spectra (as detailed in Ref. [43]), and find a dramatic decrease in lifetime as the noise increases (Supplementary Fig. S9b). Interestingly, what we find is that $\tau_0 \sim e^{\Delta/k_b T_{\text{eff}}}$, where Δ is the quasi-energy activation barrier as defined in Ref. [9] (arXiv:2305.08904), indicating a (thermally) “activated” time crystal. Such a dramatic transition is consistent with our experimental results. Indeed, in experiments, it is difficult to precisely control the noise level (for example, temporal perturbation); as we vary the perturbation, the system behavior changes dramatically from a time crystal state (which can typically persist for an extremely long time) to a disordered state (which shows a very short or nearly zero lifetime, Fig. 6d,e). In addition, inspired by this idea, we revisit the analysis of temporal correlation function (Fig. 9) of the 1+1D DSTC and find that $G(t)$ could be fitted by e^{-t/τ_0} (Chapter V.A in Ref [9]). We thank the reviewer again for these helpful remarks, which help us to categorize our findings.

Reviewer #4:

With that in mind, I still have some conceptual questions about the nature of their observations.

1.1) Throughout the paper, the authors utilize the center location of the blue domain to build the two-point correlation function that they use to state the observation of quasi-long-range order. I am still confused about this choice. In the book that they reference, power-law correlations are given in terms of the order parameter of the smectic transition (Chapter 6 if I am not mistaken), and they see power-law in spatial correlations with a power that is controlled by the parameters of the system (namely temperature).

The authors claim to see the power-law behavior in the two-time (albeit the evidence is not very strong given the small dynamical range of their fit), using a correlation function over the position center. What is the relationship between the spatial and the time fluctuations? What is the expected form of the power-law exponent in this case? And what does that inform us about the stability of the time crystalline phenomena? And what prevents the authors from considering data for thousands of time steps since they observe the robustness over hours and each period is second scale?

Authors:

We thank the reviewer for these insightful questions. First, to clearly describe how we calculate the time correlation function, in the revised paper, we have provided a more detailed explanation in the Method section. For the correlation functions in Fig. 9 and Fig. 10e, we did not calculate them utilizing the center locations of the blue domain. Instead, the time correlation function $G(t)$ is directly calculated from the video signal $\Phi_{i,j}(\tau)$ (without any center location tracking). Here, $\Phi_{i,j}(\tau)$ denotes the signal of the pixel at column i , row j at time τ , and $G(t) \sim \sum_{\tau} \sum_{i,j} \Phi_{i,j}(\tau) \Phi_{i,j}(\tau + t)$. We also calculate the spatial correlation function $G(r) \sim \sum_{\tau} \sum_{i,j} \Phi_{i,j+r}(\tau) \Phi_{i,j}(\tau)$ and spatial-temporal correlation function $G(r,t) \sim \sum_{\tau} \sum_{i,j} \Phi_{i,j+r}(\tau) \Phi_{i,j}(\tau + t)$ (Supplementary Fig. S4,8), where j and r are along the direction of spatial spacing, and i is perpendicular to the spatial spacing for 1+1D DSTC. In smectic liquid crystal systems with 1D quasi-long-range positional order, the power-law exponent is related to the inter-molecular interactions and orientational elasticity coefficients and temperature (Ref. [43]), and the emergence of defects and lattice imperfections can strongly affect it. The emergent system we study shows similar spatial correlations (in fact, this is expected as chiral nematic's elasticity is often mapped to that of smectic LCs as part of coarse-graining - Ref. [43,44]). We note that $G(r)$ can be fitted by a power-law decay in the scale of $\sim 100 \mu\text{m}$ (Supplementary Fig. S4). For the stability, we find that for larger-scale DSTC containing impurity regions, the $G(t)$ and $G(r,t)$ decay faster than in well-ordered space-time crystal without impurities (Supplementary Fig. S6-8), which informs that the stability is determined by the noise (as described in the previous comment) and lattice impurities. Regarding the question of time steps, we have added related results and minor revisions related to methodology in the revised manuscript (Supplementary Fig. S6-8).

Reviewer #4:

1.2) In the rebuttal, the authors emphasize that the file size for the data is very large, but it seems like they have access to the videos. Is this a statement that the data analysis is very challenging or a statement that they were unable to store videos for all the experiments they conducted and so they had to only store this subset of the information?

Regardless, a discussion about how the middle of the blue regions is obtained should be more prominent as it constitutes a non-trivial piece of data analysis that is important to parse the observations, as well as the correlation function discussed later.

Authors:

We thank the reviewer for these questions and comments. To obtain a long continuous video recording of high resolution (Supplementary Fig. S6-8), the experiment demands ~ 2 GB of hard drive space for 1000 drives, and it is difficult to record an even longer video for the software while precluding various drifts (even though the experiment is done on an optical table). Since this is not the focus of the current study, additional dedicated experiments may be needed in future to test theoretical expectations/models. In the Method section, we explicitly demonstrate how we track the center position for Fig. 1j,l,m and Fig. 8b using a freeware ImageJ (National Institutes of Health), and how we calculate the correlation function directly using the video signal (without tracking) for Figs. 9, 10e, and Supplementary Fig. S4, 5, 7,8. We thank the reviewer again for helping us improve the clarity of description of these methods.

Reviewer #4:

2) Role of the "Majorana-like quasiparticles"

One of the main changes done to the manuscript is the inclusion of a discussion of the period doubling behavior in terms of the dynamics of topological defects within the liquid crystal. These defects, which take half-integer value assuming that the orientation of the liquid crystal is within the 2d plane, are actually their own anti-particles, owing to the rotation freedom of the liquid crystal molecules in 3d space. The authors can then describe the observation of the time crystalline behavior by the creation and annihilation of such pairs of defects along the vertical direction, shifted by $L/2$ every period of the drive.

2.1) I am a bit confused about the meaning of Fig 4 and the dynamics being depicted. In figs g-i, it seems like the dynamics being depicted is the transmutation of the $+1/2$ into $-1/2$ defects and vice versa, rather than their annihilation and creation. Indeed, from the discussion it appears that the notion of $\pm 1/2$ defect is only valid in the large $|U|$ limit where the liquid crystal molecules' orientation is restricted to the XZ plane. In what way is that language (of $\pm 1/2$ defects) meaningful in this context?

Authors:

We thank the reviewer for these remarks. In the 2D cross-sectional plane the defect's (disclination's) winding number can be related to the so-called twist angle β : $\beta = 0$ corresponds to a $+1/2$ defect and $\beta = \pi$ corresponds to a $-1/2$ defect; the definition of these winding numbers in 2D is provided in text. In 3D samples, these winding numbers are not invariants characterizing distinct topology but just geometric quantifiers of structures. A general defect can transform between the $+1/2$ and a $-1/2$ states via tuning β (Ref. [38]). In the revised manuscript, we add the results of β in Fig. 4j: most of the time they are $+1/2$ and $-1/2$ geometric embodiments of topological defects since the voltage is high enough, and the inter-transformation of the half integer defects exhibit analogues to the inter-transformation of Majorana quasiparticles, which are described in detail in the following comments.

Reviewer #4:

2.2) It appears that the dynamics are divided into two moments; a very fast one $0 < t < 0.02T_e$, around which the topological defects bounce a lot between the top and the bottom of the system after the quench, and then an entirely different setting where their motion follows smoothly with the electric field intensity, albeit shift by $L/2$. What guarantees that the oscillations observed in the first few moments die down, and that the defects merge? What determines the position of the formation of the defects as the electric field is ramped up?

Authors: We thank the reviewers for these questions and comments. Generally speaking, the different dynamics of the defects are related to the temporal gradient (slope in the U vs. time plot) of the external voltage. During the dynamic process, except when the disclinations (defect) annihilate and re-generate, the $+1/2$ disclinations are close to the bottom surfaces when $U < 0$. If the voltage is smoothly changing across $U = 0$, the disclinations transformation follows the scenario from Fig. 4g to 4i. However, as the voltage suddenly changes from U_{\max} to $-U_{\max}$ at $t = T_E, 2T_E, \dots$, the director field cannot suddenly deform correspondingly, inducing disclinations annihilation rather than smoothly transformation at the same x positions. During the annihilation process, the director field keeps the “memory” of the periodic deformations, the neighboring $L/2$ positions are more favorable (strong localized deformations and low scalar order parameter S region) for the re-generation of new pairs of disclinations compared to other positions (spontaneously in simulation and experiments).

Reviewer #4:

One reading of the observed dynamics is that the system has two stable configurations, corresponding to two different lattices of domain walls, and the drive is such that the system is moved between the two and the stability is guaranteed by the stability of the lattice of defects. Is this picture accurate? If so what is added by the discussion of the microscopic dynamics of the "Majorana-like" defects?

Authors: We thank the reviewer for providing a new perspective on the mechanism. The interesting feature of our system is that the building blocks are topological quasiparticles that exhibit intriguing inter-transformations within each time period. There may be more than two dynamically stable configurations in our system. For example, the free energies of the soliton array are the same for a 1D lattice as its orientation rotates by any angle along z (energy degeneracy of the lattice orientation due to the homeotropic boundary condition). Nevertheless, we did not observe a transformation from the original orientation to other angles. The local microscopic transformation is very important for clearly demonstrating the dynamics of our systems: not only because the half-integer disclinations are analogous to the Majorana-like quasiparticles and its antiparticles, but also because the smooth transformations as spinors from $+1/2$ disclinations to $-1/2$ disclinations follows the Majorana equation (Ref. [27]). This concept suggests that the time crystalline order observed in our liquid crystal systems is topologically interesting and, we believe, distinct from other literature.

Reviewer #4:

2.3) Given these questions (and the previous point about the characterization of the time crystalline behavior), I find that the revised title of the manuscript is not entirely clear. Can the authors clarify what they mean by "discrete space-time crystal of Majorana-like quasiparticles"?

Authors:

We thank the reviewers for this question. In our systems, the Majorana-like quasiparticles play pivotal roles: in the spatial coordinate, these Majorana-like quasiparticles and their antiparticles self-assemble to the lattice; in the temporal coordinate, the smooth transformation of the Majorana-like quasiparticles follow the Majorana equation; these spatial-temporal elements form the space-time crystal.

Reviewer #4:

2.4) Finally, I am a bit confused about the (apparent) inconsistency between Fig 3 and Fig 4. In Fig 3, the steady state obtained by the mean field method is such that the defect structure is alternating ($+1/2, -1/2, +1/2, \dots$), whereas in Fig 4, the system now exhibits a fixed pattern of topological defects

(+1/2, +1/2, +1/2). What is the difference between the two settings? Given that the authors claimed that the mean field accurately captures the dynamics, I am confused by the observations.

Authors:

The structures in Fig.3 are the director field configurations without applying voltage, they are the initial configurations that are used to obtain the dynamic process shown in Fig. 4. In addition, these alternating $+\pi$ - and $-\pi$ -rotation Néel domain walls (topological solitons) in Fig. 3 locate at the x - y plane, which are not the $+1/2$ and $-1/2$ disclinations (singular defects) and domain walls (Néel or Bloch domain wall depended on the external voltage) embedded within the x - z plane. We have slightly revised the text description and assured that this is clear to readers and thank the reviewer for this question, we revised corresponding parts to clearly show their differences and topological nature.

Reviewer #4:

3) Observation of the Fractional DTC physics

The other novel result is the observation of the fractional discrete time crystal. Looking through Fig 10, it seems like the system returns to its initial configuration after 3 periods, and thus should exhibit a 1/3 sub-harmonic response. The shift in the correlation function appears to occur due to the presence of a non-negligible number of defects in the hexagonal lattice that is being prepared. The assertion that the system is at a fraction p/q suggests that the system performs p orbits over q periods -- this is not supported by the observations, since the response is clearly affected by the presence of defects.

Unless the authors have additional evidence that they have not included in the current version of the manuscript, I am unconvinced of the claim that they have a "fractional discrete time crystal". Indeed, returning to the point of the characterization of the behavior using the center of the blue region, I find that the authors' analysis further complicates one's ability to parse the significant of this response.

Authors:

We thank the Reviewer #4 for these comments. Although the POM images with $\Delta t=3T_E$ look similar, they are not identical. In the revised manuscript, we added a new supplementary figure (Supplementary Fig. S10) with $\Delta t=9T_E$: if the system has a rigorous $1/3$ sub-harmonic response, they should not show such a difference after 9 periods (no defect emerging during Δt), which indicates that the system may not have a period-tripling phenomenon. In addition, as mentioned in previous questions, the Fourier transform spectrum and time correlation function are obtained directly from the signals of video pixels, both of them do not exhibit a clear integer response.

We agree that due to the presence of a non-negligible number of defects, it is difficult to unambiguously conclude whether they are a fractional discrete time crystal. In the revised manuscript, we tune down the statements and emphasize that the quasi-hexagonal case potentially could be a candidate of fractional discrete time crystals, which awaits further studies.

Reviewer #4:

Minor Comments:

1) In the rebuttal, the authors claim. "After several hours, we still observe the time crystal behaviours in the same camera-captured region." Can they make precise what they mean? Locally (in time) the

system is still toggling between two configurations? Or there is some form of locking with the initial oscillation in the system? What does this mean about the stability of the time crystalline phenomena?

Authors:

We thank the Reviewer #4 for these questions. In the experiments, we apply the external drive for hours to one day (continuously switching on), and monitor the camera-captured region, the system can exhibit the period-doubling phenomenon locally in time as shown in Fig. 8a and Supplementary Video S9. However, it is difficult to claim that these period-doubling observed after several hours is locked to the very first period-doubling, since sometimes small impurity/defect regions emerge (Fig. 7a). Nevertheless, the impurity/defect regions disappear after tens of drives, and the system recurs to period-doubling behaviors. This indicates that our time crystal is very stable against small spatial-temporal fluctuations.

Reviewer #4:

2) In the rebuttal, the only time the authors engage with the concern about the analysis of the data is in the addition of the oscillation of the amount of blue within an arbitrarily chosen square of the image. It is not explained why this is the correct metric (the blue channel or the chosen region), nor how it relates to the metric being analyzed. I find that the authors engaged with the letter of my previous comment but didn't engage with the point that I was raising, namely that the data analysis pipeline is complex so it is hard to parse the importance of the correlation function

Authors: We thank Reviewer #4 for these questions and previous suggestions. In Fig. 1k, as suggested by reviewer #4 in the previous report, the blue signal shows a clear period-doubling phenomenon, where the vertical axis represents the average blue signal from the region marked in Fig. 1j. This selected region cannot be too large or too small, if the size is too large (for example, the full image of Fig. 1j), it's hard to verify the period doubling only from the blue signals; if the marked region size is too small, the signal fluctuation will be larger. Therefore, to clearly show the period-doubling behavior and how the signals auto-correlates in time, we calculate the time correlation function of each pixel from the full images over time (Fig. 9), and then sum the results from all pixels, as explicitly described in the Method section and in the previous comments. We thank the reviewer again for helping us improve the clarity of the analysis, and inspiring us to exhibit the period-doubling phenomenon in different ways.

Reviewer #4:

*3) In the rebuttal, the authors note the experimental settings when asked about the "types" of DSTCs. I am confused about this response since the authors continue to emphasize that they observe different "types" of DSTCs, but cannot highlight what is the sharp feature that distinguishes them. It seems like all exhibit some square pattern of correlations that shifts between different periods of the drive. While the shape of the pattern is different, the symmetry properties seem to be the same, they all live in a square lattice. Is there something that I am missing? How do the authors define the different types?
Are these patterns captured by the mean-field numerics?*

Authors: We thank Reviewer #4 for these questions and comments, we agree that all of these patterns are square lattice and have same symmetry properties, the differences of patterns indicate that they have different director field configurations. To clearly indicate the difference and avoid the misunderstanding, in the revised manuscript we labelled them as configurations 1-3. Regarding for the numerical simulations, to reconstruct the space-time configuration of 1+1D DSTC, the simulation box size of the director field is 300×150 , and the time step is 5000 within one temporal

period (the simulations contain over 100 temporal periods). To capture the 2+1D DSTC dynamics using the tensor-based simulation, the box size should be larger than $300 \times 300 \times 150$, which is beyond our simulation ability. Although the quasi-static structures (Fig. 3k-o) from vector-based simulation shows similarity of 2+1D DSTC patterns, we cannot reconstruct a whole spatial-temporal dynamic from it.

Reviewer #4:

4) The authors now emphasize the defect perspective on the dynamics of the 1+1D DSTC. Does something apply to the 2+1D case? If not why? What about the hexagonal case?

Authors:

We thank the reviewers for these insightful questions. For the 1+1D DSTC, we find that the dynamic is related to the Majorana-like quasiparticles and domain wall solitons (1D topological solitons), which are labelled as first homotopy group $\pi_1(S^2/\mathbb{Z}_2) = \mathbb{Z}_2$ (the transformation follows the 1+1D Majorana equation). For 2+1D, the basic units also consist of pairs of $\pi_2(S^2/\mathbb{Z}_2) = \mathbb{Z}$ topological defects and skyrmion tubes (2D topological solitons), which are labelled as second homotopy group. In liquid crystal systems, such an assembly is called toron [31,57], which exhibits similar optical patterns compared to Fig. 1. However, these phenomena are rather complex and more disciplinary, with the power of numerical modelling still insufficient to capture details that could be presented in an accessible form. We therefore leave this aspect to future more specialized articles.

We believe that the above-described revisions address all remarks and account for all suggestions that were raised by the original manuscript. We also feel that they improve the paper significantly. We thank the Referees for the thoughtful valuable comments that directed these changes.

Sincerely,

Prof. Ivan I. Smalyukh, Department of Physics, 390 UCB

University of Colorado at Boulder

Phone: 303-492-7277 (office); Email: Ivan.Smalyukh@colorado.EDU

<http://www.colorado.edu/physics/SmalyukhLab/index.html>

[revised manuscript text omitted]

quasiparticles and the period-doubling phenomenon (Fig. 4a-f and Supplementary Video 2)³⁵.
This is directly exhibited by the snapshots of the director field and scalar order parameter
distribution (Fig. 4c,f). To understand the underlying mechanism, we thoroughly analyse the
dynamic process, finding the inter-transformations, generations, and annihilations of coexisting
topological solitons and singular disclinations (Fig. 4g-o). The singular disclination structures, as
well as the fragments of solitonic nonsingular walls between them, are the building blocks of the
1+1D time crystal, which can be treated as Majorana-like quasiparticles and respective anti-

particles (Fig. 4k)^{26–28}, because the studied disclination profiles in their cross-sections smoothly
transform as spinors following the Majorana equation (see Ref. ²⁷ for details).

In the time crystal phase, the particle-antiparticle inter-transformation occurs when the
external voltage U smoothly changes from negative to positive (Fig. 4g-i). The director field
around the disclination region can be characterized by the characteristic twist angle β ($\beta \in [0,$
$\pi]$)^{36–38}, whereas the core (centre) of the disclination line is a topological singularity, as the
director orientations cannot be defined accordingly. When $U < 0$, β equals to π at the top and 0 at
the bottom, indicating wedge disclinations³⁹ with winding numbers $-1/2$ and $+1/2$ defined in the
two-dimensional cross-sections orthogonal to the defect lines (Fig. 4j), respectively (note that the
$-1/2$ and $+1/2$ structures are topologically the same in 3D as they can morph one to another).
This winding number $\pm 1/2$ of disclinations relates to the accumulated angle of $\mathbf{n}(\mathbf{r})$ rotation as
one circumnavigates the disclination core once, divided by 360° , with the positive sign
corresponding to the counterclockwise rotation matching that of circumnavigation and negative
sign referring to the opposite clockwise case. Between the two disclination regions, the $-\pi$ -
rotation Néel domain wall solitons (with $+/-$ defined by counterclockwise/clockwise rotation
when traversing the topological soliton from left to right) are spatially embedded (Fig. 4g). The
domain wall solitons are labelled as elements of the first homotopy group ⁴⁰, $\pi_1(\mathbb{S}^2/\mathbb{Z}_2) = \mathbb{Z}_2$,
and allowed to smoothly inter-transform to have 2D cross-sections of different types while here
being terminated by two disclinations $\pi_1(\mathbb{S}^2/\mathbb{Z}_2) = \mathbb{Z}_2$ also inter-transforming
[revised manuscript text omitted]

**Fig. 3 | Quasi-static initial director structures.** a-d, Experimental (a,b) and numerically
simulated (c,d) POM images of the translationally invariant $\mathbf{n}(\mathbf{r})$. e,f, $\mathbf{n}(\mathbf{r})$ in mid-planes (x - y planes)
of regions marked in (c) and (d), respectively, which can be interpreted as containing a pair of $-\pi$
and $+\pi$ elementary domain-wall solitons, as marked. g,h, $\mathbf{n}(\mathbf{r})$ in vertical x - z planes marked in (e)
and (f), respectively, shown in an inset of (e) with cylinders coloured according to the order
parameter manifold, the sphere with diametrically opposite points identified. i,j, Experimental (i)
and numerically simulated (j) three-photon excitation fluorescence polarizing microscopy
images³³ of two repeat units of a structure shown in (g), where polarizations of excitation light are
marked by blue arrows. k,l, Experimental (k) and numerically simulated (l) POMs of the 2+1D
DSTC (configuration 2). m-o, $\mathbf{n}(\mathbf{r})$ (m) in mid-plane corresponding to the marked region in (l)
and in x - z (n) and y - z (o) cross-sectional planes marked in (m). Scale bars are 10 μm in (a-d,k and
13 l) and 5 μm in (i,j). Transmitting axes of polarizer and analyser are marked by black double arrows;
the slow axis of a retardation plate is marked by a green double arrow.

**Fig. 4 | Majorana quasiparticles and period-doubling in DSTCs.** **a**, Numerically simulated
director field of 1+1D DSTC based on the Landau-de Gennes free energy functional, the
background is coloured by the scalar order parameter S (right-side inset). The bottom inset
schematically shows the many-body interactions among neighbouring building blocks, which
consist of topological solitons and disclinations. **b**, Interaction energy per unit length (translation
invariant along y) versus displacement relative to the equilibrium length (along x). **c**. Time-
dependent director field showing the period-doubling phenomenon, with the selected region

marked in (a). **d,e**, Enhanced contrast experimental (top) and numerically simulated (bottom)
 POM images under a small (d) and large (e) electric field within one T_E . **f**. Snapshots of S
 distribution at different times within the selected region marked in (a). Black dashed lines in
 fixed positions are shown for reference. **g-i**, Snapshots of the director field (left) showing the
 topological transformation, angle β of the disclination and the topological solitons in between
 vary smoothly when the voltage is close to zero. The corresponding schematics revealing their
 topological nature are shown on the right. Black circles show disclination cores; black dashed
 rings characterize director on closed loops. **j**, External drive and twist angle β of the disclination
 near the bottom surface within two external drive periods, T_E . In the second T_E , the structures are
 shifted by half of the spatial period. **k**, Director field profiles showing the transition from $-1/2$ to
 $+1/2$ configurations of disclinations, which can be viewed as a Majorana-like quasiparticle and
 its antiparticle. The twist angle $\beta = \cos^{-1}(\boldsymbol{\tau}_v \cdot \boldsymbol{\Omega})$, where $\boldsymbol{\tau}_v$ (blue arrow) is the tangent vector of the
 disclination (black cylinder) and $\boldsymbol{\Omega}$ is the rotation vector (red arrow). **l**, Snapshots of the director
 field showing the transition after the voltage suddenly changes from $+U_{\max}$ to $-U_{\max}$ (left),
 followed by annihilation (middle) and generation (right) processes. **m-o**, Snapshots of the
 director field during the second T_E (j), when topological solitons and disclinations shift by $L/2$
 compared to (g-i), respectively. Black dashed lines in (g-i,m-o) show relative positions for
 reference.

1

2

3 **Fig. 5 | Phase diagrams containing DSTCs. a,** Phase diagram in coordinates of T_E and the voltage4 amplitude U_{max} for a sample of cell thickness $d = 5 \mu\text{m}$ at room temperature. **b,** Phase diagram as5 a function of the temperature and U_{max} for the same sample as in (a) for $T_E = 0.5\text{s}$. **c,** Phase diagram6 as a function of T_E and U_{max} for $d = 10 \mu\text{m}$ at room temperature. **d,** Phase diagram as a function of7 temperature and U_{max} for the same sample as in (c) for $T_E = 0.5\text{s}$. **e-i,** POM snapshots of the time-

8 symmetry-unbroken phase (e), disordered phase (f), 1+1D DSTC phase (g), 2+1D DSTC phase (h)

9 and co-existence phase (i), respectively. The helicoidal pitch is $p = 5 \mu\text{m}$. Scale bars indicate $5 \mu\text{m}$.

10 Transmitting axes of polarizer and analyser are marked by black double arrows; the slow axis of a

11 retardation plate is marked by a green double arrow.

**Fig. 6 | Formation of DSTCs and their rigidity against temporal perturbations.** **a**, Space-time
image captured for the same time interval T_E within a spatial stripe-like region showing the 1+1D
DSTC “boil out”¹⁸ from a disordered state. **b**, POM snapshots showing the 2+1D DSTC “boil out”
from a disordered state. The elapsed time is marked on the panels. **c**, POM snapshots showing the
nucleation and growth of the DSTC region. The time interval of the two snapshots is 20 s (40
drives). **d,e**, POM snapshot (left) and space-time plot (right) of a 1+1D DSTC (d) and a disordered
state (e) with temporal perturbation ΔT_E randomly distribute within $[-0.2\bar{T}_E, 0.2\bar{T}_E]$ (d) and $[-0.4\bar{T}_E,$

1 $0.4\bar{T}_E]$ (e), where the average external temporal periodicity $\bar{T}_E = 0.5\text{s}$. The tracked region of space-
2 time plot is marked on the left, where the $0.1\bar{T}_E$ time interval of each snapshot step. Scale bars are
3 $10\ \mu\text{m}$ in (a), $5\ \mu\text{m}$ in (b) and $20\ \mu\text{m}$ in (c-e).
4

1

2

[revised manuscript text omitted]

 DSTC (configuration 2) spatial configurations spontaneously emerge (Fig. 3m-o) in each drive.
 The spatial discretization is performed on large 3D square-periodic $80 \times 80 \times 40$ grids, and the
 spatial derivatives are calculated using finite-difference methods with the second-order accuracy.
 For all simulations, the following parameters are used if not specified: $d/p = 2$, $K_{11} =$
 $6.4 \times 10^{-12} \text{ N}$, $K_{22} = 3 \times 10^{-12} \text{ N}$, $K_{33} = 10 \times 10^{-12} \text{ N}$, $W = 10^{-4} \text{ J m}^{-2}$ and $\gamma = 77 \text{ mPas}$.
 While this approach captures many fine features of quasi-static configurations seen in
 experiments for some parts of phase diagrams, it does not yield the dynamic features of the time-
 crystalline structures, this is because the numerical simulation based on Frank-Oseen free energy
 cannot simulate disclinations or low scalar order parameter regions^{35,40}, for which we adopt a
 different approach described below.

**Numerical modeling based on the Landau–de Gennes free energy functional**

The quasi-static spatial structures of 1+1D DSTC (Fig. 3e-h) can also be revealed by the
 Landau–de Gennes free energy functional, in which flexoelectric terms and ion-induced electric

1 field are incorporated. The tensor order parameter is defined as $\mathbf{Q} = S(\mathbf{nn} - \mathbf{I}/3)$, where S is the
 2 LC's scalar order parameter and \mathbf{I} is the identity matrix, and the energy cost of the spatial
 3 deformations of $\mathbf{Q}(\mathbf{r})$ can be expressed as:

$$4 \quad F_{\text{elastic}}^{\text{LdG}} = \int d^3\mathbf{r} \left\{ \frac{L_1}{2} \frac{\partial Q_{ij}}{\partial x_k} \frac{\partial Q_{ij}}{\partial x_k} + \frac{L_2}{2} \frac{\partial Q_{ij}}{\partial x_j} \frac{\partial Q_{ik}}{\partial x_k} + \frac{L_3}{2} \frac{\partial Q_{ij}}{\partial x_k} \frac{\partial Q_{ik}}{\partial x_j} + \frac{L_6}{2} Q_{ij} \frac{\partial Q_{kl}}{\partial x_i} \frac{\partial Q_{kl}}{\partial x_j} \right. \\ 5 \quad \left. + \frac{4\pi}{p} L_4 \varepsilon_{ikl} Q_{ij} \frac{\partial Q_{lj}}{\partial x_k} \right\}, \quad (4)$$

where ε_{ikl} is the Levi-Civita symbol and L_i 's are the elasticity parameters. In addition, the
 Landau–de Gennes free energy functional includes thermotropic terms that describe the nematic-
 isotropic transition of the LC:

$$9 \quad F_{\text{thermotropic}}^{\text{LdG}} = \int d^3\mathbf{r} \left\{ \frac{A}{2} \left(1 - \frac{U_{\text{LdG}}}{3} \right) Q_{ij} Q_{ji} - \frac{AU_{\text{LdG}}}{3} Q_{ij} Q_{jk} Q_{ki} + \frac{AU_{\text{LdG}}}{4} (Q_{ij} Q_{ji})^2 \right\}, \quad (5)$$

where A and U_{LdG} are the nematic material parameters. When an external electric field $\mathbf{E}_{\text{external}}$ is
 applied, the total electric field \mathbf{E} in the LC is a superposition of $\mathbf{E}_{\text{external}}$ and \mathbf{E}_{ion} , where \mathbf{E}_{ion} is the
 ion-induced electric field, and can be calculated by the Poisson equation²⁹:

$$13 \quad \nabla \cdot (\boldsymbol{\varepsilon} \cdot \mathbf{E} + \mathbf{P}_{\text{flexo}}) = \rho_{\text{el}}, \quad (6)$$

where $\boldsymbol{\varepsilon}$ is dielectric tensor, $\mathbf{P}_{\text{flexo}}$ is polarization field due to flexoelectric, and ρ_{el} is the ionic
 charge satisfying $\nabla \cdot (\boldsymbol{\sigma} \cdot \mathbf{E}) = -\partial_t \rho_{\text{el}}$, with $\boldsymbol{\sigma}$ being the conductivity tensor. The dielectric and
 conductivity tensor are related to the Q-tensor via $\boldsymbol{\varepsilon} = \bar{\varepsilon} \mathbf{I} + \varepsilon_a^{\text{mol}} \mathbf{Q}$ and $\boldsymbol{\sigma} = \bar{\sigma} \mathbf{I} + \sigma_a \mathbf{Q}$, where $\bar{\varepsilon}$
 and $\bar{\sigma}$ are the mean dielectric and conductivity constants, respectively, and $\varepsilon_a^{\text{mol}}$ and σ_a are the
 dielectric anisotropy and conductivity anisotropy, respectively.

The free energy is supplemented by the following electric coupling terms:

$$20 \quad F_{\text{electric}}^{\text{LdG}} = \int d^3\mathbf{r} \left\{ -\frac{1}{2} \varepsilon_0 \bar{\varepsilon} E_i^2 - \frac{1}{3} \varepsilon_0 \varepsilon_a^{\text{mol}} Q_{ij} E_i E_j + \zeta_1 \frac{\partial Q_{ij}}{\partial x_j} E_i + \zeta_2 Q_{ij} \frac{\partial Q_{jk}}{\partial x_k} E_i \right\}, \quad (7)$$

where ζ_i 's are the flexoelectric constants. The first two terms describe dielectric coupling between
 \mathbf{Q} and the electric field \mathbf{E} , and the last two terms describe the flexoelectric effect.
 The surface free energy describing the surface anchoring at the substrates reads

$$4 \quad F_{\text{surface}}^{\text{LdG}} = - \int d^2\mathbf{r} \frac{W}{2} (Q_{ij} - Q_{ij}^{(0)})^2, \quad (8)$$

where $Q_{ij}^{(0)}$ defines the preferred orientation and order of LC at the surfaces, corresponding to
 perpendicular boundary conditions in our experiments. The total Landau–de Gennes free energy
 $F = F_{\text{elastic}}^{\text{LdG}} + F_{\text{thermotropic}}^{\text{LdG}} + F_{\text{electric}}^{\text{LdG}} + F_{\text{surface}}^{\text{LdG}}$. The evolution of the system follows the
 Ginzburg–Landau equation³⁵:

$$9 \quad \partial_t \mathbf{Q} = -\Gamma \left[\frac{\delta F}{\delta \mathbf{Q}} \right]^{st},$$

where $[...]^{st}$ is a symmetric and traceless operator and the relaxation coefficient Γ is determined
 by the rotational viscosity γ_1 via $\Gamma = 2S_0^2/\gamma_1$ with S_0 being the constant equilibrium bulk order
 parameter ($S_0 = \frac{1}{4} + \frac{3}{4} \sqrt{1 - \frac{8}{3U_{\text{LdG}}}}$). Note that the above free energy is time-dependent because of
 the time-varying external electric field $\mathbf{E}_{\text{external}}$.

[revised manuscript text omitted]

Supplementary Video Captions

Supplementary Video 1 | Videos showing the 1+1D and 2+1D DSTCs. The POM video for 1+1D DSTC (top left) is obtained for a cell gap $d = 5 \mu\text{m}$, the POM videos for configuration 1 (top right) and configuration 2 (bottom left) 2+1D DSTCs are obtained for cell gaps $d = 10 \mu\text{m}$, and the POM video for configuration 3 (bottom right) 2+1D DSTC is obtained for a cell gap $d = 15 \mu\text{m}$. The external drive periodicity $T_E = 0.5\text{s}$. The elapsed time (in units of T_E) and scale bar are marked on the video frames. The transmitting axes of the polarizer and analyser are marked by black double arrows and the slow axis of the retardation plate is marked by a green double arrow.

Supplementary Video 2 | Videos showing director field and scalar order parameter intensity when U close to zero. Numerically simulated director field (left) based on the Landau-de Gennes free energy functional, the background is coloured by the scalar order parameter S where the colouring scheme is the same as Fig. 4. The videos in the right sides are the zoomed-in region marked on the left. The corresponding voltage is marked on the video frames.

Supplementary Video 3 | Phases of DSTCs. POM videos showing the time-symmetry-unbroken phase (left), disordered phase (middle) and co-existence phase (right), respectively. The POM videos are obtained for $d = 5 \mu\text{m}$. The elapsed time and scale bar are marked on the video frames. The transmitting axes of the polarizer and analyser are marked by black double arrows and the slow axis of the retardation plate is marked by a green double arrow.

Supplementary Video 4 | Videos showing DSTCs under different conditions. The POM videos are obtained for $d = 5 \mu\text{m}$, $T_E = 0.35 \text{ s}$ and $U_{\text{max}} = 90 \text{ V}$ (top left), $d = 5 \mu\text{m}$, $T_E = 0.6 \text{ s}$ and $U_{\text{max}} = 90 \text{ V}$ (top middle), $d = 5 \mu\text{m}$, $T_E = 1.0 \text{ s}$ and $U_{\text{max}} = 90 \text{ V}$ (top right), $d = 10 \mu\text{m}$, $T_E = 0.3 \text{ s}$ and $U_{\text{max}} = 50 \text{ V}$ (bottom left), $d = 10 \mu\text{m}$, $T_E = 0.6 \text{ s}$ and $U_{\text{max}} = 50 \text{ V}$ (bottom middle) and $d = 10 \mu\text{m}$, $T_E = 0.9 \text{ s}$ and $U_{\text{max}} = 90 \text{ V}$ (bottom right), respectively. The elapsed time and scale bar are marked on the video frames. The transmitting axes of the polarizer and analyser are marked by black double arrows and the slow axis of the retardation plate is marked by a green double arrow.

Supplementary Video 5 | Formation of the 2+1D DSTC. POM video showing dynamics of the 2+1D DSTC “boil out” from a disordered state. The POM video is obtained for a cell gap $d = 10 \mu\text{m}$, and the external drive periodicity $T_E = 0.5\text{s}$. The elapsed time and scale bar are marked on the video frames. The transmitting axes of the polarizer and analyser are marked by black double arrows and the slow axis of the retardation plate is marked by a green double arrow.

Supplementary Video 6 | Rigidity of the 2+1D DSTC against temporal perturbations. POM video showing the 2+1D DSTC against temporal perturbations ΔT_E randomly distributed within $[-0.2\bar{T}_E, +0.2\bar{T}_E]$ (left) and $[-0.4\bar{T}_E, +0.4\bar{T}_E]$ (right), where $\bar{T}_E = 0.5\text{s}$. The POM videos are obtained for cell gaps $d = 10 \mu\text{m}$. The elapsed time and scale bar are marked on the video frames. The transmitting axes of the polarizer and analyser are marked by black double arrows and the slow axis of the retardation plate is marked by a green double arrow.

Supplementary Video 7 | Emergence and disappearance of a defect region in a 2+1D DSTC. To clearly show the dynamics, we capture the snapshots with temporal interval T_E , and then compile them into a video. The POM video is obtained for a cell gap $d = 10 \mu\text{m}$, and the external drive periodicity $T_E = 0.5\text{s}$. The scale bar is marked on the video frames. The transmitting axes of the polarizer and analyser are marked by black double arrows and the slow axis of the retardation plate is marked by a green double arrow.

Supplementary Video 8 | Healing of 1+1D DSTC after generating a defect by a laser tweezer. The POM video is obtained for a cell gap $d = 5 \mu\text{m}$, and the external drive periodicity $T_E = 0.5\text{s}$.

The elapsed time and scale bar are marked on the video frames. The transmitting axes of the
polarizer and analyser are marked by black double arrows and the slow axis of the retardation plate
is marked by a green double arrow.

**Supplementary Video 9 | Video showing the 1+1D DSTC over long driving time periods.** The
POM video is obtained for a cell gap $d = 5 \mu\text{m}$, and the external drive periodicity $T_E = 0.5\text{s}$. The
elapsed time and scale bar are marked on the video frames. The transmitting axes of the polarizer
and analyser are marked by black double arrows and the slow axis of the retardation plate is marked
by a green double arrow.

**Supplementary Video 10 | An ordered quasi-hexagonal lattice under external Floquet drives.**
The external drive periodicity $T_E = 0.5\text{s}$. The elapsed time (in units of T_E) and scale bar are marked
on the video frames. The transmitting axes of the polarizer and analyser are marked by white
double arrows in the left and black double arrows in the right, and the slow axis of the retardation
plate is marked by a green double arrow.

Nature Communications manuscript NCOMMS-24-59238C
**Emergent discrete space-time crystal of Majorana-like quasiparticles in chiral
liquid crystals**

H. Zhao, R. Zhang and I. I. Smalyukh

Summary of Changes and Responses to Requests/Remarks

Report of Referee 2:

I reviewed the manuscript on “Emergent discrete space-time crystal of Majorana-like quasiparticles in chiral liquid crystals” in its entirety, in light of all the changes made due to all review feedback in the last round. The result is a well rounded overview of the period-doubling phenomena found in experiment. While I am not an expert on time crystals, and therefore hesitant to apply this terminology, the computational modeling based on Frank-Oseen and Landau-de Gennes theory appears to capture the overall dynamics well.

In particular, my request for a description suitable to a more general audience is addressed satisfactory. I have no further remarks that would improve the manuscript within a reasonable time frame.

Authors:

We thank the Reviewer #2 for the recommendation and the positive reports.

Report of Referee 3:

Report of Referee 4:

The author's have significantly improved their paper, adding important discussions and clarifying some important aspects of their work. At this point, I understand their experiment and observations and I find them novel and interesting to the broader audience. However, I still find some points confusing or ill-explained in the manuscript. Pending addressing these points, I support publication in Nature Communications.

Authors: We appreciate the reviewer for this positive report saying that “*The author's have significantly improved their paper, adding important discussions and clarifying some important aspects of their work*” and our experiment and observations are “*novel and interesting to the broader audience*”. We also thank the reviewer for the helpful suggestions, which we have carefully addressed in the revised manuscript.

Reviewer #4:

Clarification of the definition of the Majorana-like quasiparticles

1) I remain confused about the definition of the Majorana quasi-particle and its relationship to the the director orientation profile. The authors emphasize that the homotopy class of the director in 3D is $\pi^1(S^2/\mathbb{Z}_2) = \mathbb{Z}_2$, which leads either defect or no-defect. In this language the origin of the Majorana is somewhat clear, since two defects annihilate one another -- this is also the discussion within the PNAS cited by the paper.

2) Despite being equivalent, when the applied field is large, the director appears to be restricted to the zx plane and the winding number becomes well-defined to the point that one can assign $\pm 1/2$ charge to each "Majorana-like" particle. The authors emphasize how the DTC dynamics can be understood in terms of the annihilation and creation of pairs of such pairs of particles (which is the more general language are the Majorana quasiparticles)

3) This begs three questions:

1) This discussion all occurs in the translationally invariant case, where these defects are not point-like but rather extended objects along the y direction. In this case the director orientation is ill-defined, not on a single point, but along a line, changing the dimensionality of the defect. In this case, why does the same characterization of the defect still persist?

Authors: We thank the reviewer for these questions and helpful remarks. The homotopy group $\pi_1(S^2/\mathbb{Z}_2) = \mathbb{Z}_2$ (Ref. [44]) characterizes both the singular point defects in 2D, where director is allowed to adopt 3D orientations according to the nonpolar director's order parameter space S^2/\mathbb{Z}_2 , as well as the translationally invariant line defects in 3D samples, in which case the cross-section has this configuration. The latter are found experimentally in our system, as well as are modelled numerically. Here π_1 refers to the measuring circle (one-sphere), which is the same for translationally invariant line defects in 3D and point defects in 2D. Our system is slightly different from that in the PNAS article but the same principles apply and we vividly see the dynamics of transformations. In the revised manuscript, we slightly expand the discussion and further clarify this matter.

Reviewer #4:

2) The authors also emphasize (in Page 6) how the nature of the domain wall is also captured by the same homotopy class. These, however, are extended two dimensional objects. How does the homotopy class of the solitons affects our understanding of their dynamics? Does it simply state that any two solitons can annihilate one another? At the same time, is there some important connection between the presence of the aforementioned defect lines and the solitons (more on this below)? And what is the relationship between the solitons and the Neel/Bloch domains walls emphasized in Fig4?

Authors: We thank the reviewer for these questions, which help us in making our writing more accessible to the readership. Because the one-dimensional space \mathbb{R}^1 can be compactified on S^1 (one-sphere mathematical notation, which is a circle, corresponding to π_1) for the case of uniform embedding background (one-point compactification), the 1D solitons can be classified by the same first homotopy group $\pi_1(S^2/\mathbb{Z}_2) = \mathbb{Z}_2$. It is natural that the $\pi_1(S^2/\mathbb{Z}_2) = \mathbb{Z}_2$ solitonic walls are terminated by the topologically analogous line defects $\pi_1(S^2/\mathbb{Z}_2) = \mathbb{Z}_2$ described within the same homotopy theory classification. In our observations, the topological solitons and singular lines inter-transform between geometrically different but topologically equivalent states synchronously. This is because different geometric states of the soliton (Néel or Bloch type) and the states of singular defect lines with different cross-sectional structures (which, in a purely 2D system that allow for only 2D orientations of director as at some values of applied field, would correspond to defects of opposite-sign winding number, but are topologically analogous and deformable to one another when 3D rotations of director are allowed, e.g. at other values of instantaneous voltage of our driving) are all

described by $\pi_1(S^2/\mathbb{Z}_2) = \mathbb{Z}_2$. The solitons have geometric embodiments of Néel and Bloch types and also the Majorana quasiparticle nature, similar to the singular defect lines. In the revised manuscript, we slightly expand the discussion and further clarify this matter.

Reviewer #4:

3) The numerical evidence (and ability to peer into the dynamics of the director) seems to be limited to the 1+1D case. In this case, I understand how the line defects (labeled by $\pm 1/2$) paint a simple description of the dynamics. However, the authors extend their claims of the Majorana description to the 2+1D DTC (see abstract quote below), where the authors have no picture or description in terms of the director topological defect.

In the report, they have also stated that "these phenomena are rather complex and more disciplinary, with the power of numerical modelling still insufficient to capture details that could be presented in an accessible form".

To this end, I think that either additional evidence / descriptions are added to the paper, or the claim of a time crystal driven by the dynamics of Majorana-like particles should be restricted to the analysis of the 1+1D case (upon a clarification of the above point)

> "Here we describe the observation of classical analogues of both 1+1-dimensional and 2+1-dimensional discrete space-time crystals in a liquid crystal system driven by a Floquet electrical signal. These classical time crystals comprise particle-like structural features and exist over a wide range of temperatures and electrical driving conditions. The phenomenon-enabling period-doubling effect comes from their topological Majorana-like quasiparticle features"

Authors: We thank the reviewer for these helpful questions and remarks. In the revised abstract and main text, we further clarified that the 1+1D space-time crystals are related to the reported dynamic transformations of the Majorana-like quasiparticle features. In the revised abstract we updated the statement to read: "*The phenomenon-enabling period-doubling effect in 1+1-dimensional discrete space-time crystals comes from their topological Majorana-like quasiparticle features.*"

In the manuscript's conclusions and future perspectives part we now also mention that: "*The future studies should extend the detailed analyses of the reported period-doubling effect in 1+1D space-time crystals to high-dimensional cases, including 2+1D that we already observed experimentally.*" We appreciate referee's remarks that guided these improvements of presentation of our findings.

Reviewer #4:

Confusing point about the initial state:

In my previous reply I alluded to some confusion with regards to the relationship between the profile in Fig 3 and 4 which I now better understand how the two correspond to the initial state vs a description of the dynamical process, respectively. From reading the text, I still find these two plots and their relationship hard to parse (magnified by the lack of description of Fig3 in the main text, something that is more extreme for Fig 2 which I would recommend moving to the Methods or SM).

Authors: We thank the reviewer for these helpful remarks. We have added additional details in the description of figures, including a more detailed discussion of Fig. 2 while fully accounting for this feedback.

Reviewer #4:

However, even with this clarification, the role of the initial state remains confusing. Let me first start by laying out my understanding/confusion.

1) In Fig 3 the authors emphasize the presence of soliton domain whose winding along the XY plane is alternative between $+$ and $-\pi$ and there are no defect lines of the form presented in Fig 4 (the $\beta = \pm 1/2$ defects). Indeed, in Fig.4 it seems like the directors predominantly point along the z axis even when the applied field is small (around $t = 0.5TE$ Fig 4h).

2) This suggests that the first period is very important because it takes a symmetry breaking state (lattice of solitons) and maps it into a completely different symmetry breaking state (a lattice of Néel domain walls) which are embedded in a different plane of the liquid crystal (note that in the response, the authors still called both domain walls, Néel domains, although they emphasize they are distinct because of the plane of the orientation).

Authors: We thank the reviewer for these helpful remarks. We provide additional details in the description of the transformation between the initial state and the time crystalline structures that are the focus of our study. The sample has perpendicular boundary conditions on confining substrates, defining a topologically neutral background. The director structures are topologically neutral of the initial state as it consists of alternating $+\pi$ and $-\pi$ domain-wall solitons in the x-y plane. In the 1+1D DSTC phase, pairs of disclination lines can be annihilated (Fig. 4l), which also indicates a topologically neutral state, dictating that the topological objects' invariants self-compensate. Both in simulations and experiments, we observe that the initial state transforms into the 1+1D DSTC state within tens of driving periods. Indeed, the symmetry breaking becomes different during the time crystal formation. We have now mentioned this detail and appreciate this referee's remark.

Reviewer #4:

3) The different defects and their transformations are made extra confusing when looking at Fig1d where the 1+1D response looks the same as the initial state presented in Fig3a, however, the authors are emphasizing that the director ordering is actually very different (Fig3g vs Fig4h).

4) Moreover, it seems to be the case that the same initial state (1D symmetry breaking) induces the dynamics of the 2+1D state ("When we set the 1+1D DSTC spatial structures as initial configurations and apply the time-dependent Floquet electrical field (sawtooth wave, maximum amplitude $U_{max} = 0.2V$ and temporal periodicity $TE = 0.5s$), the 2+1D DSTC (configuration 2) spatial configurations spontaneously emerge (Fig. 3m-o) in each drive"). This point is stated in the methods, but further emphasizes that there is a lot of complexity in the way that the initial state induces the time crystalline behavior.

This distinction between the initial state and the dynamical state is very important, but it is not addressed. Much like a careful discussion of the nature of the defects (and a clearer justification of their Majorana nature), it warrants more space in the text to guide the reader through the differences between the different defects considered (the two domain walls and the Majorana defects), what their relationship is and how to understand the transformation between the initial state and the dynamical steady state.

Authors: We thank the reviewer for these helpful remarks. Taking into consideration these questions and comments, we have explicitly clarified the details of initial states and dynamic steady states, topological defects and topological solitons (together with the Majorana nature) in the revised manuscript. We have also moved Fig. 3a–d to the Supplementary Materials following the reviewer's suggestion, which we appreciate.

Reviewer #4:

Minor points:

- In S7, the authors emphasize having data averaged over 1500 drives (I assume this means 1500 periods since the caption emphasized different driving durations), yet $G(t)$ only goes to 100. This axis should be extended to help emphasize the nature of the decay.

Authors: We thank the reviewer for this helpful remark. We have extended the time axis in the revised Supplementary Fig. S8 to 150 drives - the results are consistent with the previous ones. We also mention that extending to larger number of drives will require enhancing system controls to preclude issues like thermally or vibrations-induced drift, etc.

Reviewer #4:

- Ln 9. Page 1: "*period of the driven system*" should read "*period of the system's dynamics*"

Authors: We thank the reviewer for this helpful remark, which we accounted for.

We believe that the above-described revisions address all remarks and account for all suggestions that were raised by the original manuscript. We also feel that they improve the paper significantly. We thank the Referees for the thoughtful valuable comments that directed these changes.

Sincerely,
Prof. Ivan I. Smalyukh, Department of Physics, 390 UCB
University of Colorado at Boulder
Phone: 303-492-7277 (office); Email: Ivan.Smalyukh@colorado.EDU
<http://www.colorado.edu/physics/SmalyukhLab/index.html>

[revised manuscript text omitted]

 $10 \times 10^{-12} \text{ N}$, $W = 10^{-4} \text{ J m}^{-2}$ and $\gamma = 77 \text{ mPas}$. While this approach captures many fine
 features of quasi-static configurations seen in experiments for some parts of phase diagrams, it
 does not yield the dynamic features of the time-crystalline structures, this is because the
 numerical simulation based on Frank-Oseen free energy cannot simulate disclinations or low
 scalar order parameter regions^{35,40}, for which we adopt a different approach described below.

**Numerical modeling based on the Landau–de Gennes free energy functional**

The quasi-static spatial structures of 1+1D DSTC (Fig. 3a-d, Supplementary Fig. S1) can be
 revealed by the Landau–de Gennes free energy functional, in which flexoelectric terms and ion-
 induced electric field are incorporated. The static or quasi-static spatial structures (Fig. 3)
 obtained via vectorial or tensorial modeling can be used as initial states of the tensor-based
 numerical simulations to model the dynamic behaviours. Only tensorial modelling is capable of
 capturing the details of time crystalline structural evolution because of the singular nature of the
 observed defect lines.

The tensor order parameter is defined as $\mathbf{Q} = S(\mathbf{nn} - \mathbf{I}/3)$, where S is the LC's scalar
 order parameter and \mathbf{I} is the identity matrix, and the energy cost of the spatial deformations of
 $\mathbf{Q}(\mathbf{r})$ can be expressed as:

$$11 \quad F_{\text{elastic}}^{\text{LdG}} = \int d^3\mathbf{r} \left\{ \frac{L_1}{2} \frac{\partial Q_{ij}}{\partial x_k} \frac{\partial Q_{ij}}{\partial x_k} + \frac{L_2}{2} \frac{\partial Q_{ij}}{\partial x_j} \frac{\partial Q_{ik}}{\partial x_k} + \frac{L_3}{2} \frac{\partial Q_{ij}}{\partial x_k} \frac{\partial Q_{ik}}{\partial x_j} + \frac{L_6}{2} Q_{ij} \frac{\partial Q_{kl}}{\partial x_i} \frac{\partial Q_{kl}}{\partial x_j} \right. \\ 12 \quad \left. + \frac{4\pi}{p} L_4 \varepsilon_{ikl} Q_{ij} \frac{\partial Q_{lj}}{\partial x_k} \right\}, \quad (4)$$

where ε_{ikl} is the Levi-Civita symbol and L_i 's are the elasticity parameters. In addition, the
 Landau–de Gennes free energy functional includes thermotropic terms that describe the nematic-
 isotropic transition of the LC:

$$16 \quad F_{\text{thermotropic}}^{\text{LdG}} = \int d^3\mathbf{r} \left\{ \frac{A}{2} \left(1 - \frac{U_{\text{LdG}}}{3} \right) Q_{ij} Q_{ji} - \frac{AU_{\text{LdG}}}{3} Q_{ij} Q_{jk} Q_{ki} + \frac{AU_{\text{LdG}}}{4} (Q_{ij} Q_{ji})^2 \right\}, \quad (5)$$

where A and U_{LdG} are the nematic material parameters. When an external electric field $\mathbf{E}_{\text{external}}$ is
 applied, the total electric field \mathbf{E} in the LC is a superposition of $\mathbf{E}_{\text{external}}$ and \mathbf{E}_{ion} , where \mathbf{E}_{ion} is the
 ion-induced electric field, and can be calculated by the Poisson equation²⁹:

$$20 \quad \nabla \cdot (\boldsymbol{\varepsilon} \cdot \mathbf{E} + \mathbf{P}_{\text{flexo}}) = \rho_{\text{el}}, \quad (6)$$

where $\boldsymbol{\varepsilon}$ is dielectric tensor, $\mathbf{P}_{\text{flexo}}$ is polarization field due to flexoelectric, and ρ_{el} is the ionic
 charge satisfying $\nabla \cdot (\boldsymbol{\sigma} \cdot \mathbf{E}) = -\partial_t \rho_{\text{el}}$, with $\boldsymbol{\sigma}$ being the conductivity tensor. The dielectric and
 conductivity tensor are related to the Q-tensor via $\boldsymbol{\varepsilon} = \bar{\boldsymbol{\varepsilon}}\mathbf{I} + \varepsilon_a^{\text{mol}}\mathbf{Q}$ and $\boldsymbol{\sigma} = \bar{\boldsymbol{\sigma}}\mathbf{I} + \sigma_a\mathbf{Q}$, where $\bar{\boldsymbol{\varepsilon}}$
 and $\bar{\boldsymbol{\sigma}}$ are the mean dielectric and conductivity constants, respectively, and $\varepsilon_a^{\text{mol}}$ and σ_a are the
 dielectric anisotropy and conductivity anisotropy, respectively.

The free energy is supplemented by the following electric coupling terms:

$$7 \quad F_{\text{electric}}^{\text{LdG}} = \int d^3\mathbf{r} \left\{ -\frac{1}{2}\varepsilon_0\bar{\boldsymbol{\varepsilon}}E_i^2 - \frac{1}{3}\varepsilon_0\varepsilon_a^{\text{mol}}Q_{ij}E_iE_j + \zeta_1\frac{\partial Q_{ij}}{\partial x_j}E_i + \zeta_2Q_{ij}\frac{\partial Q_{jk}}{\partial x_k}E_i \right\}, \quad (7)$$

where ζ_i 's are the flexoelectric constants. The first two terms describe dielectric coupling between
 \mathbf{Q} and the electric field \mathbf{E} , and the last two terms describe the flexoelectric effect.

The surface free energy describing the surface anchoring at the substrates reads

$$11 \quad F_{\text{surface}}^{\text{LdG}} = -\int d^2\mathbf{r} \frac{W}{2} (Q_{ij} - Q_{ij}^{(0)})^2, \quad (8)$$

where $Q_{ij}^{(0)}$ defines the preferred orientation and order of LC at the surfaces, corresponding to
 perpendicular boundary conditions in our experiments. The total Landau–de Gennes free energy

$F = F_{\text{elastic}}^{\text{LdG}} + F_{\text{thermotropic}}^{\text{LdG}} + F_{\text{electric}}^{\text{LdG}} + F_{\text{surface}}^{\text{
[revised manuscript text omitted]

H. Zhao, R. Zhang and I. I. Smalyukh

Responses to Referee Remarks

Report of Referee 4:

The authors have carefully engaged with the comments I have made in previous reports and have expanded and clarified their observations and conclusions.

At present, I do not have more comments that I think are crucial for the understanding of the paper, and, as such, I am happy supporting the publication of the manuscript in Nature Communications.

Authors:

We thank the Reviewer #4 for the recommendation and the positive reports.

Sincerely,

Prof. Ivan I. Smalyukh, Department of Physics, 390 UCB

University of Colorado at Boulder

Phone: 303-492-7277 (office); Email: Ivan.Smalyukh@colorado.EDU

<http://www.colorado.edu/physics/SmalyukhLab/index.html>